# Cell-type specialization is encoded by specific chromatin topologies

Warren Winick-Ng[1,22 ✉], Alexander Kukalev[1,22], Izabela Harabula[1,2,22], Luna Zea-Redondo[1,2,22], Dominik Szabó[1,2,22], Mandy Meijer[3], Leonid Serebreni[1,16], Yingnan Zhang[4], Simona Bianco[5], Andrea M. Chiariello[5], Ibai Irastorza-Azcarate[1], Christoph J. Thieme[1], Thomas M. Sparks[1], Sílvia Carvalho[1,6,7,8], Luca Fiorillo[5], Francesco Musella[5], Ehsan Irani[1,9], Elena Torlai Triglia[1,17], Aleksandra A. Kolodziejczyk[10,11,18], Andreas Abentung[12,19], Galina Apostolova[12], Eleanor J. Paul[13,20,21], Vedran Franke[14], Rieke Kempfer[1,2], Altuna Akalin[14], Sarah A. Teichmann[10,11], Georg Dechant[12], Mark A. Ungless[13], Mario Nicodemi[5,9], Lonnie Welch[4], Gonçalo Castelo-Branco[3,15] & Ana Pombo[1,2,9 ✉]

The three-dimensional (3D) structure of chromatin is intrinsically associated with gene regulation and cell function[1–3]. Methods based on chromatin conformation capture have mapped chromatin structures in neuronal systems such as in vitro differentiated neurons, neurons isolated through fluorescence-activated cell sorting from cortical tissues pooled from different animals and from dissociated whole hippocampi[4–6]. However, changes in chromatin organization captured by imaging, such as the relocation of *Bdnf* away from the nuclear periphery after activation[7], are invisible with such approaches[8]. Here we developed immunoGAM, an extension of genome architecture mapping (GAM)[2,9], to map 3D chromatin topology genome-wide in specific brain cell types, without tissue disruption, from single animals. GAM is a ligation-free technology that maps genome topology by sequencing the DNA content from thin (about 220 nm) nuclear cryosections. Chromatin interactions are identified from the increased probability of co-segregation of contacting loci across a collection of nuclear slices. ImmunoGAM expands the scope of GAM to enable the selection of specific cell types using low cell numbers (approximately 1,000 cells) within a complex tissue and avoids tissue dissociation[2,10]. We report cell-type specialized 3D chromatin structures at multiple genomic scales that relate to patterns of gene expression. We discover extensive 'melting' of long genes when they are highly expressed and/or have high chromatin accessibility. The contacts most specific of neuron subtypes contain genes associated with specialized processes, such as addiction and synaptic plasticity, which harbour putative binding sites for neuronal transcription factors within accessible chromatin regions. Moreover, sensory receptor genes are preferentially found in heterochromatic compartments in brain cells, which establish strong contacts across tens of megabases. Our results demonstrate that highly specific chromatin conformations in brain cells are tightly related to gene regulation mechanisms and specialized functions.

To explore how genome folding is related to cell specialization, we applied immunoGAM to mouse brain tissue slices and analysed three cell types with diverse functions (Fig. 1a): oligodendroglia (oligodendrocytes and their precursors (OLGs)) from the somatosensory cortex; pyramidal glutamatergic neurons (PGNs) from the cornu ammonis 1 (CA1) of the dorsal hippocampus; and dopaminergic neurons (DNs) from the ventral tegmental area (VTA) of the midbrain. OLGs are important for neuronal myelination and circuit formation[11], whereas PGNs are important for temporal and spatial memory formation and

consolidation[12], and DNs are activated during cue-guided reward-based learning[13]. Publicly available GAM data from mouse embryonic stem (mES) cells[9] were used for comparison (Supplementary Table 1).

We selected cell types from brain tissue slices by immunofluorescence with cell marker antibodies before genomic extraction (Fig. 1b). A detailed flowchart of immunoGAM quality control (QC) measures and normalization is shown in Extended Data Fig. 1a–d and Supplementary Table 2. GAM contact matrices, each from about 850 cells, had low biases in GC content and mappability (Extended Data Fig. 2a–c). We

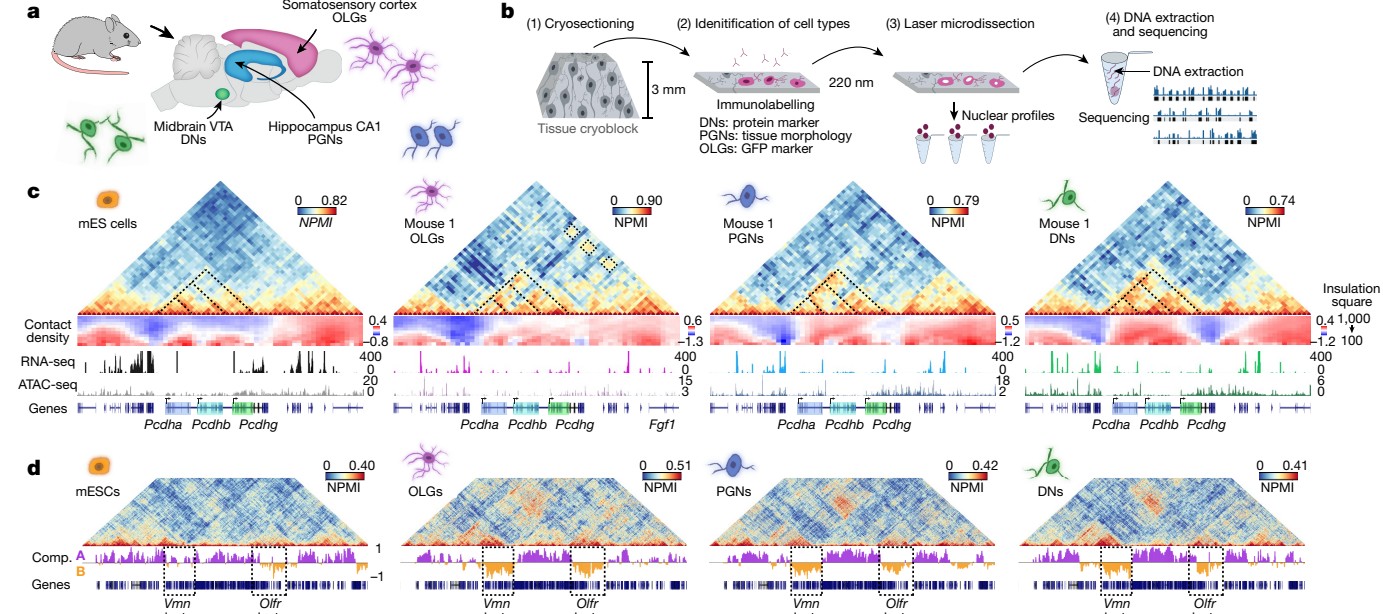

**Fig. 1 | ImmunoGAM captures cell-type-specific chromatin contacts in the mouse brain. a**, ImmunoGAM was applied to three brain cell types: OLGs, DNs and PGNs (one independent biological replicate for OLGs and two replicates for DNs and PGNs). **b**, Schematic of the ImmunoGAM workflow. OLGs were selected by immunolabelling with GFP, DNs with tyrosine hydroxylase and PGNs using tissue morphology. Nuclear profiles were laser microdissected, each from a single cell, with three collected together, as described for multiplex-GAM[9]. **c**, Example of cell-type-specific contact differences at the *Pcdh* locus (chromosome 18: 36–39 Mb). GAM matrices represent co-segregation frequencies of 50-kb genomic windows using normalized pointwise mutual information (NPMI). Dashed lines illustrate cell-type differences. NPMI scales range between 0 and 99th percentile per cell type. Contact density heatmaps represent insulation scores using 100–1,000 kb square sizes. RNA-seq and ATAC-seq tracks represent normalized pseudobulk reads from scRNA-seq and scATAC-seq, respectively, except for bulk ATAC-seq from mES cells. **d**, Strong contacts between *Vmn* and *Olfr* receptor gene clusters on chromosome 17 (0–60 Mb) within B compartments (Comp.), separated by ~35 Mb, are observed in brain cells but not in mES cells. Compartments A and B were classified using normalized PCA eigenvectors[2].

calculated local contact densities and topological domains using the insulation square method[14], and calculated compartments associated with open chromatin (compartment A) and closed chromatin (compartment B) using principal component analysis (PCA)[2] (Supplementary Tables 3–5).

As an example of cell-type-specific organization, we considered the *Pcdh* locus, which contains three clusters of cell adhesion genes (*Pcdha*, *Pcdhb* and *Pcdhg*) and occupies two topologically associating domains (TADs) in mES cells, as previously described[15] (Fig. 1c, see Extended Data Fig. 3a for replicates). Mapping contact densities using 100–1,000 kb insulation squares showed that the locus is generally open above 500 kb. Higher expression of *Pcdha* and *Pcdhb* coincides with increased long-range contacts between the three clusters in neurons[16] and OLGs[17] and with additional long-range contacts with the highly expressed *Fgf1* gene in OLGs. We also discovered contacts spanning tens of megabases in brain cells. For example, strong contacts connected two regions approximately 3- and 5-Mb wide, separated by ~35 Mb, which contained clusters of vomeronasal (*Vmn*) and olfactory (*Olfr*) receptor genes (Fig. 1d, see Extended Data Fig. 3b for replicates). Thus, the application of immunoGAM in specific brain cell types reveals large rearrangements in 3D chromatin architecture at short-range and long-range genomic lengths.

To further investigate how cell-type-specific 3D genome topologies relate to gene expression and chromatin accessibility, we produced or collected published single-cell RNA sequencing (scRNA-seq) data and single-cell assay for transposase-accessible chromatin with high-throughput sequencing (ATAC-seq) data from mES cells, the cortex, the hippocampus and the midbrain (Methods, Extended Data Fig. 4, Supplementary Table 6). After selecting cell populations equivalent to those captured by immunoGAM, we compiled cell-type-specific pseudobulk RNA-seq and ATAC-seq datasets.

## TADs extensively rearrange between cell types

Complex and extensive cell-type-specific changes in TAD-level contacts were frequent, for example, at a 4-Mb region that contains *Scn* genes that encode sodium voltage-gated channel subunits (Fig. 2a, see Extended Data Fig. 5a for replicates). We obtained a total of approximately 2,300 TADs across cell types, with a median length of about 1 Mb, which is in line with previous reports[6] (Extended Data Fig. 5b). Although pairwise comparisons of TAD border positions confirmed previous levels of conservation[4,6] (78–89%; Extended Data Fig. 5c), multiway comparisons showed high cell-type specificity (Fig. 2b, see Extended Data Fig. 5d for sparser combinations). One-third of the borders were unique and significantly more insulated in other cell types (Extended Data Fig. 5e), with some variability noted between biological replicates (59–65%) (Extended Data Fig. 5f). By contrast, only 8% of the total set of borders was shared by brain cells and 14% by all cell types. Shared borders showed significantly stronger insulation in brain cells than in mES cells (Extended Data Fig. 5g), which suggests that there is structural stabilization after terminal differentiation. Unique boundaries often contained expressed genes (52–55% in brain cells, 38% in mES cells) (Extended Data Fig. 5h) and genes with enriched Gene Ontology (GO) terms relevant to the specialized cell type (Fig. 2c, Supplementary Table 7), such as 'membrane depolarization' and 'cognition' in PGNs or genes important for dopaminergic differentiation and dopamine synthesis in DNs.

## Long neuronal genes melt in brain cells

Many neuronal genes involved in specialized functions are long (>300 kb) and produce many isoforms owing to complex RNA processing[18]. Chromatin reorganization was most apparent at long genes

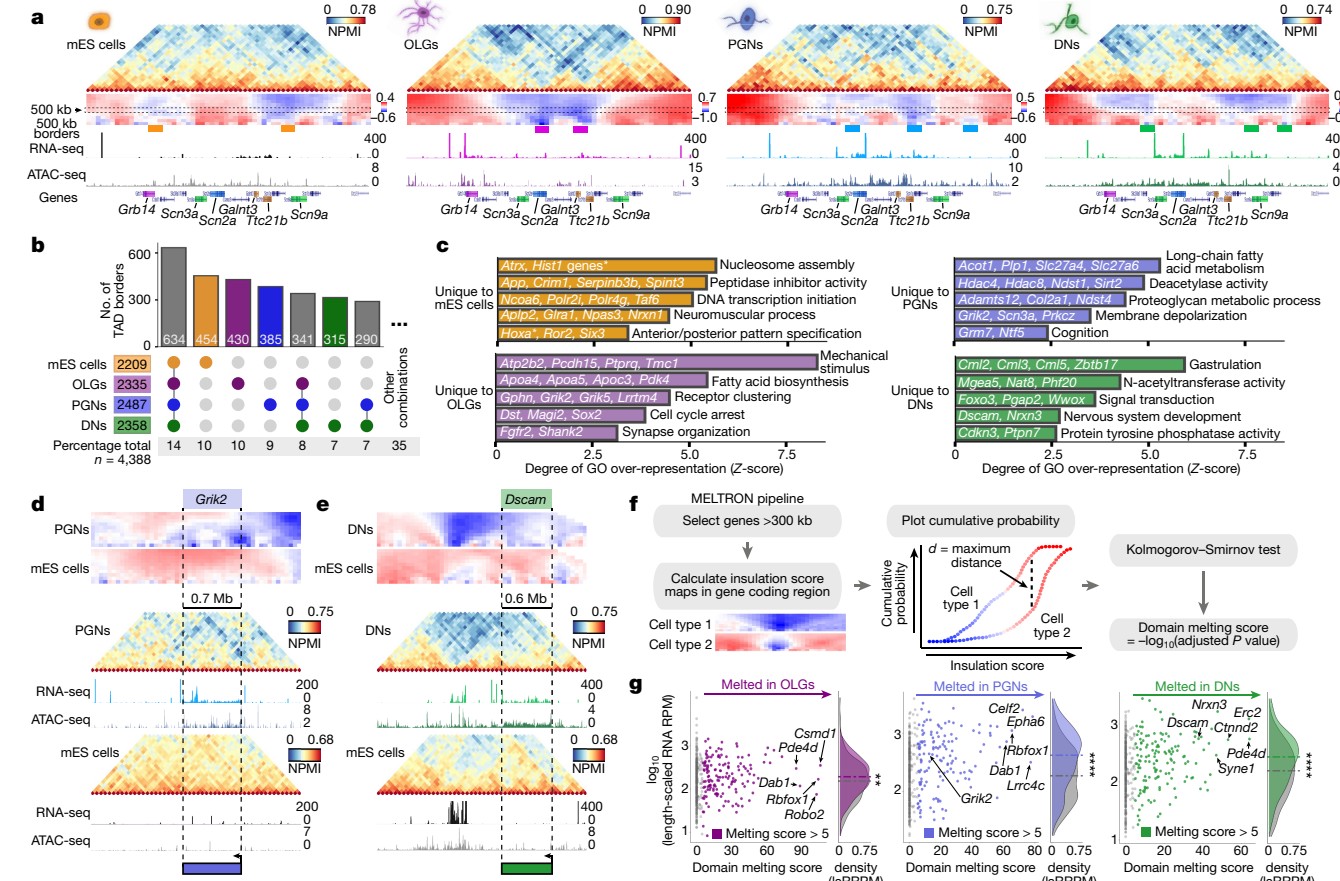

**Fig. 2 | Chromatin domains rearrange extensively in brain cells, notably at long genes that undergo melting events. a**, Example of cell-type-specific contacts at genomic regions (chromosome 2: 64.3–67.3 Mb) with differential expression. Dashed boxes represent 500 kb insulation scores used to determine TAD boundaries (indicated with coloured boxes below). Replicate 1 is shown for brain cells. **b**, UpSet plots representing multiway TAD boundary comparisons show extensive cell-type specificity. Boundaries were defined as 150 kb genomic regions centred on the lowest insulation score windows and were considered different when separated by >50 kb edge-to-edge. **c**, Cell-type-specific borders contain genes with GO terms relevant for cell functions. The top four GO terms were the most enriched, and the fifth was selected (over-representation measured by Z-score; one-sided Fisher's exact permuted P values < 0.01).

Asterisk indicates multiple *Hist1* genes. **d**, **e**, *Grik2* and *Dscam* overlap with cell-type-specific TAD borders and extensively decondense, or 'melt', in PGNs and DNs, respectively. **f**, The MELTRON pipeline was applied at long genes (>300 kb, 479 genes) to determine melting scores from contact density maps that represent insulation score values using 100–1,000 kb squares. Genes were considered to melt if the melting score computed across their coding region was >5 ($P < 1 \times 10^{-5}$; one-sided Kolmogorov–Smirnov testing using maximum distances between distributions). **g**, Melting associates with higher expression, especially in PGNs and DNs (two-sided Wilcoxon rank-sum test; **$P < 0.01$, ****$P < 0.0001$; P values from left to right, $P = 3.5 \times 10^{-3}$, $P = 1.8 \times 10^{-8}$, $P = 8.3 \times 10^{-6}$). lsRRPM, length-scaled RNA reads per million; RPM, reads per million.

in both PGNs and DNs (Fig. 2d, e). For example, *Grik2* loses contact density in PGNs compared to mES cells, especially around the transcription start site (TSS) and transcription end site (TES) (Fig. 2d). By contrast, *Dscam* decondenses across its entire gene body in DNs (Fig. 2e). To assess whether decondensation relates to the expression of long genes, we compared the insulation of the most and least expressed long genes (Extended Data Fig. 5i). Highly expressed genes were significantly less insulated at TSSs and TESs and throughout gene bodies in both DNs and PGNs, but not in OLGs or mES cells. The general contact loss at highly expressed long neuronal genes is reminiscent of the decondensation, or 'melting', observed by microscopy at polytene chromosome puffs[19] or tandem gene arrays[20].

To detect melting genome-wide in an unbiased manner, we devised the MELTRON pipeline. MELTRON calculates a 'melting score' as the significant difference between cumulative probabilities of insulation scores across a range of genomic scales (100–1,000 kb) between two cell types and within regions of interest, here defined as all (479) long genes (Fig. 2f). We found 120–180 melting genes with melting scores of >5 (Kolmogorov–Smirnov test, $P < 1 \times 10^{-5}$) between brain cells and mES cells (Fig. 2g, Supplementary Table 8). *Grik2* had melting scores

of 12 and 26 in PGNs (replicates 1 and 2, respectively), whereas *Dscam* had scores of 38 and 50 in DNs (replicates 1 and 2, respectively) and *Magi2* had a score of 73 in OLGs (Extended Data Fig. 6a, b). Melting scores in the PGN and DN replicates correlated well (Extended Data Fig. 6c).

Melting genes were significantly more transcribed and showed higher chromatin accessibility than non-melting long genes, especially in PGNs and DNs (Fig. 2g, Extended Data Fig. 6d–f). Of interest, many top (3%) melting genes (24 out of 44) are sensitive to topoisomerase I inhibition in ex vivo neuronal cultures[21], which was in contrast to 16% (42 out of 261) with intermediate melting scores or 16% of non-melting genes (Extended Data Fig. 6g). This result suggests that extensive melting of long genes is associated with the resolution of topological constraints[21]. Melting genes often belonged to compartment A in both mES cells and the corresponding brain cell (43–58%), especially when highly transcribed in both cell types (Extended Data Fig. 6h). Genes melting in OLGs and DNs were less likely to be lamina-associated or nucleolus-associated in mES cells, whereas PGNs did not show any preferred association (Extended Data Fig. 6i, j). Therefore, melting of long genes is not trivially associated with a transition from a heterochromatic state in mES cells

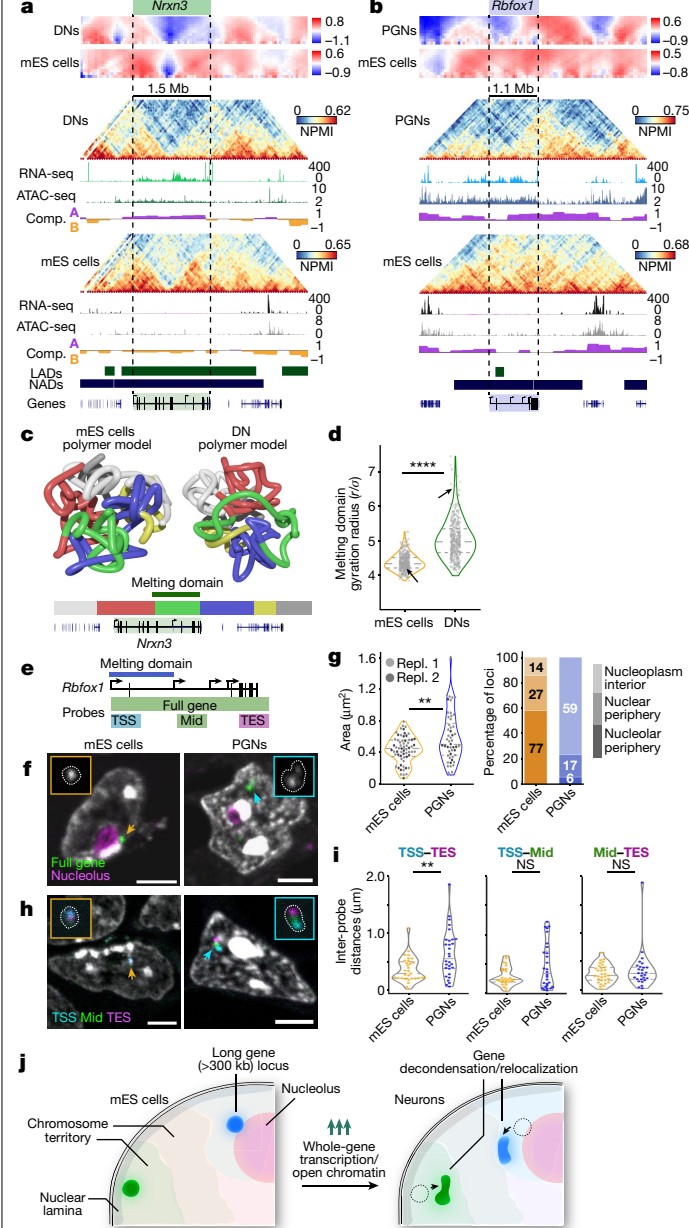

Fig. 3 | Extensive decondensation and relocalization of highly expressed long neuronal genes. a, b, Examples of two melting genes. Nrxn3 occupies two dense TADs in mES cells but melts in DNs where it is most highly expressed and accessible (a; chromosome 12: 87.6–92.4 Mb). Rbfox1 is highly condensed in mES cells and melts in PGNs where it is highly expressed and accessible (b; chromosome 16: 4.8–9.8 Mb). Compartment tracks are shown for each cell type, and published lamina-associated domains (LADs[47]) and nucleolus-associated domains (NADs[48]) for mES cells. c, Polymer models show extensive Nrxn3 melting in DNs compared to mES cells. Colour bars shows DN domain positions. d, Gyration radii of green melting domains are significantly higher in DNs than in mES cells (****$P = 1.1 \times 10^{-92}$; two-sided Mann–Whitney test, $n = 450$). Arrows indicate positions of exemplar models. e, Genomic regions covered by cryo-FISH probes across the entire Rbfox1 gene, or targeting the gene TSS, middle of the coding region (Mid) or TES (Supplementary Table 11 contains the probe list). f, Rbfox1 (pseudocoloured green) occupies small, rounded foci in mES cells, often at the nucleolus periphery (immunostained for nucleophosmin 1, ref.[49]; pseudocoloured purple). In PGNs, Rbfox1 occupies larger, decondensed foci away from nucleoli. Arrows indicate Rbfox1 foci in mES cells (orange) and PGNs (blue). Scale bars, 3 μm. g, Rbfox1 occupies significantly larger areas in PGNs than in mES cells (**$P = 0.008$; two-sided Mann–Whitney test; two experimental replicates (Repl. 1 and Repl. 2) with $n = 13, 39$ and $38, 25$ respectively). Most Rbfox1 foci localize at the nucleolar periphery in mES cells, but away from the nucleolus in PGNs. h, Cryo-FISH experiments that target TSS, Mid and TES regions of Rbfox1 (pseudocoloured cyan, green, purple) show extensive separation in PGNs compared with mES cells. Arrows indicate Rbfox1 foci in mES cells (orange) and PGNs (blue). Scale bars, 3 μm. i, The TSS and TES regions of Rbfox1 are significantly more separated in PGNs than mES cells (two-sided Mann–Whitney test; **$P < 0.01$; from left to right, $P = 0.003, P = 0.179, P = 0.331$; NS, not significant). j, Schematics summarizing the melting of long genes in neurons, which is accompanied by locus relocalization away from repressive nuclear landmarks.

models in mES cells and DNs from GAM matrices (Fig. 3c, Supplementary Tables 9 and 10). 3D models were validated by reconstructing in silico GAM matrices (Extended Data Fig. 7c). mES cell models showed intermingled globular domains, including the green and red domains that contain Nrxn3 (Supplementary Video 1, see Extended Data Fig. 7d for additional examples). In DNs, the melted green domain becomes highly extended and has high gyration radii (Fig. 3c, d, Supplementary Video 2), while the upstream (grey) and downstream (blue) domains condense (Fig. 3a, Extended Data Fig. 7e).

Next, we applied fluorescence in situ hybridization on cryosections (cryo-FISH)[2,23] to visualize Rbfox1 in mES cells and PGNs (Fig. 3e, Supplementary Table 11). In mES cells, a fluorescence-labelled probe across Rbfox1 revealed circular foci (average area of 0.44 ± 0.17 μm², mean ± s.d.) often localized at the nucleolar surface (59%) or the nuclear periphery (27%; Fig. 3f, g, Extended Data Fig. 7f). In PGNs, Rbfox1 decondensed and elongated with significantly high areas (0.59 ± 0.31 μm²; Mann–Whitney test, $P < 0.01$) and localized to the nucleoplasm interior (77%). Using specific probes for the TSS, the middle and the TES of Rbfox1 revealed increased separation between the TSS and the TES in PGNs compared to mES cells (Fig. 3h, i; 0.65 ± 0.41 μm and 0.37 ± 0.22 μm, respectively; Mann–Whitney test $P < 0.01$; Extended Data Fig. 7g).

The extensive changes in Rbfox1 localization and condensation led us to ask whether melting is generally related to changes in intrachromosomal and interchromosomal contacts. We assessed this by comparing their trans–cis contact ratios (Methods). Melted genes had significantly lower trans–cis values (higher intrachromosomal contacts) in DNs and PGNs than in mES cells (Extended Data Fig. 8a–c), but not in OLGs or in non-melting long genes (Extended Data Fig. 8a, d). Of note, Rbfox1 had a higher trans–cis ratio in PGNs, whereas Nrxn3 had a lower trans–cis ratio in DNs (Extended Data Fig. 8e, f). Decreased trans–cis ratios of melting genes in DNs or PGNs were independent of NAD association in mES cells (Extended Data Fig. 8g), whereas non-melting genes with low trans–cis values were generally associated with NADs in mES cells (Extended Data Fig. 8h).

to open chromatin in brain cells, although such events can occur (for example, Magi2 in OLGs or Dscam in DNs) (Supplementary Table 8).

We next examined in more detail melting in neurexin 3 (Nrxn3) and RNA binding Fox 1 homologue 1 (Rbfox1) genes, both of which are highly sensitive to topoisomerase I inhibition[21]. Nrxn3 encodes a membrane protein involved in synaptic connections and plasticity. In mES cells, Nrxn3 spans two TADs with high contact density, localizes in compartment B and associates with the nuclear lamina and the nucleolus. In DNs, Nrxn3 extensively melts (replicate scores of 48 and 49), is highly transcribed and accessible and belongs to compartment A (Fig. 3a, see Extended Data Fig. 7a for all cell types and replicates). Rbfox1 encodes a RNA-binding protein that regulates alternative splicing. In mES cells, Rbfox1 lies within a dense contact domain in compartment A, has very low expression and low chromatin accessibility. It also has nucleolar-associated domain and partial lamina-associated domain memberships. Rbfox1 extensively melts in PGNs (scores of 65 and 39), which coincides with its highest expression and high accessibility in these cells (Fig. 3b, Extended Data Fig. 7b).

To further understand the melting process in the Nrxn3 region, we used a polymer-physics-based approach[22] to generate ensembles of 3D

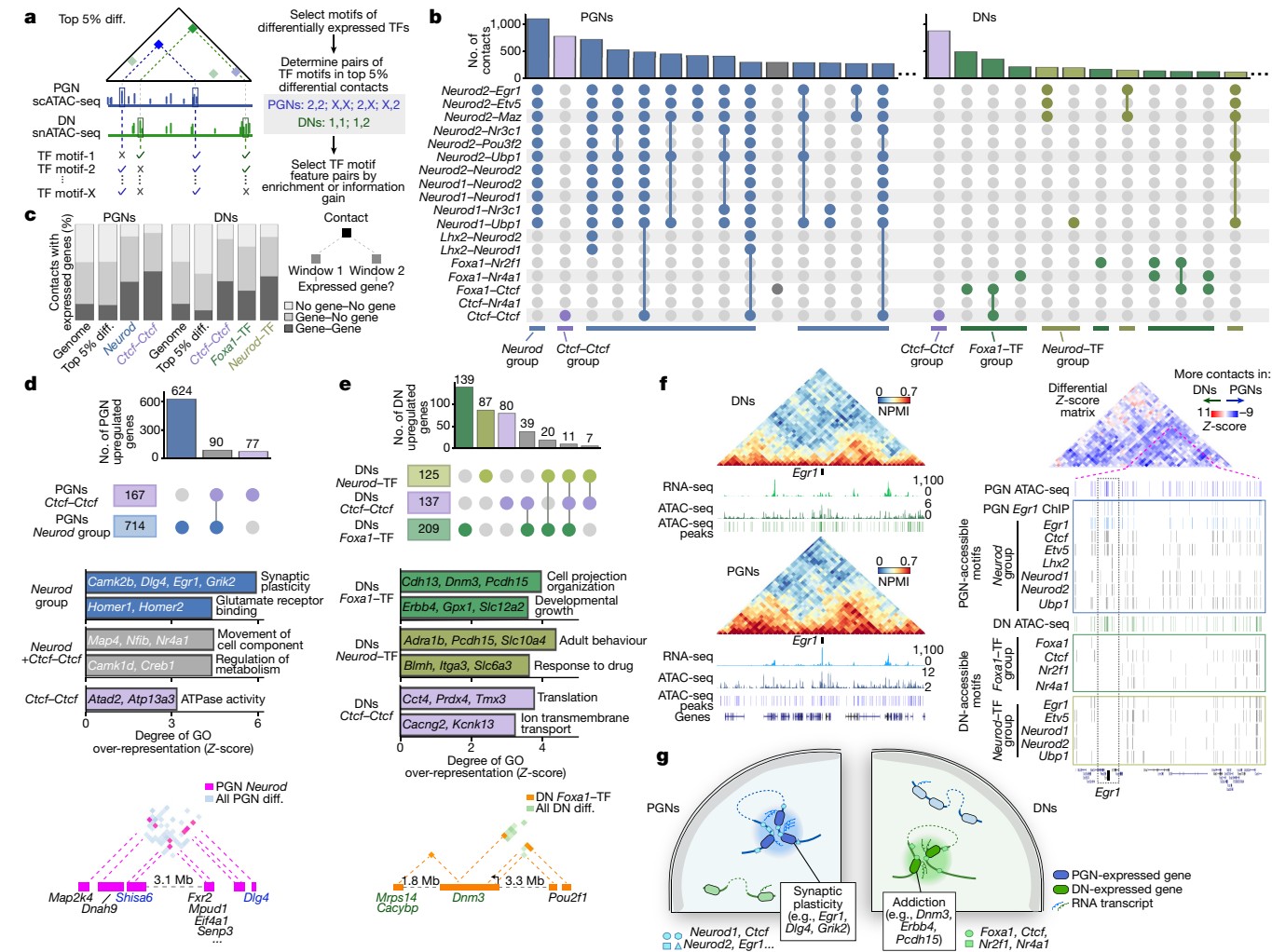

**Fig. 4 | Neuron-specific genes establish specific contacts rich in putative TF-binding sites. a**, GAM contacts from PGNs and DNs (mouse replicate 1) were normalized (*Z*-score) and subtracted to produce differential contacts matrices. The top 5% most differential contacts (top 5% diff.) ranged from 0.05 to 5 Mb. Contacts containing TF motifs within accessible chromatin on each contacting window were selected in the most (top five) enriched in PGNs or DNs or with the highest discriminatory power (information gain; Extended Data Fig. 9f). **b**, Multiple TF pairs coincide in the same PGN (left) or DN (right) differential contacts. The most abundant groups of contacts are shown for each cell type. **c**, Differential contacts with the most enriched combination TF feature pairs contain expressed genes in both windows. **d**, Differential contacts with the most abundant TF feature pairs in PGNs contain differentially expressed genes (top), with PGN-specific roles (middle; one-sided Fisher's exact permuted *P* < 0.01). The top enriched GO terms show that differential contacts between PGN upregulated genes (bottom) contain genes upregulated in PGNs (blue)

and other expressed genes. **e**, Differential contacts with the most abundant TF feature pairs in DNs contain differentially expressed genes (top) with DN-specific functions (middle; one-sided Fisher's exact permuted *P* < 0.01). The top enriched GO terms show that differential contacts between DN upregulated genes (bottom) contain genes upregulated in DNs (green) and other expressed genes. **f**, Left, *Egr1* is highly expressed (chromosome 18: 33.7–36.0 Mb) and contacts with its downstream domain in PGNs compared with DNs. Right, the differential contact matrix shows increased PGN-specific contacts in the entire region surrounding *Egr1* (right). The *Egr1*-containing TAD (inset; chromosome 18: 34.65–35.85 Mb) has multiple putative TF-binding sites found within PGN-accessible regions, most notably surrounding the *Egr1* gene (grey dashed box), not found in DNs. **g**, Schematics summarizing the presence of genes related to synaptic plasticity in PGN-specific contacts and to drug addiction in DN-specific contacts, with accessible chromatin harbouring binding sites for differentially expressed TFs.

Together, polymer modelling from GAM data and single-cell imaging highlight that domain melting is a previously unappreciated topological feature of very long genes. Domain melting occurs when genes are highly expressed, or highly accessible, in brain cell types, and the process is robustly captured by immunoGAM (Fig. 3j). The decondensation of long genes in brain cells relative to mES cells often coincides with extensive reorganization of their chromosomal contacts, preferentially alongside increased intrachromosomal contacts.

## Differential hubs of expressed genes

To explore how extensive chromatin rearrangements relate to changes in *cis*-regulatory elements and expressed genes, we extracted the top

(5%) most differential contacts between PGNs and DNs within 5 Mb (ref. [9]) (Fig. 4a, a detailed pipeline is provided in Extended Data Fig. 9a). We searched for binding motifs in accessible regions, which typically cover about 1.3 kb of the 50-kb contacting windows (Extended Data Fig. 9b), from differentially expressed transcription factors (TFs) that covered >5% of differential contacts (16 DN-specific and 32 PGN-specific TFs; Extended Data Fig. 9c, d, Supplementary Table 12). Out of 1,275 possible combinations of TF motif pairs, we prioritized 19 pairs (combinations of 14 TF motifs) that were most enriched in contacts of a given cell type or with a high ability to distinguish cell types (information gain; a full pipeline and criteria are provided in Extended Data Fig. 9e, f, and see Supplementary Table 13 for all TF pairs).

We searched for differential contacts containing the most common TF-pair combinations (Fig. 4b, a full list is shown in Extended

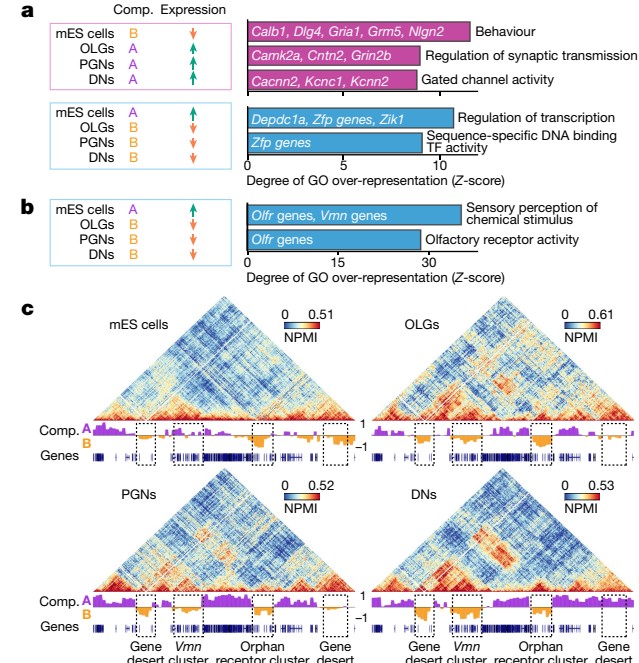

**Fig. 5 | Sensory receptor gene clusters preferentially belong to B compartments in brain cells and form megabase-range interactions.**
**a**, Selected top enriched GO terms for genes that increase expression in all brain cells relative to mES cells and move from compartment B in mES cells to compartment A in brain cells (pink box), and for genes that decrease expression in brain cells and move to compartment B compared to mES cells (blue box). All enriched GO terms had one-sided Fisher's exact permuted $P = 0$. **b**, Top enriched GO terms for genes silent in all cell types that gain membership to compartment B in brain cells. Most genes are *Olfr* and *Vmn* sensory receptor cluster genes. All enriched GO terms had one-sided Fisher's exact permuted $P = 0$. **c**, GAM contact matrices containing *Vmn* and orphan receptor genes (chromosome 7: 35–55 Mb) show large clusters of strong interactions between B compartments in OLGs, PGNs and DNs, but not mES cells. Dashed boxes indicate interacting regions.

Data Fig. 9g). In PGNs, homodimers and heterodimers for *Neurod1* and/or *Neurod2* putative binding sites characterized the most abundant contacts, together with *Egr1*, *Etv5*, *Lhx2*, *Maz*, *Nr3c1*, *Pou3f2* and *Ubp1* (*Neurod* group; 5,572 contacts). In DNs, contacts containing *Neurod1* and *Neurod2* appeared as heterodimers (660 contacts). The most frequent TF-motif pair in DNs, and the second most in PGNs, is a *Ctcf* homodimer (892 and 781 contacts, respectively). The next most abundant DN-specific contacts contained *Foxa1* combined with *Ctcf*, *Nr2f1* or *Nr4a1* (*Foxa1*–TF group; 1,612 contacts). All groups spanned 0.05–5 Mb and captured strong contacts (Extended Data Fig. 10a, b). The selected differential contacts rarely coincided with two TAD borders (Extended Data Fig. 10c) and often involved compartment A windows (Extended Data Fig. 10d). Networks of differential contacts, built on the basis of motif co-occurrence using all 50 differentially expressed TFs, confirmed connectivity between multiple TF motifs in PGNs, and between *Foxa1* or *Neurod* and specific TFs in DNs (Extended Data Fig. 10e, f, Supplementary Table 14).

Many contacts in each TF-motif group contained expressed genes in both contacting windows (30–45% in DNs, 40–50% in PGNs) that were significantly above the genome-wide or top 5% contact frequencies (10–16%; Fig. 4c, Extended Data Fig. 10g). Many of these genes were differentially expressed between PGNs and DNs (1,490 and 975, respectively, out of 3,537 differentially expressed genes; Extended Data Fig. 10h). In PGN-specific contacts, both the *Neurod* and *Ctcf*–*Ctcf* groups contained PGN upregulated genes with GO terms related to synaptic plasticity (Fig. 4d). Two PGN upregulated genes, *Dlg4* (which

is important for long-term potentiation[24]) and *Shisa6* (which prevents desensitization of AMPA receptors during plasticity[25]) were present within a hub of *Neurod* contacts that contained other activity-related genes, including *Map2k4* and *Dnah9* (see Extended Data Fig. 10i for the differential contact matrix). DN upregulated genes found with the *Foxa1*–TF (139 out of 1,844), the *Neurod*–TF (87) or the *Ctcf*–*Ctcf* (80) pair are involved in synaptic organization and addiction pathways (Fig. 4e). For example, *Dnm3* has altered protein expression in an alcohol-dependence paradigm[26] and makes contacts containing the *Foxa1*–TF pair with *Mrps14* (downregulated after nicotine exposure[27]), *Cacynp* (upregulated following alcohol exposure[28]) and *Pou2f1* (a co-factor associated with alcohol dependence[29]) (see Extended Data Fig. 10j for the differential contact matrix). Of note, *Egr1*, an immediate early gene upregulated in activated neurons[30], establishes PGN-specific contacts containing accessible regions covered by *Egr1* and *Neurod* motifs (Fig. 4f, see Extended Data Fig. 10k for replicate data). *Egr1* was highly upregulated in PGNs ($\log_2$(fold-change) = 3, PGNs compared to DNs) and gained contacts with its adjacent TAD. It also contained accessible chromatin peaks rich in TF motifs belonging to the *Neurod* group that are not seen in DNs. Binding of EGR1 protein to its own promoter is confirmed in published chromatin immunoprecipitation with sequencing (ChIP-seq) data from the cortex[31].

Together, our strategy identifies hubs of chromatin contacts specific for different neuron types that contain putative binding sites for differentially expressed TFs (Fig. 4g). These interconnected hubs bring together distal genes with specialized neuronal functions, such as synaptic plasticity in PGNs or drug addiction in DNs.

## Extensive A/B compartment reorganization

Last, we found broad changes in A/B compartmentalization between all cell types (Extended Data Fig. 11a, b), with lowest Pearson's correlations of compartment eigenvector values between brain cells and mES cells and highest correlations between neuronal replicates (Extended Data Fig. 11c). Only 12% of genomic windows changed from compartment B in mES cells to compartment A in brain cells or between compartment A in mES cells to compartment B in brain cells (7%; see Extended Data Fig. 11d, e for per-chromosome transitions). Similar mean and total genomic lengths occupied contiguously by A or B compartments characterized all cell types (Extended Data Fig. 11f). B-to-A transitions from mES cells to brain cells contained 335 genes more strongly expressed in brain cells than in mES cells (Extended Data Fig. 12a). Their enriched GO terms included 'behaviour' and 'gated ion channel activity' (Fig. 5a). A-to-B transitions in mES cells to brain cells contained mostly silent genes in all cell types (572 out of 715 genes), except 50 transcriptional regulation genes highly expressed in mES cells (Fig. 5a, Extended Data Fig. 12b).

We found that A-to-B transitions were enriched for sensory receptor genes such as *Vmn* (149 genes out of 572 silent genes in the group) and *Olfr* (179 genes), and these were often found in clusters[32,33] (Fig. 5b). Although silent, only 35% of *Vmn* and 66% of *Olfr* genes belonged to compartment B in mES cells compared with 82–96% and 72–85%, respectively, in brain cells (Extended Data Fig. 12c). *Vmn* and *Olfr* genes were often involved in strong clusters of contacts in brain cells that spanned up to 50 Mb (Fig. 5c, additional examples in Fig. 1d, Extended Data Fig. 12d, e). Long-range contacts in brain cells were significantly stronger when B compartments contained *Vmn* or, to a lesser extent, *Olfr* genes (at distances >3 Mb) (Extended Data Fig. 12f). This result suggests that sensory genes are not only more likely to belong to heterochromatic B compartments but also to more strongly contact other B compartments in brain cells.

## Discussion

Here we introduced immunoGAM to capture genome-wide chromatin conformation states of specialized cell populations in the mouse brain.

We discovered extensive reorganization of chromatin topology across genomic scales, including cell-type-specific TAD reorganization that involves genes relevant to brain cell specialization (Extended Data Fig. 12g).

We reported melting of long genes (>300 kb) with highest expression levels and/or accessible chromatin in brain cells. Single-cell imaging of *Rbfox1* in PGNs showed that the most prominent decondensation occurred between TSSs and TESs. Many long genes have specialized regulation in brain cells, for example, by topoisomerase activity[21] or DNA methylation[34], by long stretches of H3K27ac or H3K4me1 acting as enhancer-like domains[35] or by large transcription loops[36]. Their regulation is further complicated by intricate RNA processing dynamics[18], which are required for adaptive responses based on activation state. Many of the highlighted genes, including *Nrxn3*, *Rbfox1*, *Grik2* and *Dscam*, have genetic variants associated with or directly causal of neuronal diseases[37–40]. Thus, understanding how gene melting relates to regulation will become important to understanding the mechanisms of neurological disease.

Cell-type-specific networks of contacts were enriched for putative binding sites of differentially expressed TFs and connected hubs of differentially expressed genes with specialized functions[24,25,30], which is reminiscent of transcription factories[41]. DN-specific loops contained genes related to drug-exposure response and addiction paradigms. Midbrain VTA DNs are the first brain cells that respond to addictive substances, including amphetamines, nicotine and cocaine[42,43]. Future studies can explore the relationship between DN-specific chromatin landscapes and the regulation of these critical genes, with potential implications for the onset of addiction. PGN-specific contacts connected hubs of synaptic plasticity genes. Of note, PGN-specific contacts at the *Egr1* gene, which is involved in the activation of long-term potentiation, contained *Egr1* binding motifs, which suggests that there may be self-activation mechanisms. Together with reports that de novo chromatin looping can accompany transcriptional activation[5], our work suggests that coordinated TF binding at distant locations in the linear genome, but in close contact due to the 3D chromatin landscape, may be critical for the induction of long-term potentiation.

Our results also highlighted the specialization of repressive long-range contacts in brain cells. Repressed *Olfr* genes form a large interchromosomal hub in mature olfactory sensory neurons to regulate specificity of single *Olfr* gene activation[44]. We showed that sensory genes also form strong *cis*-contacts in brain cells not directly involved in sensory processes, a result confirmed in adult cortical neurons[45]. Tight 3D compartmentalization of *Vmn* and *Olfr* genes may be important for their repression in brain cells, as *Olfr* genes can be stochastically activated and mis-expressed in neurodegenerative diseases[46].

Finally, we showed that immunoGAM requires low cell numbers (approximately 1,000 cells) from single individuals while retaining the spatial organization of cells within brain tissues. This highlights its potential to provide insights into the aetiology and progression of neurological disease. Collectively, our work showed that cell specialization in the brain and chromatin structure are intimately linked at multiple genomic scales.

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

[1]Max-Delbrück Centre for Molecular Medicine, Berlin Institute for Medical Systems Biology, Epigenetic Regulation and Chromatin Architecture Group, Berlin, Germany. [2]Institute of Biology, Humboldt-Universität zu Berlin, Berlin, Germany. [3]Laboratory of Molecular Neurobiology, Department of Medical Biochemistry and Biophysics, Karolinska Institutet, Stockholm, Sweden. [4]School of Electrical Engineering and Computer Science, Ohio University, Athens, OH, USA. [5]Dipartimentio di Fisica, Università di Napoli Federico II, and INFN Napoli, Complesso Universitario di Monte Sant'Angelo, Naples, Italy. [6]UCIBIO, Department of Life Sciences, NOVA School of Science and Technology, Universidade NOVA de Lisboa, Caparica, Portugal. [7]Instituto de Ciências Biomédicas Abel Salazar, Universidade do Porto, Porto, Portugal. [8]Graduate Program in Areas of Basic and Applied Biology, Universidade do Porto, Porto, Portugal. [9]Berlin Institute of Health, Berlin, Germany. [10]Cavendish Laboratory, University of Cambridge, Cambridge, UK. [11]Wellcome Sanger Institute, Wellcome Genome Campus, Hinxton, Cambridge, UK. [12]Institute for Neuroscience, Medical University of Innsbruck, Innsbruck, Austria. [13]Institute of Clinical Sciences, Imperial College London, London, UK. [14]Max-Delbrück Centre for Molecular Medicine, Berlin Institute for Medical Systems Biology, Bioinformatics and Omics Data Science Platform, Berlin, Germany. [15]Ming Wai Lau Centre for Reparative Medicine, Stockholm node, Karolinska Institutet, Stockholm, Sweden. [16]Present address: Research Institute of Molecular Pathology (IMP), Vienna Biocenter (VBC), Vienna, Austria. [17]Present address: Broad Institute of MIT and Harvard, Cambridge, MA, USA. [18]Present address: Immunology Department, Weizmann Institute of Science, Rehovot, Israel. [19]Present address: Department of Clinical and Molecular Medicine, Norwegian University of Science and Technology, Trondheim, Norway. [20]Center for Developmental Neurobiology, Institute of Psychiatry, Psychology and Neuroscience, King's College London, London, UK. [21]MRC Center for Neurodevelopmental Disorders, King's College London, London, UK. [22]These authors contributed equally: Warren Winick-Ng, Alexander Kukalev, Izabela Harabula, Luna Zea-Redondo and Dominik Szabó. ✉e-mail: warren.winick-ng@mdc-berlin.de; ana.pombo@mdc-berlin.de

## Methods

### Randomization, blinding, and sample size

Randomization and blinding were not relevant for the current study. The experiments and the subsequent analyses were performed on wild-type animals or cell lines, for which no clinical trial, treatment or disease comparison was performed. Samples were processed in different laboratories by different people, and there was no selection criteria for the wild-type mice used in the study. The appropriate number of samples for a GAM dataset varies and depends on multiple parameters such as nuclear volume, level of chromatin compaction, quality of DNA extraction, and so on. Because most of these parameters can be assessed only after the data have been collected and processed, we recommend that the optimal resolution is defined during the collection of each GAM dataset, rather than trying to estimate optimal sample size before data collection. GAM data can be collected in multiple batches from the same starting material, therefore the sample size can be increased until the desired resolution is achieved. For scRNA-seq experiments in mES cells, no statistical method was used to predetermine sample size. Libraries were generated twice, from mES cells from different biological replicates, to account for experimental variability. For scATAC-seq experiments, no statistical method was used to predetermine sample size.

### Animal maintenance

Collection of GAM data from DNs was performed using one C57Bl/6NCrl (RRID: IMSR_CR:027; WT) mouse, which was purchased from Charles River, and from one tyrosine hydroxylase–green fluorescent protein (TH–GFP; B6.Cg-Tg(TH-GFP)21-31/C57B6) mouse, obtained as previously described[50,51]. All procedures involving WT and TH–GFP animals were approved by the Imperial College London's Animal Welfare and Ethics Review Body. Adult male mice aged 2–3 months were used. All mice had access to food and water ad libitum and were kept on a 12-h light/12-h dark cycle at 20–23 °C and 45 ± 5% humidity. WT and TH–GFP mice received an intraperitoneal injection of saline 14 days or 24 h, respectively, before tissue collection, and they were part of a larger experiment for a different study. Collection of single-nucleus ATAC-seq (snATAC-seq) data from the midbrain VTA was performed using male C57Bl/6Nl (RRID: IMSR_CR:027; WT) mice, aged 7 and 9 weeks, which were a gift from M. Gotthardt. Mice for snATAC-seq were housed in a temperature-controlled room at 22 ± 2 °C with humidity of 55 ± 10% in individually ventilated cages with 12-h light/12-h dark cycles and with access to food and water ad libitum. All experiments involving snATAC-seq animals were carried out following institutional guidelines as approved by LaGeSo Berlin and following the Directive 2010/63/EU of the European Parliament on the protection of animals used for scientific purposes. Organ preparation was done under license X9014/11.

Collection of GAM data from somatosensory oligodendrocyte cells was performed using Sox10::cre-RCE::loxP-EGFP animals[52], which were obtained by crossing Sox10::cre animals[53] on a C57BL/6j genetic background with RCE::loxP-EGFP animals[54] on a C57BL/6×CD1 mixed genetic background, both available from The Jackson Laboratory. The cre allele was maintained in hemizygosity, whereas the reporter allele was maintained in hemizygosity or homozygosity. Experimental procedures for Sox10::cre-RCE::loxP-EGFP animals were performed following the European directive 2010/63/EU, local Swedish directive L150/SJVFS/2019:9, Saknr L150 and Karolinska Institutet complementary guidelines for the procurement and use of laboratory animals, Dnr 1937/03-640. The procedures described were approved by the local committee for ethical experiments on laboratory animals in Sweden (Stockholms Norra Djurförsöksetiska nämnd), licence number 130/15. One male mouse was killed at post-natal day 21 (P21). Mice were housed to a maximum number of 5 per cage in individually ventilated cages with the following light/dark cycle: dawn 6:00–7:00, daylight 7:00–18:00, dusk 18:00–19:00, night 19:00–6:00. All mice had access to food and water ad libitum and were housed at 22 °C and 50% humidity.

Collection of GAM data from hippocampal CA1 PGNs was performed using two 19-week-old male Satb2^flox/flox mice. C57Bl/6NCrl (RRID: IMSR_CR:027; WT) mice were purchased from Charles River, Satb2^flox/flox mice that carry the loxP flanked exon 4 have been previously described[55]. The experimental procedures were done according to the Austrian Animal Experimentation Ethics Board (Bundesministerium für Wissenschaft und Verkehr, Kommission für Tierversuchsangelegenheiten). All mice had access to food and water ad libitum and were kept on a 12-h light/12-h dark cycle at 22.5 °C and 55 ± 10% humidity.

### Tissue fixation and preparation

WT, TH–GFP and Satb2^flox/flox mice were anaesthetised under isoflurane (4%), given a lethal intraperitoneal injection of pentobarbital (0.08 μl, 100 mg ml^-1 Euthatal) and transcardially perfused with 50 ml ice-cold PBS followed by 50–100 ml 4% depolymerized paraformaldehyde (PFA; electron microscopy grade, methanol-free) in 250 mM HEPES–NaOH (pH 7.4–7.6). Sox10::cre-RCE::loxP-EGFP animals were killed using an intraperitoneal injection of ketamino and xylazine followed by transcardial perfusion with 20 ml PBS and 20 ml 4% PFA in 250 mM HEPES (pH 7.4–7.6). Brains from WT or TH–GFP mice were removed, and the tissue containing the VTA was dissected from each hemisphere at room temperature and rapidly transferred to fixative. For Satb2^flox/flox mice, the CA1 field ippocampus was dissected from each hemisphere at room temperature. For Sox10^cre/RCE mice, brain tissue containing the somatosensory cortex was dissected at room temperature. Following dissection, tissue blocks were placed in 4% PFA in 250 mM HEPES–NaOH (pH 7.4–7.6) for post-fixation at 4 °C for 1 h. Brains were then placed in 8% PFA in 250 mM HEPES and incubated at 4 °C for 2–3 h. Tissue blocks were then placed in 1% PFA in 250 mM HEPES and kept at 4 °C until tissue was prepared for cryopreservation (up to 5 days, with daily solution changes).

### Cryoblock preparation and cryosectioning

Fixed tissue samples from different brain regions were further dissected to produce about 1.5 × 3 mm tissue samples suitable for Tokuyasu cryosectioning[2] (Extended Data Fig. 1a) at room temperature in 1% PFA in 250 mM HEPES. For the hippocampus, the dorsal CA1 region was further isolated. Approximately 1–3 × 1–3 mm blocks were dissected from all brain regions and were further incubated in 4% PFA in 250 mM HEPES at 4 °C for 1 h. The fixed tissue was transferred to 2.1 M sucrose in PBS and embedded for 16–24 h at 4 °C, before being positioned at the top of copper stub holders suitable for ultracryomicrotomy and frozen in liquid nitrogen. Cryopreserved tissue samples are kept indefinitely immersed under liquid nitrogen.

Frozen tissue blocks were cryosectioned with an Ultracryomicrotome (Leica Biosystems, EM UC7), with an approximate 220–230 nm thickness[2]. Cryosections were captured in drops of 2.1 M sucrose in PBS solution suspended in a copper wire loop and transferred to 10-mm glass coverslips for confocal imaging or onto a 4.0-μm polyethylene naphthalate (PEN; Leica Microsystems, 11600289) membrane on metal framed slides for laser microdissection.

### Immunofluorescence detection of GAM samples for confocal microscopy

For confocal imaging, cryosections were incubated in sheep anti-TH (1:500; Pel Freez Arkansas, P60101-0), mouse anti-pan-histone H11-4 (1:500; Merck, MAB3422) or chicken anti-GFP (1:500; Abcam, ab13970) followed by donkey anti-sheep or goat anti-chicken IgG conjugated with Alexa Fluor-488 (for TH and GFP; Abcam) or donkey anti-mouse IgG conjugated with Alexa Fluor-555 or Alexa Fluor-488 (for pan-histone; Invitrogen).

For PGNs, cryosections were washed (3 times, 30 min in total) in PBS, permeabilized (5 min) in 0.3% Triton X-100 in PBS (v/v) and incubated (2 h, room temperature) in blocking solution (1% BSA (w/v), 5% fetal

bovine serum (FBS (w/v), Gibco, 10270), 0.05% Triton X-100 (v/v) in PBS). After incubation (overnight, 4 °C) with primary antibody in blocking solution, the cryosections were washed (3–5 times, 30 min) in 0.025% Triton X-100 in PBS (v/v) and immunolabelled (1 h, room temperature) with secondary antibodies in blocking solution followed by 3 washes (15 min) in PBS. Cryosections were then counterstained (5 min) with 0.5 µg ml$^{-1}$ 4′,6′-diamino-2-phenylindole (DAPI; Sigma-Aldrich, D9542) in PBS, and then rinsed in PBS and water. Coverslips were mounted in Mowiol 4-88 solution in 5% glycerol, 0.1 M Tris-HCl (pH 8.5).

The number of SATB2-positive cells present in the hippocampal CA1 area of the *Satb2$^{flox/flox}$* control mice was determined by counting nuclei positive for SATB2 immunostaining (1:100; Abcam, ab10563678). To avoid counting the same nuclei, only every 30th ultrathin section cut through the tissue was collected, and the remaining sections discarded. Twenty-five nuclei were identified in the pyramidal neuron layer per image in the DAPI channel, and only SATB2-positive cells were counted. We confirmed that most cells (96%) within the CA1 layer were PGNs (data not shown).

For DNs and OLGs, cryosections were washed (3 times, 30 min in total) in PBS, quenched (20 min) in PBS containing 20 mM glycine, then permeabilized (15 min) in 0.1% Triton X-100 in PBS (v/v). Cryosections were then incubated (1 h, room temperature) in blocking solution (1% BSA (w/v), 0.2% fish-skin gelatin (w/v), 0.05% casein (w/v) and 0.05% Tween-20 (v/v) in PBS). After incubation (overnight, 4 °C) with the antibody in blocking solution, the cryosections were washed (3–5 times, 1 h) in blocking solution and immunolabelled (1 h, room temperature) with secondary antibodies in blocking solution, followed by 3 washes (15 min) in 0.5% Tween-20 in PBS (v/v). Cryosections were then counterstained with 0.5 µg ml$^{-1}$ DAPI in PBS, then rinsed in PBS. Coverslips were mounted in Mowiol 4-88.

Digital images were acquired with a Leica TCS SP8-STED confocal microscope (Leica Microsystems) using a ×63 oil-immersion objective (numerical aperture of 1.4) or a ×2 oil-immersion objective, using a pinhole equivalent to 1 Airy disk. Images were acquired using 405-nm excitation and 420–480-nm emission for DAPI, 488-nm excitation and 505–530-nm emission for TH or GFP, and 555-nm excitation and 560-nm emission using a long-pass filter at 1,024 × 1,024 pixel resolution. Images were processed using Fiji (v.2.0.0-rc-69/1.52p), and adjustments included the optimization of the dynamic signal range with contrast stretching.

**Immunofluorescence detection of GAM samples for laser microdissection**

For laser microdissection, cryosections on PEN membranes were washed, permeabilized and blocked as for confocal microscopy, and incubated with primary and secondary antibodies as indicated above except for the use of higher concentrations of primary antibodies, as follows: anti-TH (1:50), anti-pan-histone (1:50) or anti-GFP (1:50). Secondary antibodies were used at the same concentration. Cell staining was visualized using a Leica laser microdissection microscope (Leica Microsystems, LMD7000) using a ×3 dry objective. Following detection of cellular sections of the cell types of choice containing nuclear slices (nuclear profiles (NPs)), individual NPs were laser microdissected from the PEN membrane and collected into PCR adhesive caps (AdhesiveStrip 8C opaque, Carl Zeiss, 415190-9161-000). We used multiplex-GAM[9], for which three NPs were collected into each adhesive cap and the presence of NPs in each lid was confirmed with a ×5 objective using a 420–480-nm emission filter. Control lids not containing NPs (water controls) were included for each dataset collection to keep track of contamination and noise amplification of whole-genome amplification (WGA) and library reactions, and can be found in Supplementary Table 2.

**WGA of NPs**

WGA was performed using an in-house protocol. In brief, NPs were lysed directly in the PCR adhesive caps for 4 h (or 24 h for 160 out of 585

GAM samples from DN replicate 1) at 60 °C in 1.2× lysis buffer (30 mM Tris-HCl pH 8.0, 2 mM EDTA pH 8.0, 800 mM guanidinium-HCl, 5% (v/v) Tween 20, 0.5% (v/v) Triton X-100) containing 2.116 units ml$^{-1}$ Qiagen protease (Qiagen, 19155). After protease inactivation at 75 °C for 30 min, the extracted DNA was amplified using random hexamer primers with an adaptor sequence. The pre-amplification step was done using 2× DeepVent mix (2× Thermo polymerase buffer (10×), 400 µm dNTPs, 4 mM MgSO$_4$ in ultrapure water), 0.5 µM GAT-7N primers (5′-GTG AGT GAT GGT TGA GGT AGT GTG GAG NNN NNN N) and 2 units µl$^{-1}$ DeepVent (exo-) DNA polymerase (New England Biolabs, M0259L) in the programmable thermal cycler for 11 cycles. Primers that annealed to the general adaptor sequence were then used in a second exponential amplification reaction to increase the amount of product. The exponential amplification was done using 2× DeepVent mix, 10 mM dNTPs, 100 µM GAM-COM primers (5′-GTGAGTGATGGTTGAGGTAGTGTGGAG) and 2 units µl$^{-1}$ DeepVent (exo-) DNA polymerase in the programmable thermal cycler for 26 cycles. For a small number of NPs from DNs (Supplementary Table 2), WGA was performed using a WGA4 kit (Sigma-Aldrich) using the manufacturer's instructions; the recent formulation of this kit is no longer suitable for GAM data production from subcellular nuclear slices.

**GAM library preparation and high-throughput sequencing**

Following WGA, the samples were purified using SPRI beads (0.725 or 1.7 ratio of beads per sample volume). The DNA concentration of each purified sample was measured using a Quant-iT Pico Green dsDNA assay kit (Invitrogen, P7589) according to the manufacturer's instructions. GAM libraries were prepared using an Illumina Nextera XT library preparation kit (Illumina, FC-131-1096) following the manufacturer's instructions with an 80% reduced volume of reagents. Following library preparation, the DNA was purified using SPRI beads (1.7 ratio of beads per sample volume) and the concentration for each sample was measured using a Quant-iT PicoGreen dsDNA assay. An equal amount of DNA from each sample was pooled together (up to 196 samples), and the final pool was additionally purified three times using the SPRI beads (1.7 ratio of beads per sample volume). The final pool of libraries was analysed using DNA High Sensitivity on-chip electrophoresis on an Agilent 2100 Bioanalyzer to confirm the removal of primer dimers and to estimate the average size and DNA fragment size distribution in the pool. NGS libraries were sequenced on an Illumina NextSeq 500 machine according to the manufacturer's instructions using single-end 75 bp reads. The number of sequenced reads for each sample can be found in Supplementary Table 2.

Tn5-based libraries are preferred for GAM data sequencing to increase fragment sequence variation, which helps avoid the need for dark cycles in the current Illumina machines. This choice greatly reduces the cost of sequencing and decreases the frequency of noise reads from absent windows seen with the previous protocol[3].

**GAM data sequence alignment**

Sequenced reads from each GAM library were mapped to the mouse genome assembly GRCm38 (December 2011, mm10) with Bowtie2 (v.2.3.4.3) using default settings[56]. All non-uniquely mapped reads, reads with mapping quality <20 and PCR duplicates were excluded from further analyses.

**GAM data window calling and sample QC**

Positive genomic windows present within ultrathin nuclear slices were identified for each GAM library. In brief, the genome was split into equal-sized windows (50 kb), and the number of nucleotides sequenced in each bin was calculated for each GAM sample with bedtools[57]. Next, we determined the percentage of orphan windows (that is, positive windows that were flanked by two adjacent negative windows) for every percentile of the nucleotide coverage distribution and we identified the percentile with the lowest percentage of orphan windows for each GAM sample in the dataset. The number of nucleotides that corresponds to the percentile

with the lowest percentage of orphan windows in each sample was used as an optimal coverage threshold for window identification in each sample. Windows were called positive if the number of nucleotides sequenced in each bin was greater than the determined optimal threshold.

Each dataset was assessed for QC by determining the percentage of orphan windows in each sample, the number of uniquely mapped reads to the mouse genome and the correlations from cross-well contamination for every sample (Supplementary Table 2). Most GAM libraries passed the QC analyses (86–96% in each dataset; Extended Data Fig. 1b, c). To assess the quality of sampling in each GAM dataset, we measured the frequency with which all possible intrachromosomal pairs of genomic windows are found in the same GAM sample; we found that 98.8–99.9% of all mappable pairs of windows were sampled at least once at resolution 50 kb at all genomic distances. Each sample was considered to be of good quality if they had <70% orphan windows, >50,000 uniquely mapped reads and a cross-well contamination score determined per collection plate of <0.4 (Jaccard index). The number of samples in each cell type that passed QC is summarized in Extended Data Fig. 2a. Following QC analysis, we noted that the 160 (out of 585) DN replicate 1 samples incubated with lysis buffer for 24 h had decreases in orphan windows (median = 26% and 36% for 24 h and 4 h, respectively) and increases in total genome coverage (median = 9% and 6% for 24 h and 4 h, respectively). Although these differences were minor, we recommend 24 h lysis for future work.

## Publicly available GAM datasets from mES cells

For mES cells, GAM datasets were downloaded from the 4D Nucleome portal (https://data.4dnucleome.org/). We used 249 × 3 NP GAM datasets from mES cells (clone 46C), which were grown at 37 °C in a 5% $CO_2$ incubator in Glasgow modified Eagle's medium (MEM), supplemented with 10% FBS, 2 ng ml$^{-1}$ leukaemia inhibitory factor (LIF) and 1 mM 2-mercaptoethanol, on 0.1% gelatin-coated dishes. Cells were passaged every other day. After the last passage, 24 h before collection, mES cells were re-plated in serum-free ESGRO Complete Clonal Grade medium (Merck, SF001-B). The list of 4DN sample identity numbers is provided in Supplementary Table 1.

## Visualization of pairwise chromatin contact matrices

To visualize GAM data, contact matrices were calculated using pointwise mutual information (PMI) for all pairs of windows genome-wide. PMI describes the difference between the probability of a pair of genomic windows being found in the same NP given both their joint distribution and their individual distributions across all NPs. PMI was calculated using the following formula, where $p(x)$ and $p(y)$ are the individual distributions of genomic windows $x$ and $y$, respectively, and $p(x,y)$ are their joint distribution:

$$PMI = \log(p(x,y)/p(x)p(y)) \tag{1}$$

PMI can be bounded between −1 and 1 to produce a normalized PMI (NPMI) value given by the following formula:

$$NPMI = PMI/(-\log(p(x,y))) \tag{2}$$

For visualization of the contact matrices, scale bars are adjusted in each genomic region displayed to a range between 0 and the 99th percentile of NPMI values for each cell type.

## Insulation score and topological domain boundary calling

TAD calling was performed by calculating insulation scores in NPMI GAM contact matrices at 50-kb resolution, as previously described[2,9]. The insulation square method was chosen as it was previously shown that the domain borders detected in GAM data are also found in Hi-C, for which they are the most robust (most insulated)[2,9]. The insulation score was computed individually for each cell type and biological replicate, with insulation square sizes ranging from 100 to 1,000 kb. TAD boundaries were called using a 500-kb insulation square size and based on local minima of the insulation score. This approach does not detect meta-TADs or sub-TADs, and results in numbers and lengths of domains were similar to previous reports[6,58]. Future work with higher resolution GAM datasets will enable further analyses of the reorganization of domains at finer genomic scales to investigate changes in sub-TADs, which have been previously shown to occur following cell commitment to neuronal lineages[59].

Within each dataset, boundaries that were touching or overlapping by at least one nucleotide were merged. Boundaries were further refined to consider only the minimum insulation score within the boundary and one window on each side, to produce a 3-bin 'minimum insulation score' boundary. In comparisons of boundaries between different datasets, 150-kb boundaries were considered different when separated by at least one 50-kb genomic bin, that is, if the centre of the boundaries are separated by at least 200 kb (note chromosome Y was excluded from the analysis). In Fig. 2b, we considered the boundary coordinate as the genomic window within a boundary with the lowest insulation value. TAD border coordinates for all cell types can be found in Supplementary Table 3, and the full range of insulation scores (100–1,000 kb) for all cell types can be found in Supplementary Table 4. UpSet plots for TAD border overlaps, compartments and TF motif analyses were generated using either custom Python or R scripts or using the UpSetR package (v.1.4.0)[60].

## Identification of compartments A and B

For compartment analysis, matrices of co-segregation frequency were determined using the ratio of independent occurrence of a single positive window in each sample over the pairwise co-occurrence of pairs of positive windows in a given pair of genomic windows[2]. GAM co-segregation matrices at 250-kb resolution were assigned to either A or B compartments, as previously described[2]. In brief, each chromosome was represented as a matrix of observed interactions $O(i,j)$ between locus $i$ and locus $j$ (co-segregation) and separately for $E(i,j)$, whereby each pair of genomic window is the mean number of contacts with the same distance between $i$ and $j$. A matrix of observed over expected values $O/E(i,j)$ was produced by dividing $O$ by $E$. A correlation matrix $C(i,j)$ was produced between column $i$ and column $j$ of the $O/E$ matrix. PCA was performed for the first three components on matrix $C$ before extracting the component with the best correlation to GC content. Loci with PCA eigenvector values with the same sign that correlate best with GC content were called A compartments, whereas regions with the opposite sign were B compartments. For visualizations and Pearson's correlations between datasets, eigenvector values on the same chromosome in compartment A were normalized from 0 to 1, whereas values on the same chromosome in compartment B were normalized from −1 to 0. Compartments were considered common if they had the same compartment definition within the same genomic bin. Compartment changes between cell types were computed after considering compartments that were common between biological replicates unless otherwise indicated.

To identify and visualize gene expression differences among genes in changing compartments, $k$-means clustering was performed on triplicate pseudo-replicates of each cell type using a custom Python script (Extended Data Fig. 12a, b). The number of clusters were determined using the elbow method, with $k$-means = 6 for genes in compartment B in mES cells and compartment A in brain cells, and $k$-means = 5 for compartment A in mES cells and compartment B in brain cells.

## mES cell culture for scRNA-seq and scATAC-seq

mES cells from the 46C clone, derived from E14tg2a and expressing GFP under the *Sox1* promoter[61], were a gift from D. Henrique (Instituto de Medicina Molecular, Faculdade Medicina Lisboa, Lisbon, Portugal). mES cells were cultured as previously described[62]. In brief, cells were

routinely grown at 37 °C, 5% (v/v) $CO_2$, on gelatine-coated (0.1% v/v) Nunc T25 flasks in Gibco Glasgow's MEM (Invitrogen, 21710082), supplemented with 10% (v/v) fetal calf serum (BioScience LifeSciences, 7.01, batch number 110006) for scRNA-seq or Gibco FBS (Invitrogen, 10270-106, batch number 41F8126K) for ATAC-seq, 2,000 units ml$^{-1}$ LIF (Millipore, ESG1107), 0.1 mM β-mercaptoethanol (Invitrogen, 31350-010), 2 mM L-glutamine (Invitrogen, 25030-024), 1 mM sodium pyruvate (Invitrogen, 11360070), 1% penicillin–streptomycin (Invitrogen, 15140122) and 1% MEM non-essential amino acids (Invitrogen, 11140035). Medium was changed every day and cells were split every other day. mES cell batches tested negative for *Mycoplasma* infection, which was performed according to the manufacturer's instructions (AppliChem, A3744,0020). Before collecting material for scRNA-seq or ATAC-seq, cells were grown for 48 h in serum-free ESGRO Complete Clonal Grade medium (Merck, SF001-B), supplemented with 1,000 units ml$^{-1}$ LIF, on gelatine-coated (Sigma, G1393-100 ml, 0.1% v/v) Nunc 10-cm dishes, with a change in medium after 24 h.

46C E14tg2 mES cells are not listed in the ICLAC Register of Misidentified Cell Lines. The 46C E14tg2 mES cell line was generated by insertion of an eGFP cassette under the control of the *Sox1* promoter in E14tg2 cells. Reads aligned with the GFP sequence were identified in the GAM sequencing data from mES cells. In addition, genome sequencing data from GAM mES cell samples was mined for single nucleotide polymorphisms (SNPs). Although GAM sequencing reads are sparsely distributed across the genome, there was a 64% overlap of GAM mES cell SNPs with SNPs identified from the parental E14tg2 genome sequencing data (https://www.ncbi.nlm.nih.gov/sra?term=SRX389523; data not shown).

### Single-cell mRNA library preparation
Two batches (denoted batch A and B) of single-cell mRNA-seq libraries were prepared according to the Fluidigm manual "Using the C1 Single-Cell Auto Prep System to Generate mRNA from Single Cells and Libraries for Sequencing". Cell suspension was loaded on 10–17 μm C1 Single-Cell Auto Prep IFCs (Fluidigm, 100-5760, kit 100-6201). After loading, the chip was observed under the microscope to score cells as singlets, doublets, multiplets, debris or other. The chip was then loaded again on Fluidigm C1 IFCs, and cDNA was synthesized and pre-amplified in the chip using a Clontech SMARTer kit (Takara Clontech, 634833). In batch B, we included Spike-In Mix 1 (1:1,000; Life Technologies, 4456740) as per the Fluidigm manual. Illumina sequencing libraries were prepared using a Nextera XT kit (Illumina, FC-131-1096) and a Nextera Index kit (Illumina, FC-131-1002), as previously described[63]. Libraries from each microfluidic chip (96 cells) were pooled and sequenced on 4 lanes on Illumina HiSeq 2000, 2×100-bp paired-end (batch A) or 1 lane on Illumina HiSeq 2000, 2×125-bp paired-end (batch B) at the Wellcome Trust Sanger Institute Sequencing Facility (Supplementary Table 15).

### scRNA-seq data processing, mapping and expression estimates
To calculate expression estimates, mRNA-seq reads were mapped with STAR (spliced transcripts alignment to a reference, v.2.4.2a)[64] and processed with RSEM using the 'single-cell-prior' option (RNA-seq by expectation-maximization, v.1.2.25)[65]. The references provided to STAR and RSEM were the GTF annotation from UCSC Known Genes (mm10, v.6) and the associated isoform–gene relationship information from the Known Isoforms table (UCSC), adding information for ERCC sequences in samples from batch B. Tables were downloaded from the UCSC Table browser (http://genome.ucsc.edu/cgi-bin/hgTables) and for ERCCs, from the ThermoFisher website (http://www.thermofisher.com/order/catalog/product/4456739). Gene-level expression estimates in 'Expected Counts' from RSEM were used for the analysis.

### scRNA-seq data processing QC
Cells scored as doublets, multiplets or debris during visual inspection of the C1 chip were excluded from the analysis. Datasets were also excluded if any of the following conditions were met: <500,000 reads

(calculated using sam-stats from ea-utils.1.1.2-537)[66]; <60% of reads mapped (calculated with sam-stats); <50% reads mapped to mRNA (picard-tools-2.5.0, http://broadinstitute.github.io/picard/); >15% of reads mapped to chrM (sam-stats); if present, >20% of reads mapped to ERCCs (sam-stats). Following processing, 98 single cells passed quality thresholds in the final dataset. Correlations between previously published mES cells (clone 46C) mRNA-seq bulk[62] and the scRNA-seq mES cell transcriptomes were performed to assess the quality of the single-cell data. Correlations were performed as previously described[67]. Average single-cell expression was highly correlated with bulk RNA-seq data (Extended Data Fig. 4c).

### scRNA-seq analysis
To utilize published single-cell transcriptomes from brain cell types of interest, we selected P21–22 OLGs[68], P22–32 CA1 PGNs[69] and P21–26 VTA DNs[70] on the basis of the cell type and subtype definitions provided in the respective publications. The matrices of counts provided in each publication, along with the single-cell mES cell transcriptomes produced that passed QC, were combined with no prior batch correction due to the lack of equivalent cell types across all single-cell datasets. The combined matrix of counts was normalized by applying the Log-Normalize method and scaled using Seurat (v.3.1.4)[71]. The scaled data were used for a PCA, followed by processing through dimensionality reduction using uniform manifold approximation and projection (UMAP)[72] for visualization purposes using the Seurat R package[71], with default parameters. Visualization of known cell-type-specific marker genes confirmed that the different transcriptomes are grouped into cell-type-specific clusters (Extended Data Fig. 4e). Single mES cell transcriptomes from batch A and B clustered together, and were pooled for further analyses. Genes that could not be mapped to the chosen reference GTF were removed (UCSC; accessed from iGenomes July 17, 2015; https://support.illumina.com/sequencing/sequencing_software/igenome.html).

To generate bigwig tracks for visualization, raw fastq files from each single cell within the same cell type were pooled into one fastq file. Reads were mapped to the mouse genome (mm10) using STAR with default parameters but –outFilterMultimapNmax 10. BAM files were sorted and indexed using Samtools (v.1.3.1)[73] and normalized (reads per kilobase of transcript per million (RPKM)) bigwigs were generated using Deeptools (v.3.1.3)[74] bamCoverage. To account for differences in the number of technical replicates in OLG samples, cells were divided into groups by the number of runs (1, 2 and 6). The median of the reads for the group with the lowest sequencing depth was used as a threshold to normalize the other groups (that is, the rest of the fastq files were randomly downsampled to that number of reads). The three groups of raw reads were pooled together and processed by applying the same method as for the other cell types. Pseudobulk expression was determined using the regularized log (R-log) value for each gene (Extended Data Fig. 4f, g). In each cell type, only the genes with R-log values of ≥2.5 in all pseudobulk replicates were considered expressed.

### Differential gene expression analysis
For differential expression analysis for all cell types, pseudobulk replicate samples were obtained by randomly partitioning the total number of single cells per dataset into three groups and pooling all unique molecular identifiers (UMIs) per gene of cells belonging to the same replicate. To determine differentially expressed genes, all six possible pairwise comparisons between samples were performed using DEseq2 (v.1.24.0) with default parameters[75]. In addition, shrunken log$_2$ fold-changes were added with the lfcShrink function, using default parameters. Genes classified as differentially expressed in at least one comparison were considered for further analysis (adjusted *P* value < 0.05; Benjamini–Hochberg multiple testing correction method). A summary table for the differential expression analysis of all cell types can be found in Supplementary Table 12. For the TF

motif analysis, only the differentially expressed genes obtained from the comparison between DNs and PGNs were considered for further analysis (Extended Data Fig. 9c, d).

## Tn5 purification

The pTXB1 plasmid carrying the Tn5-intein-CBD fusion construct with the hyperactive Tn5 protein containing the E54K and L372P mutations was obtained from Addgene (plasmid 60240). Tn5 expression and purification was performed as previously described[76], except that the final storage buffer was 50 mM HEPES-KOH pH 7.2, 0.8 M NaCl, 0.1 mM EDTA, 1 mM dithiothreitol and 55% glycerol.

## Tn5 adapter mix preparation

To generate 100 µM adapter mix, 200 µM Tn5MErev (5′-[phos] CTGTCTCTTATACACATC) was mixed with of 200 µM Tn5ME-A (5′-TCGTCGGCAGCGTCAGATGTGTATAAGAGACAG; Adapter_mixA, 1:1 ratio). Separately, 200 µM Tn5MErev was mixed with 1 volume of 200 µM Tn5ME-B (5′-GTCTCGTGGGCTCGGAGATGTGTATAAGAGACAG; Adapter_mixB, 1:1 ratio). The two mixtures were incubated for 5 min at 95 °C and gradually cooled to 25 °C at a ramp rate of 0.1 °C s$^{-1}$. Finally, the Adapter_mixA was mixed with Adapter_mixB at a 1:1 ratio for a final 100 µM adapter mix.

## mES cell ATAC-seq library preparation

ATAC-seq libraries were generated from approximately 75,000 mES cell nuclei following the Omni ATAC protocol[77] with a modified transposition reaction: TAPS-DMF buffer (50 mM TAPS-NaOH, pH 8.5, 25 mM MgCl$_2$, 50% DMF), 0.1% Tween-20, 0.1% digitonin, in 0.25x PBS. A total of 3 µl of the Tn5 mix (5.6 µg Tn5 and 0.143 volume of 100 µM adapter mix) was added to the transposition reaction mix. Libraries were prepared as described in the Omni ATAC protocol. The final library was sequenced with an Illumina NextSeq 500 machine according to manufacturer's instructions, using paired-end 75 bp reads (150 cycles).

## Isolation of the VTA for snATAC-seq

Male C57Bl/6Nl (RRID: IMSR_CR:027; WT) mice, aged 7 and 9 weeks, were killed by cervical dislocation. Brains were removed and the tissue containing the midbrain VTA was dissected from each hemisphere at room temperature and rapidly frozen on dry ice. Frozen midbrain samples were kept at −80 °C until further processing.

## DN snATAC-seq library preparation

Two 10X Genomics scATAC-seq libraries from the midbrain VTA, VTA-1 and VTA-2 (from mice aged 7 or 9 weeks, respectively), were generated from midbrain VTA samples according to the 10X Genomics manual "Nuclei Isolation from Mouse Brain Tissue for Single Cell ATAC Sequencing Rev B" for flash-frozen tissue with minor adjustments. In brief, 500 µl 0.1× lysis buffer (10 mM Tris-HCl, pH 7.4, 10 mM NaCl, 3 mM MgCl$_2$, 1% BSA, 0.01% Tween-20, 0.01% Nonidet P40 substitute, 0.001% digitonin, and 1× complete Mini, EDTA-free protease inhibitor cocktail, Millipore-Sigma, 11836170001) was added to the frozen samples and immediately homogenized using a pellet pestle (15 times), followed by 5 min incubation on ice. The lysate was pipette mixed 10 times, then incubated 10 min on ice. Finally, 500 µl of chilled wash buffer (10 mM Tris-HCl, pH 7.4, 10 mM NaCl, 2 mM MgCl$_2$, 1% BSA, 0.1% Tween-20) was added to the lysed cells, and the suspension was passed through a 30-µm CellTrics strainers (Th Geyer, 7648779). The final approximately 500 µl nuclei suspension was stained with DAPI (final concentration 0.03 µg ml$^{-1}$) for about 5 min.

Around 200,000 DAPI-positive events were sorted using a BD FAC-SAria III flow cytometer with 70-µm nozzle configuration with sample and sort collection device cooling set to 4 °C into 300 µl Diluted Nuclei buffer (commercial buffer from 10X Genomics) in a 1.5-ml Eppendorf tube. A first gate excluded debris in a forward scatter/side scatter plot (see examples in Extended Data Fig. 4h, i). A consecutive, second gate

in a DAPI-A/DAPI-H plot was used to exclude doublets and nuclei with incomplete DNA content (BD FACSDiva software, v.8.0.2). The collected nuclei were centrifuged at 500g for 5 min at 4 °C and resuspended in 20 µl Diluted Nuclei buffer. The nucleus concentration was determined using a Countess II FL Automated Cell Counter in DAPI fluorescence mode. snATAC-seq libraries were prepared per the Chromium Next GEM Single Cell ATAC Reagent kits v.1.1 User Guide. In brief, nuclei were loaded on a microfluidics chip together with transposition reagents, transposase enzyme, beads with oligo-dT tags and oil to create an emulsion. Afterwards, the transposase reaction takes place inside the droplets. The barcoded cDNA is recovered from the emulsion, amplified and cleaned using a bead purification process. The cDNA is then using for library construction, including enzymatic fragmentation, adapter ligation and sample index PCR. Libraries were sequenced with either an Illumina NextSeq 500 machine using paired-end 75 bp reads (for VTA-1, 150 cycles) or a NovaSeq 6000 using paired-end 75 bp reads (for VTA-2, 100 cycles).

## ATAC-seq data processing, mapping, processing and QC

For bulk mES cell ATAC-seq, paired-end reads were mapped to the mouse genome (mm10) using Bowtie with the following parameters: –minins 25 –maxins 2000 –no-discordant –dovetail – soft-clipped-unmapped-tlen. Low-quality mapped reads (MQ < 30) and mitochondrial reads were removed. Duplicated reads were removed with Sambamba[78] (v.0.6.8). Reads passing quality checks were converted to BAM format for further analyses.

For VTA snATAC-seq, paired-end reads were demultiplexed and mapped to the mouse genome (mm10) using the 10X Genomics Cellranger software (version cellranger-atac-1.2.0). The two VTA snATAC-seq libraries were analysed using ArchR software (v.0.9.1)[79]. Doublets were removed following default parameters in ArchR. Next, low-quality cells (identified as TSS enrichment score <4 and <2,500 unique fragments per cell) were removed for further analyses.

Next, dimensionality reduction was performed using the Latent semantic indexing (LSI) dimensionality reduction method from ArchR, with default parameters (except iterations = 10, resolution = 0.2, var-Features = 60,000). The ArchR addHarmony function was used to run the Harmony algorithm for batch correction with default parameters, followed by clusters calling. Gene scores were determined as specified by ArchR[79]. DNs were identified as the cluster with higher gene scores for *Th*, a well-known DN marker, and confirmed by additional DN marker expression (for example, *Lmx1b*, *Foxa2*, *Foxa1* and *Slc6a3*). The DN cluster is composed of 216 cells in total (113 from VTA-1 and 103 from the VTA-2). UMI duplicates were collapsed to one fragment. To visualize an approximation for gene expression, gene scores were calculated using the createArrowFiles (addGeneScoreMat = TRUE) function in ArchR.

## Processing of published OLG and PGN scATAC-seq

scATAC-seq BAM files for OLGs were downloaded from the sciATAC-seq in vivo atlas of the mouse brain[80]. Next, reads were extracted from the BAM file that corresponded to cells from the cluster identified as oligodendrocytes from the prefrontal cortex (458 cells), to produce a pseudobulk ATAC BAM file. The original data, mapped to the mm9 genome, were converted to mm10 using the liftOver tool from UCSC utilities (https://genome.ucsc.edu/cgi-bin/hgLiftOver).

scATAC-seq datasets were obtained from hippocampal PGNs[81]. A BAM file containing all cell types was supplied by A. Adey (Molecular and Medical Genetics, Oregon Health & Science University, Portland, OR, USA). Reads were extracted from the BAM file that corresponded to the NR1 PGN population (270 cells) to produce a pseudobulk ATAC BAM file.

## Generation of normalized ATAC-seq bigwig tracks

A size factor normalization was applied to generate ATAC-seq bigwig tracks comparable between mES cells, OLGs, PGNs and DNs. First, a count matrix was generated for all TSS regions (±250 bp), which contained reads from at least two of the four cell types.

The TSS list was extracted from the genes.gtf file included in the cell ranger reference data (refdata-cellranger-atac-mm10-1.2.0l; https://support.10xgenomics.com/single-cell-atac/software/pipelines/latest/advanced/references). To calculate size factors, the TSS count matrix was processed through DESeqDataSetFromMatrix and estimateSizeFactors from the DESeq2 package[75]. For all cell types, the scale factor (SF) = (cell type size factor) × −1.

Each pseudobulk ATAC-seq BAM file from mES cells, PGNs and OLGs was converted to the bedGraph format using the genomeCoverageBed function from bedtools[57] with the following parameters: -pc -bg -scale SF. For DNs, ATAC-seq fragment files were converted to the bedGraph format using the genomeCoverageBed function from bedtools[57] with the following parameters: -g chrom.sizes -bg -scale SF. The mm10 chrom.sizes file was downloaded from UCSC using fetchChromSize from UCSC utilities (http://hgdownload.soe.ucsc.edu/admin/exe/). The bedGraph files were then converted to bigwig using the bedGraphToBigWig function from UCSC utilities.

### DN and PGN ATAC-seq peak calling
ATAC-seq peaks were called in DNs following the iterative overlap peak merging procedure described in the ArchR package[79]. First, two pseudobulk replicates were generated by running the addGroupCoverages function and then reproducible peaks were called using the addReproduciblePeakSet function. For PGNs, peaks for the NR1 cluster were obtained from Sinnamon et al.[81]. For further analyses, peaks were considered positive if they were found in at least 10% of single nuclei (>10 nuclei in DNs; >13 cells in PGNs).

### RNA and ATAC-seq length-scaled ATAC reads per million
To calculate length-scaled RNA reads per million (lsRRPM) for 479 long genes (>300 kb), the mES cell BAM file (paired-end) was read using the readGAlignmentPairs function from the GenomicAlignments function from the GenomicAlignments package in R (v.1.20.1; https://bioconductor.org/packages/release/bioc/html/GenomicAlignments.html). For published single-cell datasets (OLGs, PGNs, DNs; single-end libraries), BAM files were loaded using the readGAlignments function from the GenomicAlignment package. Owing to the very long length of some reads, all BAM fragments were resized to the 5′ end base pair to avoid overlapping with multiple features. Next, the following formula was used to compute lsRRPM values for each cell type and per gene:

lsRRPM = number of overlaps between RNA fragments and long

gene body gene length $(10^{-6})$

× total number of RNA fragments $(10^{-6})$

To calculate length-scaled ATAC reads per million (lsARPM) for 479 long genes (>300 kb), concordant paired-end fragments were extracted for all cell types using the readGAlignmentPairs function from the GenomicAlignments function in R with the following total number of fragments: 37,261,746 (mES cells), 2,121,258 (OLGs), 4,594,229 (PGNs) and 8,939,526 (DNs). Next, the following formula was used to compute lsARPM values for each cell-type and per gene:

lsARPM = number of overlaps between ATAC fragments and

long gene body gene length $(10^{-6})$

× total number of ATAC fragments $(10^{-6})$

### GO analysis
GO term enrichment analysis was performed using GOElite (v.1.2.4)[82]. In Extended Data Fig. 4n, DN snATAC-seq marker genes were extracted with the getMarkerFeatures function from ArchR with default parameters. Marker genes were selected as genes with $\log_2$ fold change values

of >1 and false discovery rate of <0.01 in the DN cluster compared with all clusters from the VTA (total of 973 genes). All unique genes were used as the background GO dataset. In Fig. 2c, all genes expressed in at least one cell type, annotated to mm10, were used as the background dataset. In Fig. 4d, e, all genes expressed in PGNs or DNs were used as the background dataset, and in Fig. 5a, b, all unique genes were used. Default parameters were used for the GO enrichment: GO terms that were enriched above the background (significant permuted $P$ values of <0.05, 2,000 permutations) were pruned to select the terms with the largest $Z$-score (>1.96) relative to all corresponding child or parent paths in a network of related terms (genes changed >2). GO terms which had a permuted $P$ value of ≥0.01, contained fewer than 6 genes per GO term or from the 'cellular_component' ontology, were not reported in the main figures. A full list of unfiltered GO terms can be found in Supplementary Table 7.

### MELTRON pipeline
To assess gene insulation differences, insulation square values at 10 length scales (100–1,000 kb) were calculated for genes >300 kb in length ($n = 479$; calculated for a minimum 8× 50-kb bins, that is, 400 kb minimum length). Cumulative probability distributions of insulation square values were calculated for each dataset, and the brain cells were compared to mES cell probability distributions for each gene by computing the maximum distance between the distributions and applying a Kolmogorov–Smirnov test. $P$ values were corrected for multiple testing using the Bonferroni method, and $-\log_{10}$ transformed to obtain a domain melting score. Domain melting scores for each gene in each comparison can be found in Supplementary Table 8. For visualization, empirical cumulative probabilities and insulation score values were smoothed using a Gaussian kernel density estimate (adjust = 0.3).

### Calculation of the *trans*–*cis* contact ratio
To determine the interaction strength of contacts to all (*trans*) somatic chromosomes relative to interaction strength to their own (*cis*) chromosome, *cis* and *trans* NPMI-normalized matrices were calculated at 250-kb resolution. Bins detected in less than 3%, or more than 75%, of 3 NP samples were removed from the analysis. To be sensitive to outliers, NPMI values of both *cis* ($\mathrm{NPMI_C}$) and *trans* ($\mathrm{NPMI_T}$) contacts for every bin were summarized with the arithmetic mean. The *trans*–*cis* contact ratio was then obtained using the following formula:

$$trans\text{–}cis \text{ contact ratio} = \frac{\sum \mathrm{NPMI_T} \div \text{genomic bins}(n_T)}{\sum \mathrm{NPMI_C} \div \text{genomic bins}(n_C)}$$

*Trans*–*cis* values of bins spanning long genes were summarized with the median.

### Modelling and in silico GAM
To reconstruct 3D conformations of the *Nrxn3* locus, we employed the Strings & Binders Switch (SBS) polymer model of chromatin[83,84]. In the SBS model, a chromatin region is modelled as a self-avoiding chain of beads, including different binding sites for diffusing, cognate, molecular binders. Binding sites of the same type can be bridged by their cognate binders, which then drives polymer folding. The optimal SBS polymers for the *Nrxn3* locus in mES cells and DNs were inferred using PRISMR, a machine-learning-based procedure that finds the minimal arrangement of the polymer binding sites that best describe input pairwise contact data, such as Hi-C[22] or GAM[85]. Here, PRISMR was applied to the GAM experimental data by considering the NPMI normalization on a 4.8 Mb region around the *Nrxn3* gene (chromosome 12: 87,600,000–92,400,000; mm10) at 50-kb resolution in mES cells and DNs. The procedure returned optimal SBS polymer chains made of 1,440 beads, including 7 different types of binding sites, in both cell types. A full list of $x$, $y$ and $z$ coordinates for mES cell and DN polymer model structures can be found in Supplementary Tables 9 and 10, respectively.

Next, to generate thermodynamic ensembles of 3D conformations of the locus, molecular dynamics simulations were run of the optimal polymers, using the freely available LAMMPS software (v.5june2019)[86]. In these simulations, the system evolves according to the Langevin equation, with dynamics parameters derived from classical polymer physics studies[87]. Polymers are first initialized in self-avoiding conformations and then left to evolve to reach their equilibrium globular phase[83]. Beads and binders have the same diameter $\sigma = 1$, expressed in dimensionless units, and experience a hard-core repulsion by use of a truncated Lennard–Jones potential. Analogously, attractive interactions are modelled with short-ranged Lennard–Jones potentials[83]. A range of affinities between beads and cognate binders were sampled in the weak biochemical range, from $3.0\,K_B T$ to $8.0\,K_B T$ (where $K_B$ is the Boltzmann constant and $T$ the system temperature). In addition, binders interact nonspecifically with the polymer with a lower affinity, sampled from $0\,K_B T$ to $2.7\,K_B T$. For the sake of simplicity, the same affinity strengths were used for all different binding site types. The total binder concentration was taken above the polymer coil–globule transition threshold[83]. For each of the considered cases, ensembles of up to 450 distinct equilibrium configurations were derived. Full details about the model and simulations are discussed in Barbieri et al.[83] and Chiariello et al.[84].

In silico GAM NPMI matrices were obtained from the ensemble of 3D structures by applying the in silico GAM algorithm[10], here generalized to simulate the GAM protocol with 3 NPs per GAM sample and to perform NPMI normalization. In silico GAM NPMI matrices can be obtained using previously published algorithms[10], by aggregating the content of three in silico slices into one tube, and then applying the NPMI normalization formula (see the section 'Visualization of pairwise chromatin contact matrices', therein[10]). Specifically, the same number of slices were used as in the GAM experiments, 249 × 3 NPs for mES cellCs and 585 × 3 NPs for DNs. Pearson's correlation coefficients were used to compare the in silico and experimental NPMI GAM matrices.

Example of single 3D conformations were rendered by a third-order spline of the polymer bead positions, with regions of interest highlighted in different colours. To quantify the size and variability of the 3D structures in mES cells and DNs, the average gyration radius ($R_g$) was measured from the selected domains encompassing and surrounding the *Nrxn3* gene, expressed in dimensionless units $\sigma$ in Fig. 3d, Extended Data Fig. 7e. Analyses and plots were produced with the Anaconda package v.4.7.12, and 3D structure visualizations were produced with POV Ray, v.3.7 (http://www.povray.org/download/).

## Cryosections for FISH experiments

Fixed and cryopreserved hippocampal CA1 tissue and mES cells were cryosectioned as previously described (see 'Cryoblock preparation and cryosectioning' above) with an approximate thickness of 400 nm and transferred to glass coverslips (thickness number 1.5, diameter 10 mm) coated with laminin (Sigma-Aldrich, P8920) according to the manufacturer's instructions for the three-colour FISH experiment (TSS, middle and TES), or washed in 100% ethanol and autoclaved for the immunofluorescence whole-gene FISH experiment (nucleolus, *Rbfox1*).

## BAC probes labelling and precipitation

BACs targeting the *Rbfox1* locus (Supplementary Table 11) were obtained from the BACPAC Resources Center (https://bacpacresources.org) and amplified from glycerol stocks using a MIDIprep kit (NucleoBond Xtra BAC purification kit, Machery-Nagel, 740436). Purified BACs were labelled using a nick translation kit (Abbott Molecular, 7J0001) according to the manufacturer's instructions and the following fluorophores (all Invitrogen, Thermo Fisher Scientific): ChromaTide Alexa Fluor 488-5-dUTP (C11397), ChromaTide Alexa Fluor 568-5-dUTP (C11399) and Alexa Fluor 647-aha-dUTP (A32763). Labelled BAC probes were co-precipitated with yeast tRNA (20 μg μl⁻¹ final concentration; Invitrogen, AM7119) and mouse *Cot-1* DNA (3 μg μl⁻¹ final concentration;

Invitrogen, 18440-016) overnight at −20 °C. After clean up in 70% ethanol, the probes were dissolved in 100% deionized formamide (for 1 h; Sigma, F9037) before adding (1:1) a 2× hybridization mix (20% dextran sulfate, 0.1 M phosphate buffer in 4× saline-sodium citrate (SSC); mixing for 1 h), denatured (10 min, 80 °C), and reannealed (30 min, 37 °C) before hybridization.

## Immunolabelling before FISH

Immunofluorescence labelling of the nucleolus was performed as described above ('Immunofluorescence detection for confocal microscopy') by incubating the cryosections overnight (at 4 °C) with a mouse monoclonal antibody anti-nucleophosmin B23 (a gift from H. Busch[49]), followed by incubation (1 h) with donkey antibodies raised against mouse IgG conjugated with Alexa Fluor-555 (Invitrogen). Before cryo-FISH, the bound antibodies were fixed (1 h, 4 °C) in 8% depolymerized PFA (EM-grade) in 250 mM HEPES–NaOH (pH 7.6) and rinsed in PBS.

## Cryo-FISH

Cryo-FISH was performed as previously described[2,23] with a few modifications. In brief, cryosections were washed (30 min) in 1× PBS, rinsed with 2× SSC (Sigma, S6639) and incubated (2 h, 37 °C) in 250 μg ml⁻¹ RNase A (Sigma, R4642) in 2× SSC. After washing in 2× SSC, cryosections were treated (10 min) with 0.1 M HCl, dehydrated in ethanol (30%, 50%, 70%, 90%, 100% series, 3 min each on ice) and denatured (10 min) at 80 °C in 70% formamide, 2× SSC, 0.05 M phosphate buffer (pH 7.4). Cryosections were dehydrated as described above, and overlaid on hybridization mixture on HybriSlip (Invitrogen, H18202). After sealing with rubber cement and incubation (48 h, 37 °C) in a moist chamber, cryosections were washed (25 min, 42 °C) in 50% formamide in 2× SSC, (30 min, 60 °C) in 0.1× SSC and (10 min, 42 °C) in 0.1% Triton X-100 in 4× SSC. After rinsing with 1× PBS, coverslips were mounted in Vectashield mounting medium (anti-Fading) with DAPI (Vector Laboratories, H-1200).

## Cryo-FISH microscopy

Cryo-FISH images were collected sequentially with a Leica TCS SP8-STED confocal microscope (Leica Microsystems DMI6000B-CS) using Leica Application Suite X v.3.5.5.19976 and a HC PL APO CS2 ×63/1.40 oil objective (numerical aperture of 1.4, Plan Apochromat) (see 'Immunofluorescence detection for confocal microscopy') using the following settings: 405-nm excitation and 420–500-nm emission (for DAPI), 488-nm excitation and 510–535-nm emission (for probes labelled with ChromaTide Alexa Fluor-488 and for nucleophosmin), 568-nm excitation and 586–620-nm emission (for probes labelled with ChromaTide Alexa Fluor-568), 647-nm excitation and 657–700-nm emission (for probes labelled with Alexa Fluor-647), and 555-nm excitation and 586–640-nm emission (for immunofluorescence labelling of nucleophosmin with Alexa Fluor-555). All images were collected with a ×4 zoom at 1,024 × 1,024 pixel resolution (pixel size of 0.0451 μm, resolution of 22.1760 pixels μm⁻¹).

## Cryo-FISH image analysis

Images were analysed using Fiji software (v.2.0.0-rc-69/1.52p)[88]. All images were pre-processed as previously described[23]. Genomic foci were visually identified, and areas of the manually defined objects were measured using the Fiji-Area tool. For the cryo-FISH experiment combined with immunofluorescence, the location of genomic loci in relation to the nuclear lamina or nucleolus was assessed on the basis of the overlap of foci with the nucleolus (identified by nucleophosmin immunolabelling) or the nuclear lamina (as defined by the periphery of the DAPI staining) by at least three pixels. To determine the distance between the TSS, middle and TES genomic foci, we took the centre of mass of the selected objects, as defined by Fiji-Center of mass function (the brightness-weighted average of the $x$ and $y$ coordinates of all pixels within the selected areas). Distances between the objects were measured using the Fiji-Line tool between the centres of mass defined for

each object. Images for visualization in figure panels were processed using Fiji or Adobe Photoshop CS6, for which adjustments included the optimization of the dynamic signal range with contrast stretching.

### Determination of differential contacts between GAM datasets
Significant differences in pairwise contacts between a pair of GAM datasets were determined as previously described with modifications[9]. In brief, genomic windows with low detection, defined as less than 2% of the distribution of all detected genomic windows for each chromosome, were removed from both datasets to be compared. Contacts were filtered to be within 0.5–5 Mb distance and above 0.15 NPMI, and NPMI contact frequencies at each genomic distance of each chromosome were normalized by computing a $Z$-score transformation, and a differential matrix ($D$) was derived by subtracting the two $Z$-score normalized matrices[9].

### TF-binding site analysis
To find TF-binding motifs present within specific contacts, significant differential contacts were determined for DNs and PGNs. Accessible regions within the differential contacts were determined using scATAC-seq for PGNs[81] and DNs. To account for methodological differences, including lower sequencing depth in PGN scATAC-seq data (Extended Data Fig. 4l), we considered only the peaks that occurred in >10% of cells (>10 cells in DNs; >13 in PGNs). Motif finding within accessible regions in significant contacts was performed using the Regulatory Genomics Toolbox (v.0.12.3; https://www.regulatory-genomics.org/motif-analysis/introduction/) with TF motifs (from the HOCOMOCO database, v.11)[89] obtained for TFs expressed in either DNs or PGNs (R-log ≥ 2.5) to determine the percentage of windows containing each TF motif. Next, TF motifs were filtered based on (1) the percentage of windows containing the motif (>5%) and (2) the differential expression in either PGNs or DNs ($-\log_{10}$(adjusted $P$ value) > 3, see 'Differential gene expression analysis' above), which resulted in 50 TF motifs for feature pair analysis (33 TF motifs from PGNs and 17 from DNs; Extended Data Fig. 9c, d).

Feature pairs associated with specific contacts were determined as previously described[9] and testing the 1,275 combinations of motif pairs (1,225 heterotypic motif pairs and 50 homotypic motif pairs). The number of contacts containing each pair of selected TF motifs ($PGN_{TF}$ and $DN_{TF}$), together with the percentage of total significant differential contacts in PGNs and DNs (PGN and DN), were used to determine the enrichment score for all TF feature pair interactions (that is, the ratio between frequencies of contacts in PGNs or DNs, ($PGN_{TF}$/PGN)/($DN_{TF}$/DN)). The effectiveness of a TF pair for discriminating between contacts from PGNs and DNs was assessed by using the information gain measure[90]. Enrichment and information gain for all TF feature pair interactions, as well as differential expression values for TFs (DNs compared to PGNs), can be found in Supplementary Table 13. The top feature pairs were extracted on basis of the highest information gain (ten feature pairs), PGN enrichment (five feature pairs) and DN enrichment (five feature pairs) scores. Contact overlaps for top feature pairs were visualized using UpSet plots.

### Network and community detection analysis of TF-binding sites in significant differential contacts
To determine the interconnectivity between different TF motifs found in accessible regions of significant differential contacts, the number of contacts for each pair of TF motifs (1,275 pairs) was determined. After filtering pairs of TF motifs involved in less than 20% of the total contacts (15,833 and 5,400 contacts minimum in PGNs and DNs, respectively), a network was built for each cell type with TF motifs as nodes and number of contacts as weighted edges. The Leiden algorithm was used to detect communities of strongly interconnected nodes, using the leiden package in R[91,92], with a resolution of 1.01 for both PGNs and DNs (Extended Data Fig. 10f, Supplementary Table 14).

### GAM aggregated contact plots
To visualize the average contact intensity for a set of genomic contacts, NPMI contact frequencies at each genomic distance of each chromosome were first normalized by computing a $Z$-score transformation. The resulting $Z$-score values were determined for each contact and for each contact in a 4-bin radius (50-kb bins). For each chromosome, $Z$-score values for each set of contacts and for the surrounding bins were summarized by the arithmetic mean. Mean values computed for each chromosome were added together and divided by the number of chromosomes.

### Reporting summary
Further information on research design is available in the Nature Research Reporting Summary linked to this paper.

## Data availability
Raw fastq sequencing files for all samples from DN, PGN and OLG GAM datasets, together with non-normalized co-segregation matrices, normalized pair-wised chromatin contacts maps and raw GAM segregation tables are available from the GEO repository under accession number GSE148792. Raw fastq sequencing files for mES cell GAM datasets are available from 4DN data portal (https://data.4dnucleome.org/). The 4DN sample IDs for all samples used in the study are available in Supplementary Table 1. All polymer model 3D structures produced for the analyses of this work are available in Supplementary Tables 9 and 10. Raw confocal and laser microdissection images, as well as images and ROIs for cryo-FISH experiments are available at: https://github.com/pombo-lab/WinickNg_Kukalev_Harabula_Nature_2021/tree/main/microscopy_images/.

Raw single cell mES cell transcriptome data are available from ENA data portal (https://www.ebi.ac.uk/ena/browser/home). The ENA sample IDs for all samples used in the study are available in Supplementary Table 15. Position sorted BAM files for ATAC-seq data from mES cells and DNs are available from the GEO repository under accession number GSE174024, together with processed bigwig files. A public UCSC session with all data produced, as well as all published data utilized in this study is available at http://genome-euro.ucsc.edu/s/Kjmorris/Winick_Ng_2021_GAMbrainpublicsession. Source data are provided with this paper.

## Code availability
Processing and plotting scripts for MELTRON and insulation scores are available at: https://github.com/pombo-lab/Meltron/. Processing and plotting scripts for the *trans–cis* contact ratios are available at https://github.com/pombo-lab/GAM_trans_cis_ratio/. Custom python and R scripts for GAM window calling, GAM quality control, GAM genome sampling quality and resolution, production of NPMI matrices, aggregated maps, $k$-means clustering, calculation of insulation scores and compartment calling were deposited in https://github.com/pombo-lab/WinickNg_Kukalev_Harabula_Nature_2021/tree/main/code/.

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

**Acknowledgements** The authors thank S. Q. Xie and A. Ashraf for help processing midbrain samples; R. A. Beagrie for sharing access to the mES cell GAM data before publication and discussions; M. Bartosovic for help with manuscript rebuttal; E. Espel for help optimizing the whole-genome amplification protocol; M. Gotthardt for providing animals and R. Jüttner for help with midbrain dissections for snATAC-seq; C. Baugher for help developing the TF motif analysis pipeline; members of the Pombo laboratory for discussions; C. Braeuning of the Scientific Genomics Platform for scientific and technical support, and A. Schütz and her team of the Protein Production and Characterization Platform, both at the Max Delbrück Center for Molecular Medicine in the Helmholtz Association, Berlin; A. Adey for providing a BAM file of the published PGN scATAC-seq analysis; and the David Garfield group for sharing an optimized protocol for the bulk ATAC-seq tagmentation reaction. A.P. and A. Akalin acknowledge support from the Helmholtz Association (Germany). A.P. and M.N. acknowledge support from the National Institutes of Health Common Fund 4D Nucleome Program grants U54DK107977 and 1UM1HG011585, and the Berlin Institute of Health (BIH). A.P. and L.Z.-R. acknowledge support by the Deutsche Forschungsgemeinschaft (DFG; German Research Foundation) International Research Training Group (IRTG2403). A.P. acknowledges support from the DFG under Germany's Excellence Strategy–EXC-2049–390688087. G.C.-B. acknowledges European Union Horizon 2020/European Research Council Consolidator Grant (EPIScOPE no. 681893), Swedish Research Council (no. 2015-03558; 2019-01360), Swedish Brain Foundation (no. FO2017-0075), Knut and Alice Wallenberg Foundation (grant 2019-0107), The Swedish Society for Medical Research (SSMF, grant JUB2019), Ming Wai Lau Centre for Reparative Medicine and Karolinska Institutet. G.D. and G.A. acknowledge support from the Austrian FWF through DK W1206 'Signal Processing in Neurons' and SFB F44 'Cell Signaling in Chronic CNS Disorders', P25014-B24. M.A.U. acknowledges funding by the Medical Research Council (UK) (U120085816) and a Royal Society University Research Fellowship. M.N. thanks support from CINECA ISCRA Grant HP10CYFPS5 and HP10CRTY8P, by computer resources at INFN and Scope at the University of Naples (M.N.). L.W. acknowledges the support of Ohio University's GERB program. I.H. was supported by a Boehringer Ingelheim Fonds PhD fellowship, E.T.T. by an EMBO short-term fellowship (ASTF 336-2015), and I.I.-A. by a Long-Term Fellowship from the Federation of European Biochemical Societies (FEBS). S.C. is a GABBA PhD fellow supported by the FCT (Fundação para a Ciência e Tecnologia; PD/BD/135453/2017).

**Author contributions** The authors consider the joint first authors to have extensively contributed to this work and consider that A.K. and I.H. contributed equally. A.P. designed the concept for this work. W.W.-N., M.M., A. Abentung, E.J.P. and A.P. collected animal tissue samples. I.H., W.W.-N., L.S., A.K., M.M. and R.K. produced the GAM datasets. I.H., L.S., A.K., W.W.-N. and M.M. optimized the experimental immunoGAM protocol. W.W.-N., A.K., C.J.T., I.I.-A., E.I. and T.M.S. developed the computational pipelines for bioinformatics and QC analyses of the GAM data. A.K. and I.I.-A. performed the QC analyses of the GAM data. T.M.S. and C.J.T. developed the NPMI normalization of the GAM data. T.M.S. performed the bias analysis of the GAM data. W.W.-N., A.K. and D.S. performed the bioinformatics analyses of the GAM data. D.S. developed the domain melting analysis, with consultation from C.J.T. D.S. devised the MELTRON pipeline and performed the domain melting analysis of the GAM data. D.S. developed the *trans–cis* ratio analysis. C.J.T. initially developed the differential contact approach[9], and W.W.-N. and C.J.T. further developed and adapted this analysis for the current study. W.W.-N. performed the differential contact analysis. Y.Z. performed the TF motif finding enrichment, network and community analyses in differential contacts. W.W.-N. performed the post-hoc analyses of TF motif enrichment in differential contacts. E.T.T. performed the mES cell culture experiments. E.T.T. and A.A.K. produced the scRNA-seq data. L.Z.-R. and D.S. performed the RNA-seq analysis. L.Z.-R. performed the differential gene expression analysis. L.Z.-R. optimized the snATAC-seq protocol, produced and analysed the bulk and scATAC-seq data. S.B. optimized the polymer modelling method and performed PRISMR analysis. S.B. and A.M.C. produced models for polymer modelling. A.M.C. performed the statistical analyses of polymer models. L.F. and F.M. performed the in silico GAM experiments. I.H. and S.C. performed and analysed the FISH experiments. W.W.-N., D.S., I.H., L.Z.-R., Y.Z., A.K., T.M.S., S.B., A.M.C. and M.M. produced the final data plots. S.A.T. supervised the scRNA-seq experiments. M.M., G.A., G.D., M.A.U. and G.C.-B. supervised the animal tissue collection and provided animal samples. A.P. and S.C. supervised the cryo-FISH experiments. A.P., A.K. and W.W.-N. supervised the GAM experiments. A.P. and W.W.-N. supervised the ATAC-seq. V.F. and A. Akalin supervised the RNA-seq. M.N. supervised the polymer modelling and in silico GAM. L.W. supervised the TF motif and network analyses. W.W.-N., A.P., A.K., I.H., D.S., L.Z.-R., L.W., Y.Z., G.C.-B., M.M., S.B., A.M.C. and M.N. contributed to the interpretation of the results. W.W.-N. wrote the first draft of the manuscript. W.W.-N. designed the figures and illustrations, with contributions from A.P., D.S. and I.H. W.W.-N., A.P., A.K., I.H., D.S., L.Z.-R., Y.Z., L.W., G.C.-B., S.C., S.B. and A.M.C. wrote the manuscript. All authors provided critical feedback and helped revise the manuscript.

**Competing** interests In the past 3 years, S.A.T. has acted as a consultant for Genentech and Roche, and is a remunerated member of Scientific Advisory Boards of Biogen, GlaxoSmithKline and Foresite Labs. A.P. and M.N. hold a patent on GAM[93].

**Additional information**
**Correspondence and requests for materials** should be addressed to Warren Winick-Ng or Ana Pombo.

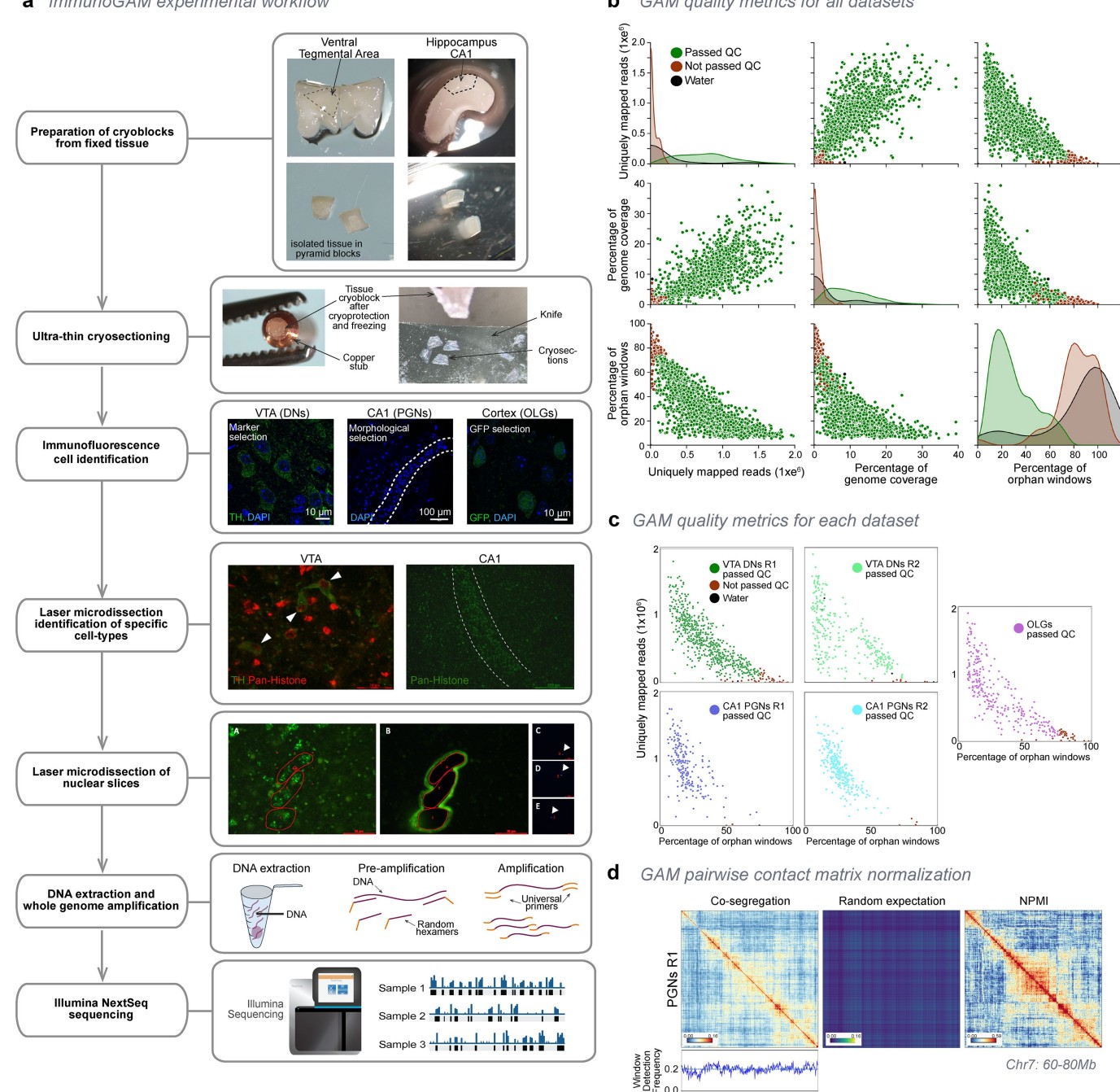

**a** *ImmunoGAM experimental workflow*

Preparation of cryoblocks from fixed tissue

Ultra-thin cryosectioning

Immunofluorescence cell identification

Laser microdissection identification of specific cell-types

Laser microdissection of nuclear slices

DNA extraction and whole genome amplification

Illumina NextSeq sequencing

**b** *GAM quality metrics for all datasets*

**c** *GAM quality metrics for each dataset*

**d** *GAM pairwise contact matrix normalization*

**Extended Data Fig. 1 | ImmunoGAM experimental pipeline and GAM data quality control. a**, ImmunoGAM experimental pipeline. VTA and CA1 dissections and cryoblock preparations are shown as examples. After fixation, brain tissue is dissected and cryopreserved in sucrose/PBS solution, before sectioning on an ultracryomicrotome (-220nm thick tissue slices; −100 °C). For confocal imaging, DAPI staining labels nuclear slices and helps to morphologically identify the CA1 PGN layer in the hippocampus, or was combined with TH immunolabelling to identify DNs in the midbrain, or with GFP immunolabelling to identify OLG lineage cells in the cortex (scale bars = 10 µm for OLGs and DNs, 100 µm for PGNs). For laser microdissection, nuclei were identified by indirect immunofluorescence using anti-pan-histone antibodies to morphologically select PGNs of the pyramidal neuron layer, or were combined with immunofluorescence detection of TH for DNs or GFP for OLGs. Laser microdissection images are shown as examples (scale bars = 30 µm for DNs, 200 µm for PGNs). Three nuclear slices were selected and laser microdissected from the tissue to fall into the same

PCR lid, as described for multiplex-GAM[9] (scale bars = 30 µm for panels a and b, 400 µm for panels c-e). Genomic DNA content was extracted from each sample and amplified using whole-genome amplification, followed by Illumina NextSeq sequencing. **b**, Quality control parameters (uniquely mapped reads, genome coverage of positive windows, and percentage of orphan windows; see Methods) for all combined GAM samples collected from brain cell types. Each data point represents a GAM sample. Samples passing QC are shown in green, samples not passing QC in red. **c**, Percentages of uniquely mapped reads and orphan windows per GAM sample, shown separately for each dataset produced in this study. Samples not passing QC are shown in red, water control samples (laser-microdissected material not containing a nuclear profile) are shown in black. **d**, Normalized point-wise mutual information (NPMI) normalization corrects for differences in the co-segregation matrix caused by changes in the window detection frequency (WDF; see Methods). Example shown for PGNs replicate 1 (R1; chr7:60,000,000-80,000,000).

**a** *Summary of GAM datasets used in this study*

| Dataset | GAM samples collected (3NPs per sample) | Samples that did not pass quality control | | | Water controls | GAM samples passing quality control | Number of cells per dataset | % locus pairs detected at least once within 5Mb (50-kb resolution) |
|---|---|---|---|---|---|---|---|---|
| | | Orphan windows >70% | Uniquely mapped reads <50,000 | Cross-contam-inated | | | | |
| **DNs R1** | 656 | 58 | 11 | 11 | 13 | **585** | **1755** | 99.8 |
| **DNs R2** | 316 | 19 | 6 | 6 | 6 | **291** | **873** | 99.9 |
| **PGNs R1** | 218 | 7 | 2 | 2 | 2 | **209** | **627** | 99.9 |
| **PGNs R2** | 288 | 7 | 1 | 1 | 6 | **275** | **825** | 99.8 |
| **OLGs** | 335 | 46 | 4 | 4 | 0 | **290** | **870** | 99.7 |
| **mESCs** | - | - | - | - | - | **249** | **747** | 99.9 |

**b** *Normalization of biases in GAM PGNs R1 dataset*

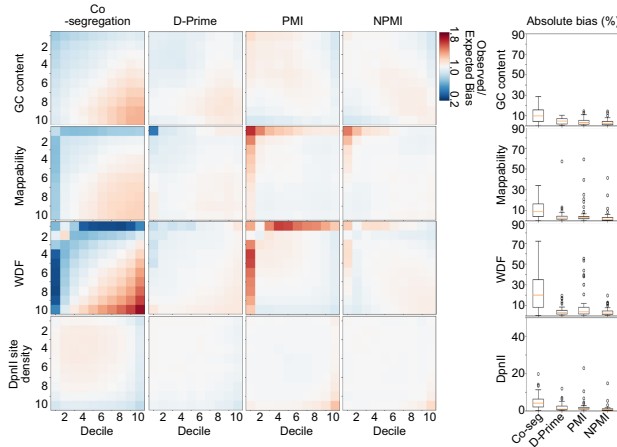

**c** *Normalization of biases in other GAM datasets*

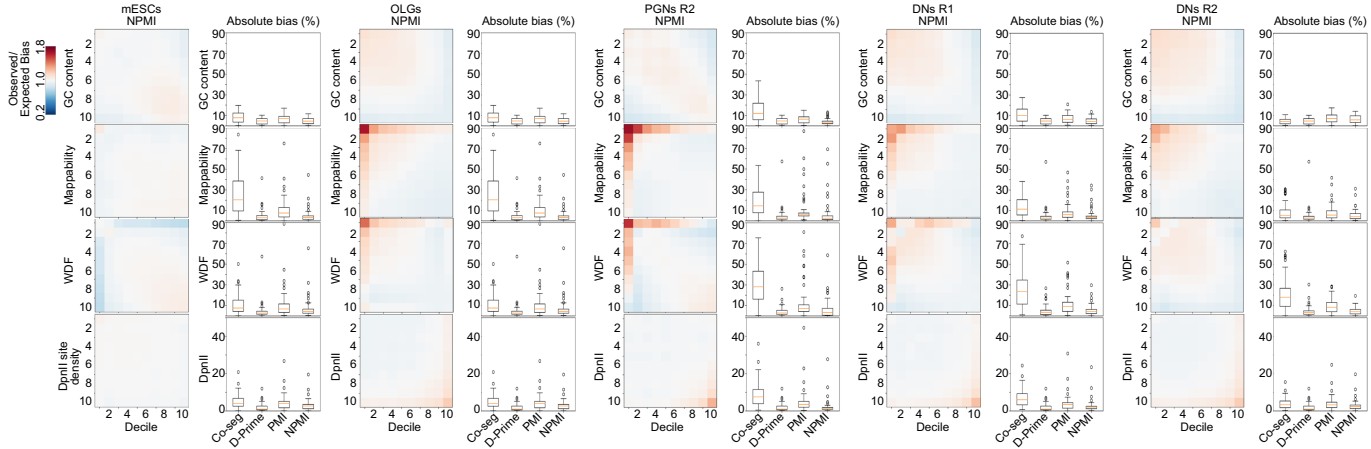

**Extended Data Fig. 2 | Normalization of immunoGAM data. a**, Summary of GAM datasets used in this study. VTA DNs were collected from two animals, an 8-week old wild-type mouse and a 10-week old mouse carrying a TH-GFP reporter. PGNs were collected from two 8-week old wildtype littermate mice. Cortical OLGs were collected based on detection of GFP expression from a 3-week old Sox10-cre-LoxP-GFP mouse. GAM data from mES cell (clone 46C) was previously published[11], and available from the 4DNucleome portal after quality control (https://data.4dnucleome.org/; Supplementary Table 1). **b**, 50-kb windows for PGNs R1 were divided into equally sized groups depending on their GC content, mappability, window detection frequency (WDF) or *DpnII* restriction density. Heatmaps of mean observed/expected bias (represented as a fold change) are shown for co-segregation, D-prime (used for previous GAM normalizations[3]), PMI and NPMI normalizations. NPMI normalization results in the lowest absolute bias percentage for all tested categories (box plots on *right*). Box plot definitions were as follows: 25th percentile lower limit, 75th percentile upper limit, and center line as the median; interquartile range (IQR) was 25th to 75th percentile; upper whisker was (75th percentile + (IQR*1.5)), lower whisker was (25th percentile − (IQR*1.5)) or zero if negative; outliers outside the whiskers were indicated with open circles. n = 100 for each bias tested, representing all combinations of deciles in PGNs R1. **c**, Absolute bias analysis for remaining immunoGAM datasets. Box plot definitions were as in panel **b**.

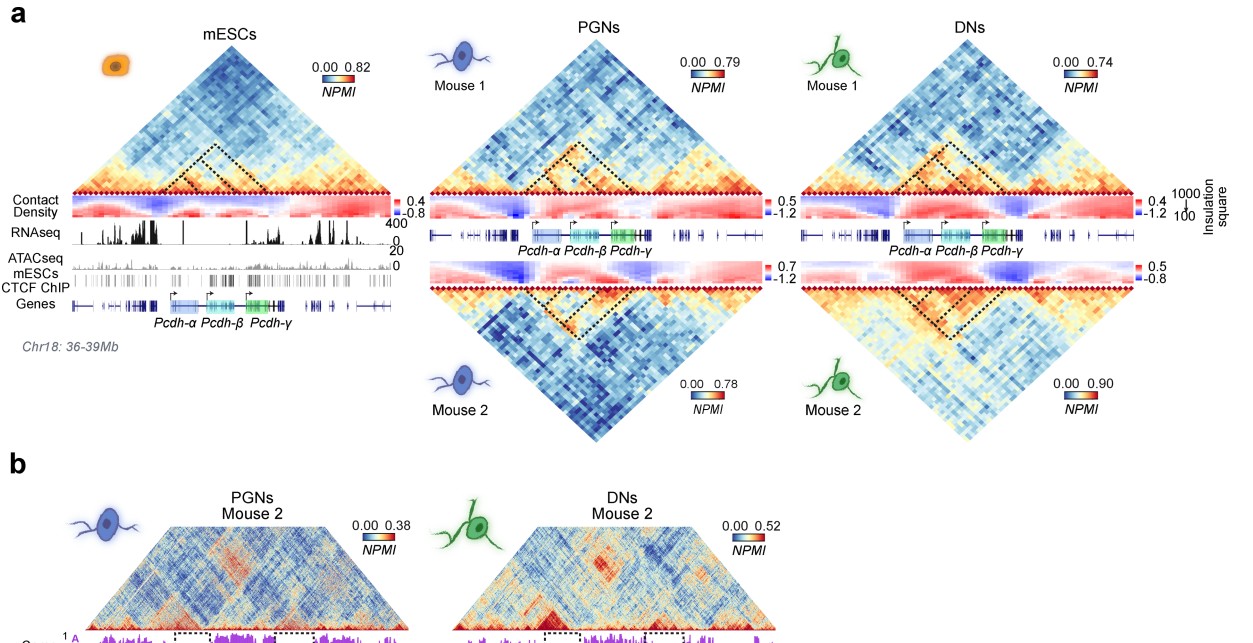

**Extended Data Fig. 3 | ImmunoGAM contact matrices from replicate mice.**
**a**, GAM contact matrices centered on the *Pcdh* gene cluster for mESC, CA1 PGN replicate 2, and VTA DN replicate 2 (Chr18: 36,000,000-39,000,000; 50-kb resolution). ChIP-seq peaks for CTCF[15] are shown below the mES cell matrix, showing extensive binding at the *Pcdh* locus. Dashed lines illustrate differences in contacts between *Pcdh*-α, -β and -γ genes for different cell types. Scale bars are adjusted to a range between the 0 value and the 99th percentile of NPMI

values for each cell type. **b**, Example matrices for replicate 2 of CA1 PGN and VTA DN, for Chr17: 0-60,000,000 at 50-kb resolution. Dashed lines illustrate vomeronasal (*Vmn*) and olfactory (*Olfr*) receptor gene clusters within B compartments, separated by ~35 Mb, are observed in brain cells but not in mES cells. Compartments A/B were classified using normalized PCA eigenvectors[2].

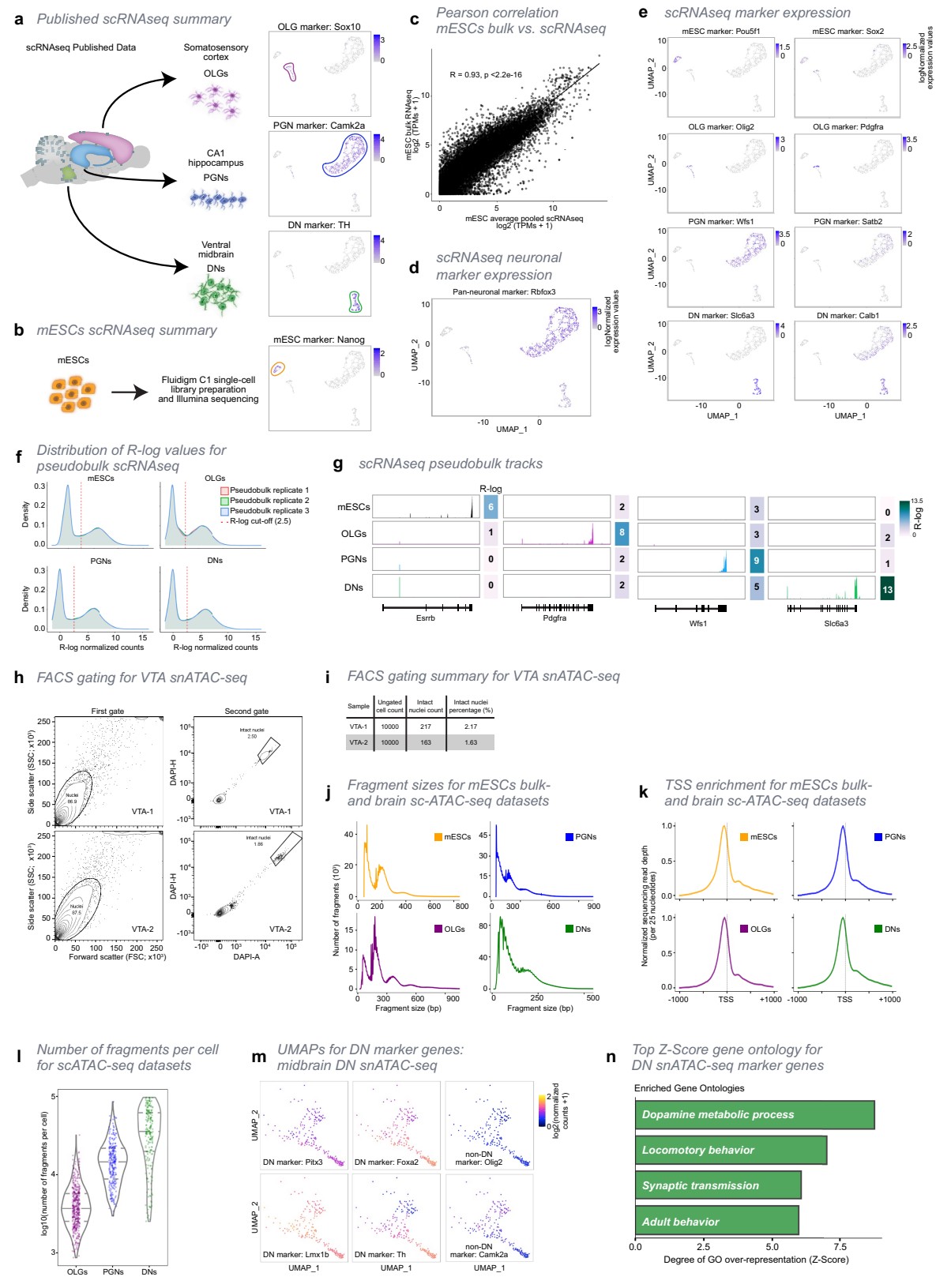

**Extended Data Fig. 4** | See next page for caption.

**Extended Data Fig. 4 | Curation of scRNA-seq and snATAC-seq data from published datasets and datasets produced for the present study.**
**a**, Schematic representation of scRNA-seq datasets used in this study. We collected published scRNA-seq datasets from cortex and hippocampus, and produced scRNA-seq from midbrain. From each of the brain tissues, we select the specific cell types that were matched with those collected for the presented GAM data. The selected datasets from each cell type were combined and visualized through UMAP embedding, coloured by expression of each marker gene: *Sox10* for OLGs, *Camk2a* for PGNs and *Th* for DNs. Cluster contours are drawn to highlight separation between cell types. All marker genes were found highly expressed in their respective cell types. **b**, scRNA-seq datasets were also generated from mES cells. UMAP clustering is coloured by the expression of *Nanog*. **c**, Pearson's correlation plot of gene expression in mES cells (clone 46C) between published bulk[26] versus single-cell RNA-seq. Average single-cell expression is highly correlated with bulk RNA-seq (two-sided Pearson's R product-moment correlation; $R = 0.93$, $p < 2.2 \times 10^{-16}$). Only genes common to both datasets are represented (total genes in bulk dataset = 22822, total genes in single cell dataset = 23208, common to both = 22045). **d**, Single cell expression of *Rbfox3*, a pan-neuronal marker, overlaid on the UMAP of single cell transcriptomes. **e**, Additional examples of UMAPs for single cell transcriptomes of cell-type markers. *Pou5f1* and *Sox2* were used as markers for mES cells, *Olig2* and *Pdgfra* for OLGs, *Wfs1* and *Satb2* for PGNs, and *Slc6a3* and *Calb1* for DNs. All markers show higher expression in their respective cell types. **f**, Distribution of regularized log (R-log) values for pseudobulk scRNA-seq datasets. For each cell type, cells were randomly partitioned into 3 pseudobulk replicates before pooling and normalizing reads. The distribution of R-log values is bi-modal for all cell types and pseudobulk replicates. To consider expressed genes for downstream analysis, a 2.5 R-log threshold (dashed red lines) was applied in all datasets. Genes with R-log ≥ 2.5 in all three pseudobulk replicates are considered expressed for that cell type. **g**, Example scRNA-seq pseudobulk tracks of sequenced reads for marker genes in each cell type. Tracks were RPKM normalized to allow for cell-type comparisons. Markers were: *Esrrb* for mES cells, *Pdgfra* for OLGs, *Wfs1* for PGNs and *Slc6a3* for DNs.

All markers are specifically expressed in their respective cell types. **h**, Exemplar plots of fluorescence-activated cell sorting (FACS) and gating strategy in midbrain VTA samples. Two biological replicate samples from independent mice, VTA-1 (*top*) and VTA-2 (*bottom*) were sorted to determine percentage of intact nuclei. Debris was excluded with a first gate (*left*; SSC/FSC plots, n = 10000 for VTA-1 and VTA-2, a total of n = 200000 DAPI positive events were sorted) and damaged nuclei with a second gate using DAPI (*right*; DAPI-H/DAPI-A plots, n=8687 and 8748 for VTA-1 and VTA-2, respectively). The frequencies of parent populations are indicated by circles within the plots, and the target intact nuclei are indicated by the boxed area. **i**, Table indicating the total number of recorded events for VTA-1 and VTA-2 exemplar FACS gating as shown in Extended Data Fig. 4h, as well as the number and percentage of intact nuclei. **j**, Distribution of fragment sizes for (sc)ATAC-seq data used in this study. Bulk ATAC-seq data was generated from mES cells. snATAC-seq was generated from midbrain VTA, from which 216 nuclei were classified as DNs (see Methods). OLG and PGN scATAC-seq was collected from published data (see Methods, Supplementary Table 6). **k**, Aggregated sequencing reads at 2kb genomic regions centered on transcription start sites (TSSs). Nucleosome-free regions (NFRs; < 147 bp) were extracted from the ATAC alignment BAM files in each cell type (i.e. fragments). NFRs are enriched at the TSS for all ATAC-seq datasets. **l**, Number of fragments per cell/nucleus for sc/snATAC-seq datasets. The number of unique fragments per nucleus was highest for DNs. **m**, Single-cell accessibility maps for DNs generated in the present study were visualized together by UMAP embedding, and coloured by expression of DN marker genes or marker genes for OLGs and PGNs. Per-cell gene scores were calculated for each DNs marker gene (see Methods). DNs expressed DN-specific markers *Pitx3*, *Foxa2*, *Lmx1b* and *Th*, while not expressing OLG and PGN markers *Olig2* and *Camk2a*, respectively. **n**, Top four enriched gene ontologies (GO) for DN marker genes (973 genes; over-representation as measured by Z-Score; see Methods for marker selection), containing terms relevant for dopamine metabolism, synaptic transmission and behaviour. All enriched GOs were highly significantly enriched (one-sided Fisher's exact permuted p-values = 0).

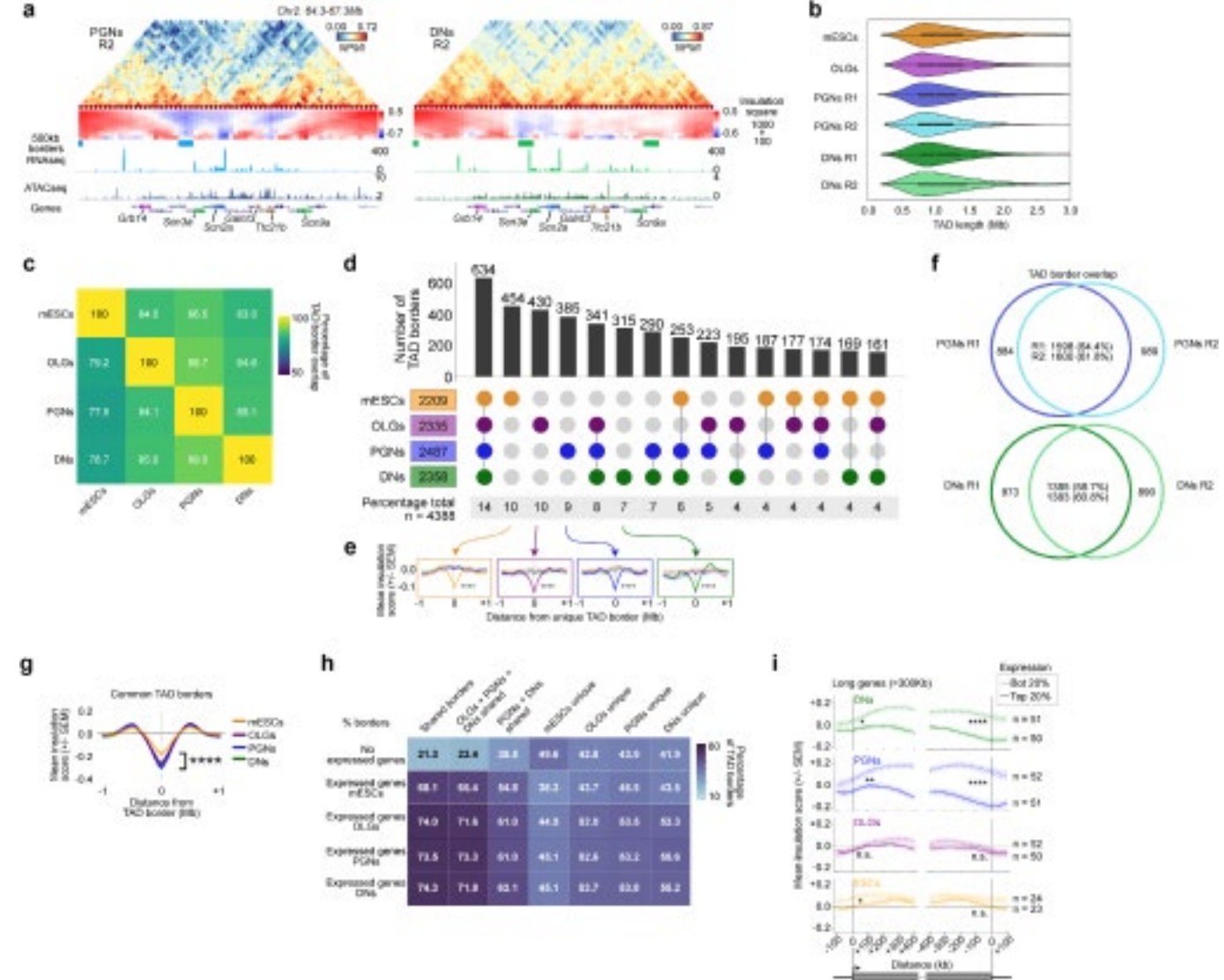

**Extended Data Fig. 5 | Identification of contact density changes, TAD borders, and differences in contacts between cell types. a**, GAM contact matrices for replicates 2 obtained from PGNs and DNs, within a 2-Mb region (50-kb resolution; Chr2:64,800,000-66,800,000). Contact density maps, TAD borders, pseudobulk scRNA-seq, and pseudobulk scATAC-seq tracks are indicated for each cell type below matrices. **b**, Distributions of TAD lengths in each GAM dataset. TAD length was calculated as the distance between two boundary points (defined as lowest insulation score point within a boundary). **c**, Pairwise comparisons of TAD boundary overlap between cell types. TAD boundaries were determined using insulation square method, using square size of 500kb, and the minimum score considered +1 bin on either side, giving a constant total of 150-kb TAD boundaries. The matrix of percentages of common TAD boundaries is not symmetrical as the percentage of overlap between boundaries varies with the direction of the comparison. The first dataset in the comparison is specified on the y axis, and the second on the x-axis. **d**, Four-way comparison of TAD boundary overlap between all cell types is shown as an UpSet plot. TAD boundaries were defined as in 5c. **e**, Average insulation score profiles centered on cell-type specific TAD borders show low average insulation scores in the cell type where the borders are detected, with highly significant differences at central border window with all other cell types (two-sided Mann-Whitney U test for central TAD border window in unique cell-type border and compared to all other cell types; ****p < 0.0001; p = 1.1x10$^{-20}$, 1.2x10$^{-17}$, and 1.0x10$^{-17}$ for mES cells compared to OLGs, PGNs and DNs, respectively; p = 6.0x10$^{-18}$, 2.4x10$^{-12}$, and 4.1x10$^{-11}$ for OLGs compared to mES cells, PGNs and DNs, respectively; p = 1.0x10$^{-10}$, 2.0x10$^{-07}$, and 1.3x10$^{-09}$ for PGNs compared to mES cells, OLGs and DNs, respectively; and p = 6.7x10$^{-10}$,

1.8x10$^{-12}$, and 8.5x10$^{-08}$ for DNs compared to mES cells, OLGs and PGNs, respectively). **f**, Venn plots show overlap between TAD boundaries in PGN or DN replicates 1 and 2. Overlaps were performed by comparing replicate 1 (R1) to replicate 2 (R2), and conversely R2 to R1. **g**, Average insulation score profiles of common TAD borders (first UpSet plot group) centered on the lowest insulation point within each TAD border are shown for each cell type (two-sided Mann-Whitney test for central TAD border window in mES cell border and compared to each brain cell-type; ****p < 0.0001; p = 8.6x10$^{-10}$, 1.5x10$^{-18}$, and 1.0x10$^{-18}$ for mES cells compared to OLGs, PGNs and DNs, respectively). **h**, Percentage of TAD borders containing expressed genes (R-log ≥ 2.5) in each cell type for the groups shown in **d**. Higher percentage of borders contain expressed genes in groups with shared borders in two or more cell types. In all groups, brain cells have a higher percentage of borders with expressed genes compared to mES cells. **i**, Average insulation score profiles at the gene TSS or TES for genes >300kb in length, using insulation square size 500kb. The top and bottom 20% expressing genes were determined using the length-normalized number of reads covering the gene body (length-scaled RNA Reads per Million; lsRRPM). The top expressing long genes have significantly lower insulation scores compared to the lowest expressed genes, at both the TSS and TES, in DNs and PGNs, while mES cells are lower at the TSS only, and OLGs show no detectable difference (two-sided Mann-Whitney test at TSS or TES windows; *p < 0.05, **p<0.01, ***p < 0.001, ****p<0.0001; p-values at the TSS, p = 0.02, 0.009, 0.328, 0.027 for DNs, PGNs, OLGs and mES cells, respectively; p-values at the TES, p = 7.2x10$^{-6}$, 1.8x10$^{-8}$, 0.323, 0.177 for DNs, PGNs, OLGs and mES cells, respectively).

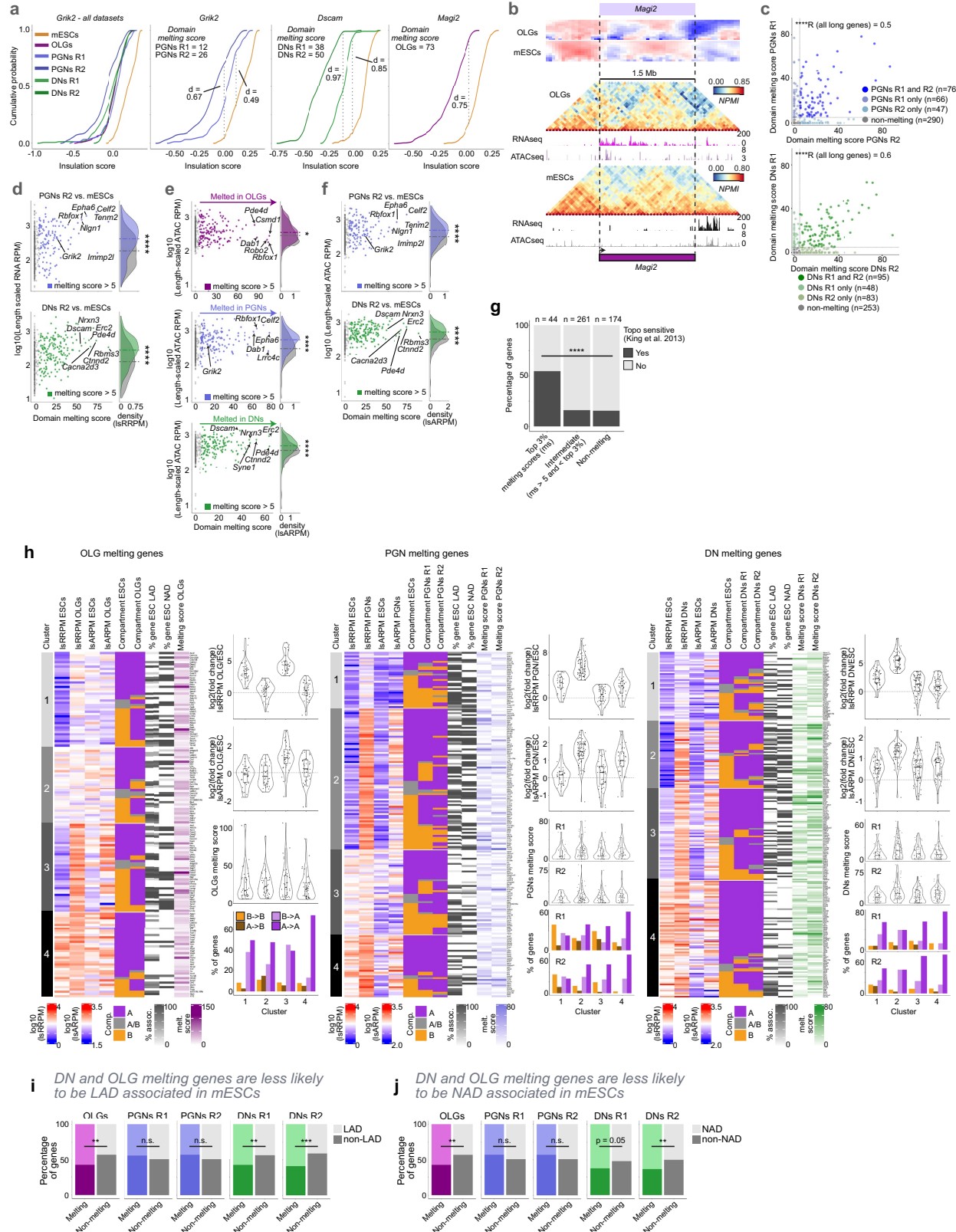

**Extended Data Fig. 6** | See next page for caption.

**Extended Data Fig. 6 | Identification of domain melting in long expressed genes. a**, Cumulative probability of insulation square scores ranging from 100 – 1000 kb for *Grik2* in all cell types and replicates (*left*). Comparison between PGNs replicates 1 and 2 and mES cells, with maximum distance (d) and TAD melting scores (*right*). Cumulative probability distributions of insulation scores and domain melting scores for *Grik2* in PGNs, *Dscam* in PGNs, and *Magi2* in OLGs (*right*). All genes were compared to mES cells, with maximum distance (d) indicated for each comparison. **b**, Example of domain melting for *Magi2* in OLGs. **c**, Correlation of replicate domain melting scores for replicates 1 and 2 in PGNs and DNs (two-sided Pearson's R product-moment correlation was calculated for all 479 long genes; ****p < 2.2x10$^{-16}$ for both PGNs and DNs;). **d**, Domain melting scores for each gene (n = 479) in PGNs R2 and DNs R2, compared to mES cells. Genes with melting scores > 5 are coloured in each cell type. Density estimates of length-scaled RNA reads per million (lsRRPM) transcription levels are shown for genes with melting scores > 5 (coloured by cell type) compared to non-melting genes (grey; two-sided Wilcoxon rank-sum test; ****p = 5.4x10$^{-9}$ and 6.5x10$^{-11}$ in PGNs and DNs, respectively). **e**, Melting genes have higher density of open chromatin regions throughout their gene bodies (length-scaled ATAC-seq RPM values; lsARPM), especially in PGNs and DNs, and to a minor extent in OLGs (two-sided Wilcoxon rank-sum test; *p<0.05, ****p<0.0001; p-values from left to right, p = 0.015, 4.0x10$^{-10}$, 1.3x10$^{-7}$). **f**, Domain melting scores compared to length-scaled ATAC-seq reads per million (lsARPM) transcription levels for each gene (n = 479) in PGNs R2 and DNs R2. Density estimates of lsARPM open chromatin levels are shown for genes with melting scores > 5 (coloured by cell type) compared to non-melting genes (grey;

two-sided Wilcoxon rank-sum test; ****p = 2.6x10$^{-6}$ and 2.2x10$^{-16}$ in PGNs and DNs, respectively). **g**, Long genes within the top 3% melting scores in any cell-type (24 of 44 genes) have a higher likelihood of sensitivity to topoisomerase inhibition[45] compared to genes with intermediate melting scores (42 of 261) and genes with no domain melting (27 of 174; two-sided $\chi^2$ test; ****p-value = 5.0e-9). **h**, Heatmaps of genes with domain melting in OLGs, and with domain melting in at least 1 replicate for PGNs and DNs, clustered by change in transcription level (length-scaled RNA RPM; lsRRPM) from mES cells to brain cell type. ATAC-seq (length-scaled ATAC RPM; lsARPM), compartments in each cell-type, and percentage of mES cell lamina- and nucleolus-associated domain (LAD[47] and NAD[48], respectively) in mES cells are shown for comparison. The density of the change in lsRRPM, lsARPM, and melting scores are shown for each cluster (violin plots on *right*). Compartment changes are shown as bar plots (*lower right*). **i**, mES cell LAD association (defined as > 50% of gene body with feature) for genes with or without melting domains in brain cell types and replicates. For DNs and OLGs, genes with domain melting were less likely to be LAD associated in mES cells, compared to non-melting genes (Two-sided Fisher's exact test; **p < 0.01, ***p<0.001; p-values from left to right, p = 0.001, 0.272, 0.209, 0.003, 0.0001). **j**, mES cell NAD association (defined as > 50% of gene body with feature) for genes with or without melting domains in brain cell-types and replicates. For DNs and OLGs, genes with domain melting were less likely to be NAD associated in mES cells, compared to non-melting genes (Two-sided Fisher's exact test; *p < 0.05, **p < 0.01; p-values from left to right, p = 0.003, 0.272 0.209, 0.055, 0.008).

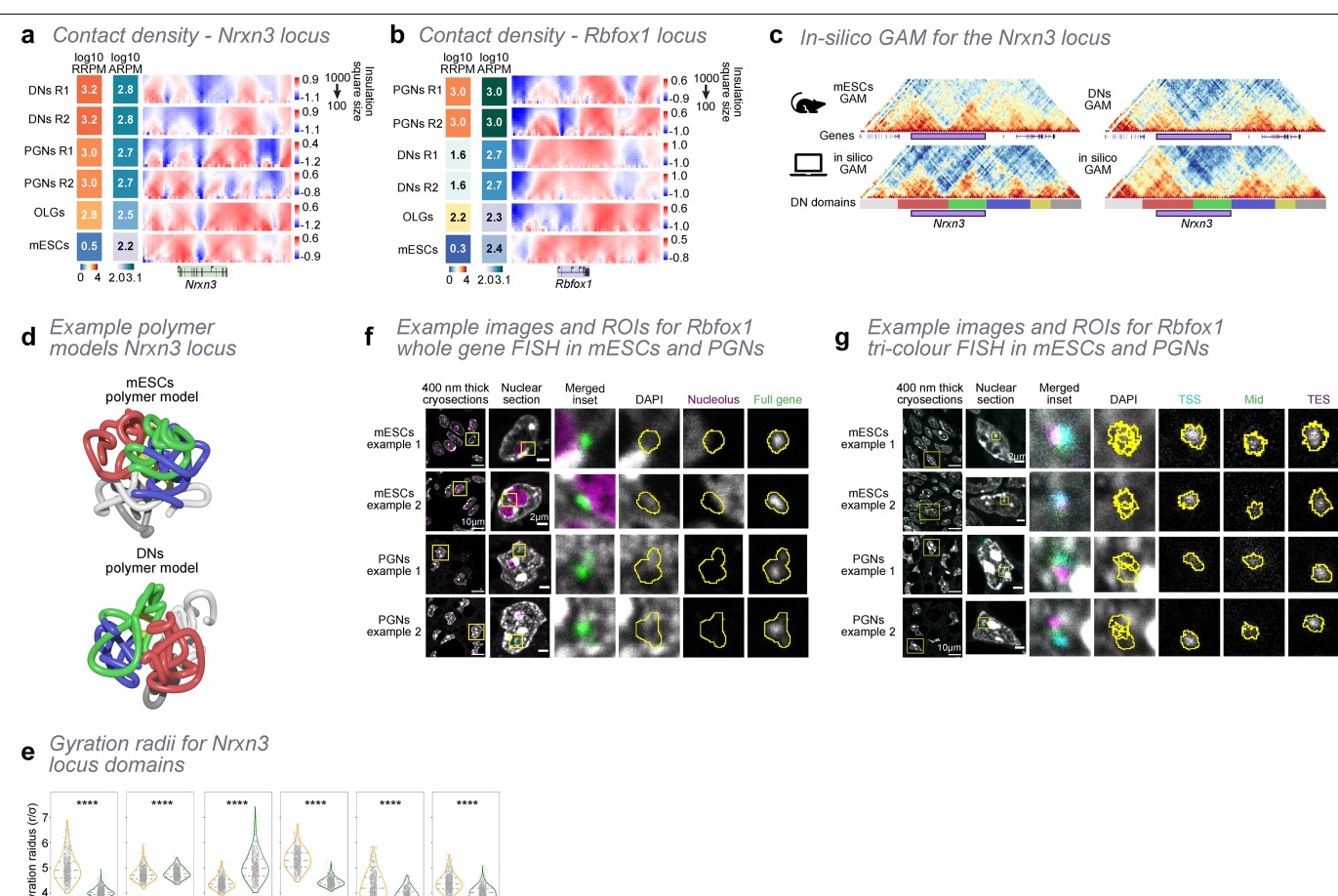

**a** *Contact density - Nrxn3 locus*

**b** *Contact density - Rbfox1 locus*

**c** *In-silico GAM for the Nrxn3 locus*

**d** *Example polymer models Nrxn3 locus*

**f** *Example images and ROIs for Rbfox1 whole gene FISH in mESCs and PGNs*

**g** *Example images and ROIs for Rbfox1 tri-colour FISH in mESCs and PGNs*

**e** *Gyration radii for Nrxn3 locus domains*

**Extended Data Fig. 7 | Characteristics and mechanisms of domain melting in long expressed genes. a**, Contact density maps for each cell type and replicate, at the *Nrxn3* locus, calculated using insulation square sizes ranging from 100 − 1000 kb. Contact density is reduced in PGNs and DNs replicate 2 (R2), similar to R1 but occurring in slightly differing regions of the gene. **b**, Contact density maps for each cell type and replicate, at the *Rbfox1* locus. Contact density is reduced in OLGs and PGNs R2, in the same region as R1. **c**, Ensembles of polymer models were produced for the *Nrxn3* locus in mES cells and in DNs from experimental GAM data using PRISMR modelling (n= 450). The quality of the models was verified by applying *in-silico* GAM to the ensemble of polymers and comparison between NPMI-normalized contact matrices from *in-silico* and experimental immunoGAM (Pearson r = 0.72 and 0.79 for mES cells and DNs, respectively). Colour bars below *in-silico* matrices highlight the position of domains in DNs and are used to colour the polymer examples shown in Fig. 3c and Extended Data Fig. 7d. **d**, Additional examples of polymer models for the *Nrxn3* locus in mES cells and DNs. The *Nrxn3* melted TAD is represented by the green coloured region and is more decondensed in

DNs than mES cells. See Fig. 3c for location and colouring of the domains. **e**, Distribution of gyration radii of all domains in polymer models for mES cells and DNs (see Fig. 3c for location and colouring of the domains; n= 450, two-sided Mann-Whitney test between mES cells and DNs; dashed lines indicate quartiles; ****p<0.0001; domains from left to right p= 3.0e-151, 0.0005, 1.1e-92, 2.0e-147, 7.3e-40, 2.5e-67). **f**, Exemplar images of whole gene cryo-FISH for *Rbfox1* (green) in mES cells and PGNs, using probes that label the whole gene. Nucleoli (purple) were detected by an anti-nucleophosmin 1 antibody. Yellow inset of the -400 nm section shows a single nucleus. Inset on nuclear section (yellow box) with *Rbfox1* FISH signal and each imaging channel. Yellow outline indicates region of *Rbfox1* signal used for area measurement and localization to nuclear landmarks. **g**, Exemplar images of tri-colour cryo-FISH for *Rbfox1* TSS (teal), Mid (green) and TES (purple) in mES cells and PGNs (see Fig. 3e for schematic). Yellow inset of the 400 nm section shows a single nucleus. Inset on nuclear section (yellow box) is shown for all three FISH signals, and each imaging channel separately. Yellow outline indicates region of *Rbfox1* signal used for center of mass distance measurements.

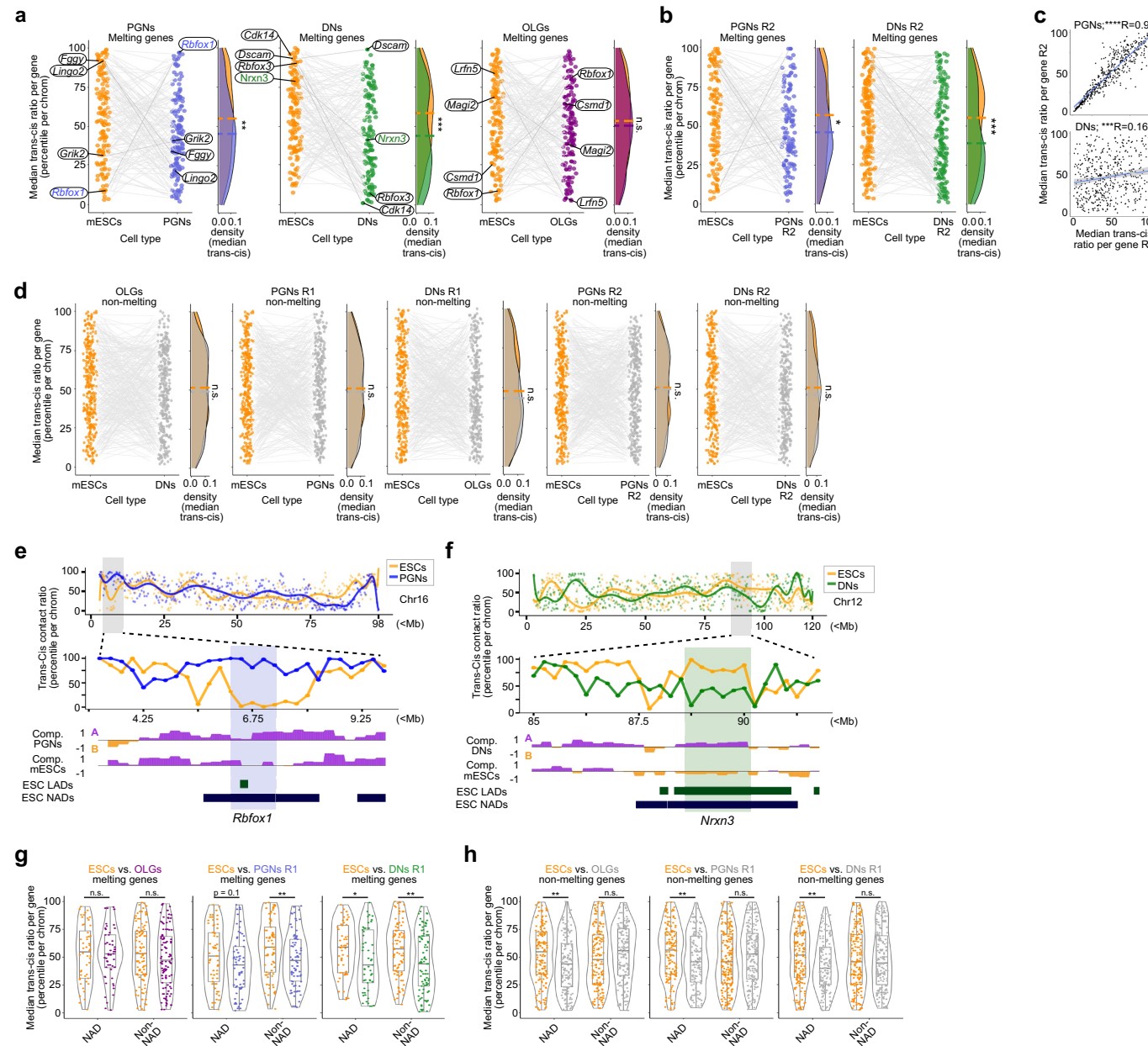

**Extended Data Fig. 8 |** See next page for caption.

**Extended Data Fig. 8 | Melting genes often show increased contacts with their own chromosome. a**, Melting genes are more likely to gain intra-chromosomal contacts in PGNs and DNs R1, but not OLGs, compared to mES cells (two-sided Wilcoxon rank-sum test; **p<0.01, ***p<0.001; p-values from left to right, p = 0.003, 0.0003, 0.329). Median *trans-cis* contact ratios were calculated for each gene with domain melting in DNs, PGNs, or OLGs, and compared to mES cells. **b**, Median *trans-cis* contact ratios were calculated for each gene with domain melting in PGNs R2 or DNs R2. Median *trans-cis* ratios were significantly lower for PGNs and DNs R2 melting genes when compared to mES cells (two-sided Wilcoxon rank-sum test; *p<0.05, ***p<0.0001; p-values were p = 0.037 and 0.0003 for PGNs and DNs, respectively). **c**, Correlation of median *trans-cis* ratios for all long genes (> 300kb) in R1 and R2 for PGNs or DNs. In PGNs, median *trans-cis* ratios were significantly correlated between replicates, with a high correlation value (Two-sided Pearson's R product-moment correlation; R=0.9, ****p < 2.2x10$^{-16}$). DNs had a lower correlation, though the correlation was still significant (R=0.16, ***p = 0.0005). **d**, Median *trans-cis* contact ratios were calculated for each gene without domain melting. Non-melting genes show no preference for changes in *trans-cis* contact ratios between brain cells and mES cells (two-sided Wilcoxon rank-sum test). **e**, The *Rbfox1* locus gains contacts with other chromosomes in PGNs, compared to mES cells. *Trans-cis* contact ratios were determined by the mean ratio between trans NPMI scores and cis NPMI scores (250kb genomic bins), and normalizing each ratio as a percentile for each chromosome. Inset (grey shaded region) shows a 7Mb region (Chr16: 3,000,000-10,000,000) containing the *Rbfox1* gene (blue shaded region). **f**, *Trans-cis* contact ratios are shown for chromosome 12 in mES cells and DNs. Inset (grey shaded region) shows a 7Mb region (Chr12: 85,000,000-92,000,000) containing the *Nrxn3* gene (green shaded region). **g**, Median *trans-cis* ratios for genes with melting domains, separated by association with NAD association (defined as > 50% of gene body with feature). For DNs, median *trans-cis* ratios were significantly decreased when compared to mES cells, regardless of association with NADs (two-sided Wilcoxon rank-sum test; *p<0.05, **p<0.01; p-values from left to right, p = 0.927, 0.233, 0.100, 0.010, 0.044, 0.003). For PGNs, median *trans-cis* ratios were significantly decreased for non-NAD associated genes (**p<0.01), and trending toward significance for NAD-associated genes, when compared to mES cells (p=0.1). OLGs had no significant differences in median *trans-cis* values for both NAD associated and non-associated genes, when compared to mES cells. **h**, Median *trans-cis* ratios for genes without melting domains, separated by association with NAD association (defined as > 50% of gene body with feature). NAD-associated genes had significantly lower *trans-cis* values in all brain cell types when compared to mES cells (two-sided Wilcoxon rank-sum test; **p<0.01; p-values from left to right, p = 0.002, 0.205, 0.013, 0.147, 0.002, 0.911). For all brain cell types, non-melting genes that were not associated with NADs had no significant differences in median *trans-cis* values when compared to mES cells.

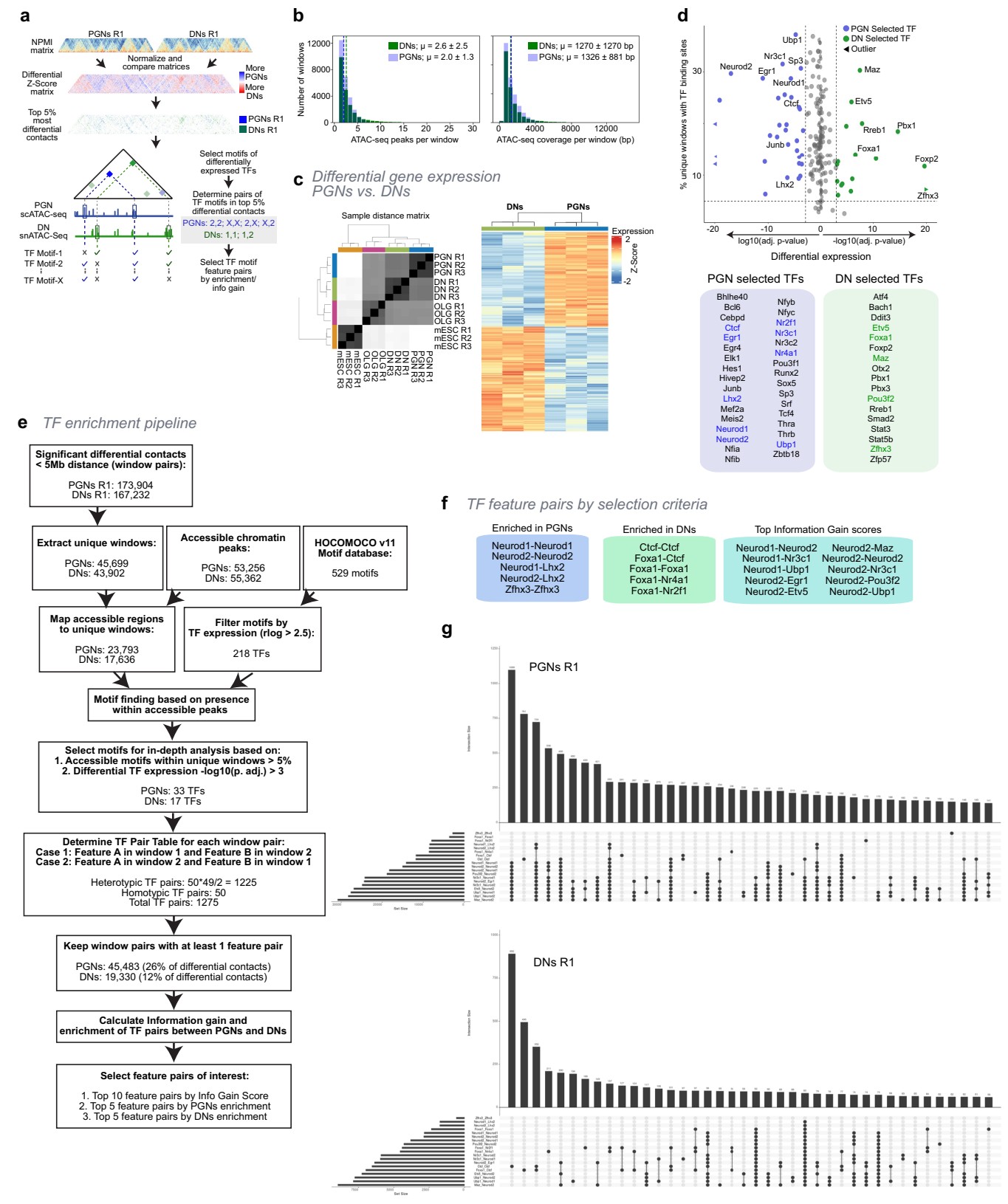

**Extended Data Fig. 9** | See next page for caption.

**Extended Data Fig. 9 | Analysis of transcription factor binding sites and differentially expressed genes in GAM differential contacts between DNs and PGNs. a**, GAM contacts from PGNs and DNs (mouse replicate 1) were normalized (Z-Score) and subtracted to produce differential contacts matrices. Top 5% differential contacts ranged 0.05-5 Mb. Contacts containing TF motifs within accessible chromatin on each contacting window were selected in most (top 5) enriched in PGNs or DNs or with highest discriminatory power (information gain). **b**, Distribution of the number of ATAC-seq peaks per 50kb GAM window in DNs and PGNs (*upper panel*; mean($\mu$) = 2.6 and 2.0 in DNs and PGNs, respectively). Number of base pairs covered by ATAC-seq peaks per 50kb GAM window in DNs and PGNs (*lower panel*; $\mu$ = 1270 and 1326 in DNs and PGNs, respectively). **c**, Correlation plot of cell type and replicates for differential gene expression analysis. Pseudobulk replicates correlate most highly with one another, followed by brain cell types. *Right*, heatmap of differentially expressed (DE) genes between PGNs and DNs, clustered by cell type. **d**, Selection of TF motifs based on percentage of TF motifs in accessible regions within unique windows (> 5%) and differential expression between PGNs (Benjamini-Hochberg corrected two-sided Wald test; log10(p. adj.) < 3) and DNs (-log10(p. adj.) > 3). PGN-selected TFs (33) are shown in blue, DN-selected TFs (17) are shown in green. A list of selected TFs are shown below, with TF motifs continuing after the TF enrichment analysis in (**f**) coloured in blue (PGNs) or green (DNs). **e**, Full pipeline to determine pairs of genomic windows in GAM differential contacts containing transcription factor binding sites[9]. GAM contacts from PGNs and DNs were normalized and compared to produce a differential Z-Score matrix with a 0.05-5 Mb distance range. The top 5% differential contacts with > 0.15 NPMI values for each dataset were extracted from the differential matrices. Accessible chromatin regions were mapped to the top differential contacts. Next, TF motifs were filtered based on expression in at least one cell type. Accessible regions in differential contacts were used to determine the percentage of TF motifs within unique windows. To find TFs with the potential to drive contact specificity between DNs and PGNs, we chose for further analyses the TF motifs that were found in DN or PGN accessible regions within differential contacts which (1) were present in at least 5% of contacts, and (2) the TFs were differentially expressed between DNs and PGNs (-log10 (p.adj.) > 3). The 50 TFs which met the requirements were further investigated to determine the frequency of each motif pair (TF feature pair) in PGN and DN differential contacts. The top-20 TF feature pairs were selected for further analyses based: (a) on Information gain score (top 10 feature pairs selected), and (b) on enrichment in either PGNs (top 5 selected) or DNs (top 5 selected). **f**, TF motif pairs selected by enrichment scores in DNs or PGNs, or by the highest Information gain scores. **g**, Overlaps of top 20 TF feature pair contacts for PGN and DN significant differential contacts. The top 40 groups with overlapping TF features are shown for each cell type.

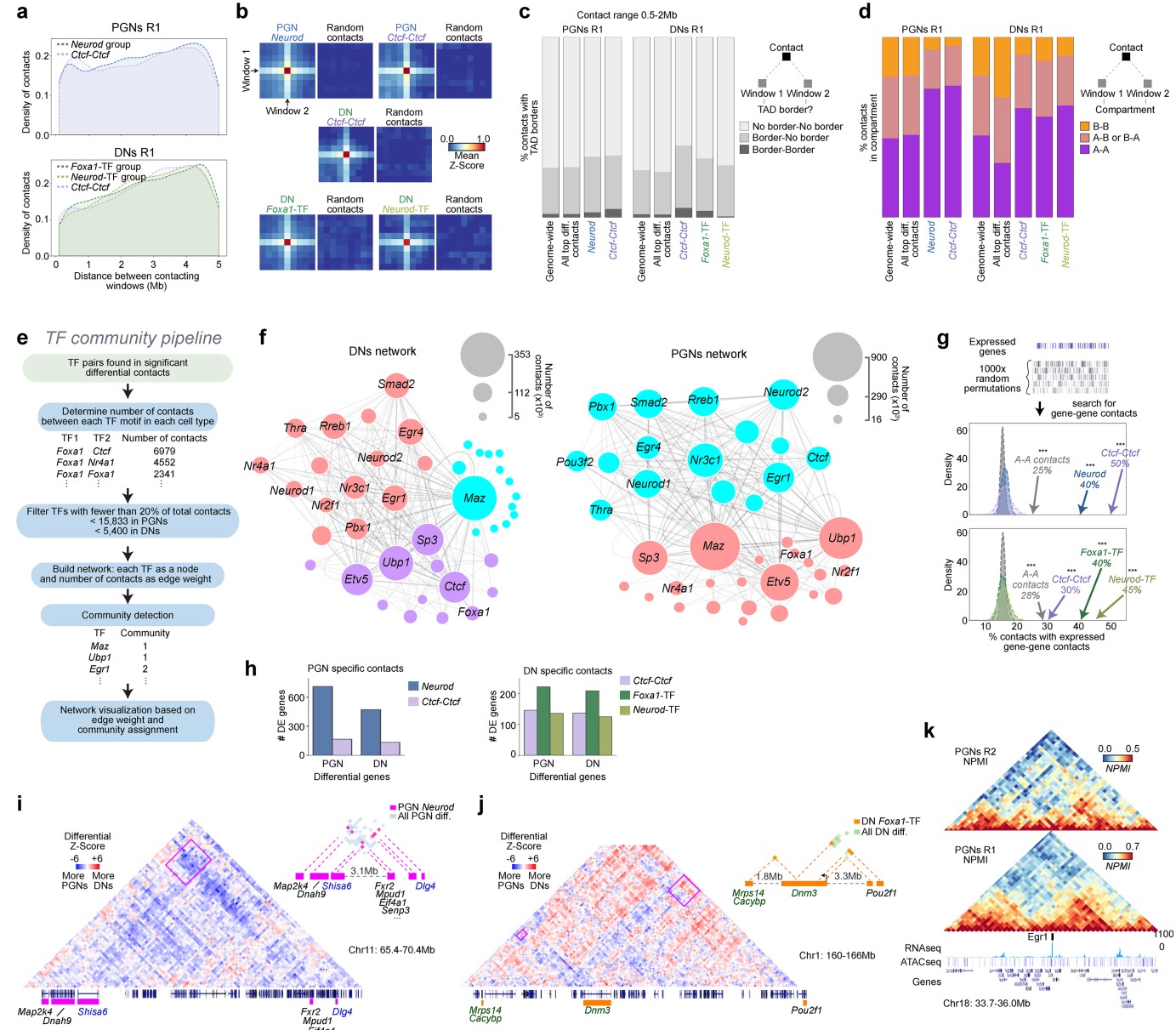

**Extended Data Fig. 10 | Features of top differential contacts containing pairs of TF binding sites. a**, Percentage of contacts at each genomic distance for top differential contacts found in TF feature pair groups. Contacts in all groups are enriched at distances > 2 Mb. **b**, Aggregated maps of average Z-scores for TF-containing contact groups in PGNs and DNs. The Z-Score was determined for each contact and a 200kb (4 genomic bin) radius. For each group, chromosome- and distance-matched contacts were randomly sampled three times from the genome-wide distribution (one exemplar is shown for each group). **c**, Percentage of contacts (< 2 Mb) that fall within a TAD border in both windows, one window or no windows. For both cell types, most contacts do not overlap with TAD borders, with a slight no differences detected for top differential contacts found in TF feature pair groups, except a modest increase for contacts that have both windows with a border for *Ctcf-Ctcf* containing contacts in both PGNs and DNs. **d**, Overlap of TF-containing contact groups with compartment identity in each contacting window. For both cell types, TF-containing contact groups were more likely to be in A-compartment in both contacting windows, compared to the genome-wide average and all top differential contacts. **e**, TF motif network and community analysis. After determining the number of contacts for each TF pair, only pairs involved in > 20% of total TF-containing contacts were considered. A network was built with each TF as a node and contacts as the edge weight. Community detection

was performed using a Leiden algorithm, before visualizing the network. **f**, Network analysis and community detection for TF motifs found within DN or PGN differential contacts. **g**, Overlap of TF-pair containing contacts with 1000 random circular permutations of PGN and DN expressed gene regions shows that the observed enrichments of contacts with genes in both windows are significantly higher than the expected distribution (two-sided t-test; ***empirical p = 0.001 for all observed values tested). The enrichments were also seen, to smaller degree than for the TF-pair containing contacts, for all contacts between A-compartment windows. **h**, Number of PGN or DN differentially expressed (DE) genes found in differential contacts according to sets of TF feature pairs. **i**, Differential Z-Score matrix showing PGN-upregulated genes that form contacts across a ~4.5-Mb linear genomic distance (pink box; Chr11: 65,400,000-70,400,000). *Upper right* inset shows PGN significant differential contacts containing the *Neurod* group (contacts are shown in pink). Genes highlighted in blue are upregulated in PGNs. **j**, Differential Z-Score matrix showing DN-upregulated genes that form contacts across a ~5-Mb linear genomic distance (pink boxes; Chr1: 160,000,000-166,000,000). *Upper right* inset shows DN significant differential contacts containing the *Foxa1*-TF group (contacts are shown in orange). Genes highlighted in green are upregulated in DNs. **k**, GAM contact matrices showing a 2.3-Mb region surrounding the *Egr1* gene for PGNs R1 and R2 (Chr18: 33,700,000-36,000,000).

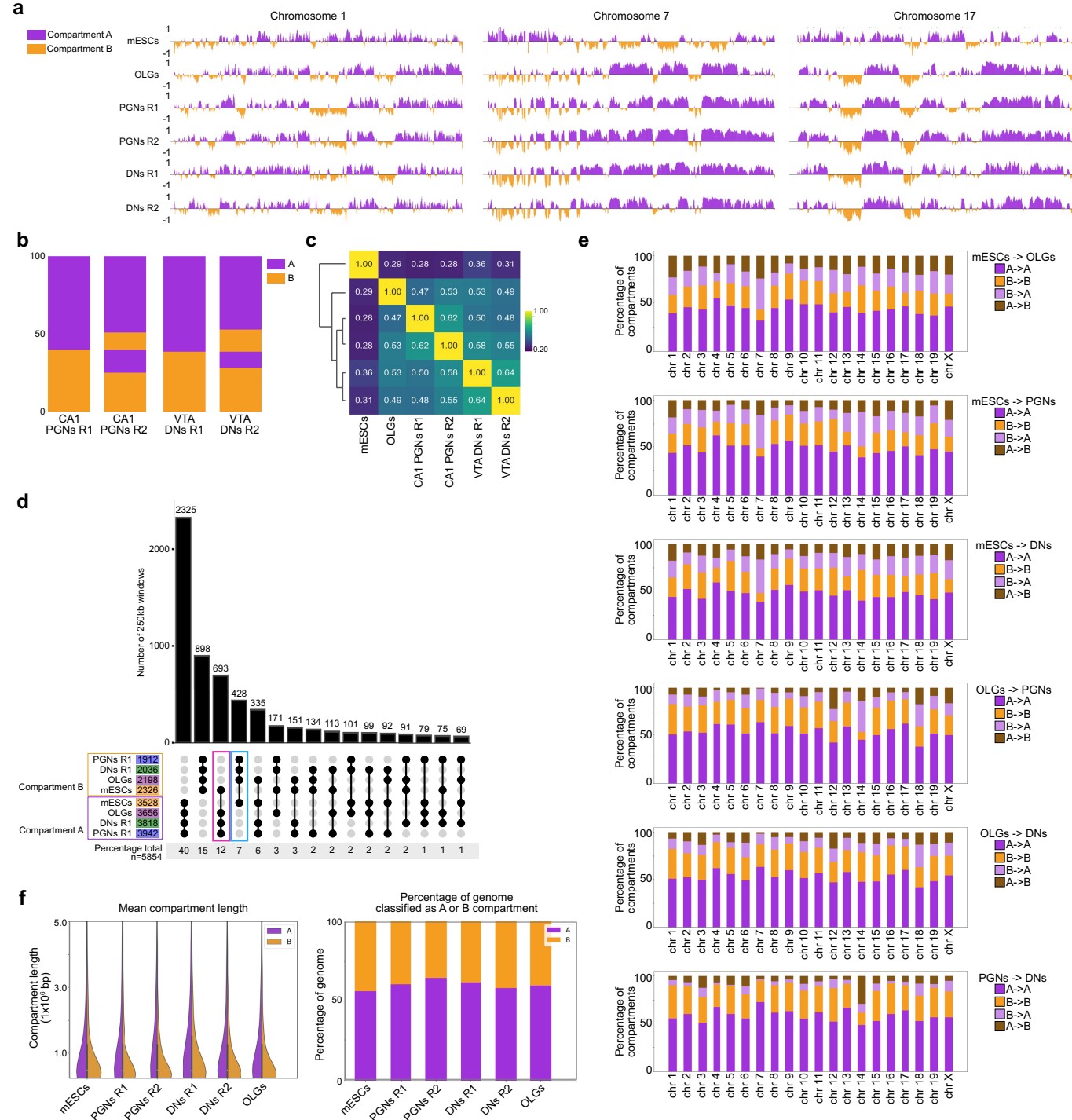

**Extended Data Fig. 11 | Identification of compartments and differences between cell types. a**, Open and closed chromatin compartments (A and B, respectively) display different genomic distributions in mES cells, OLGs, PGNs and DNs. Mouse replicates 1 and 2 (R1 and R2, respectively) are shown. Purple, compartment A; orange, compartment B. **b**, Comparison of compartment A/B membership in GAM datasets from PGNs and DNs and their replicates. Compartment changes show good overlap between replicates. Purple, compartment A; orange, compartment B. **c**, Pearson's correlation of eigenvectors shows the largest differences between mES cells and brain cell types. **d**, UpSet plot showing all combinations of compartments changes Most genomic windows share membership to compartments A, followed by B, in all

cell types. The most frequent compartment changes occur from compartment B in mES cells to A in all brain cells (pink box), followed by changes from A in mES cells to B in all brain cells (blue box). **e**, Compartment changes for each cell type comparison in each chromosome. Only compartments common to both replicates were used in the comparison. Brain cell types have higher overlap with each other as compared to mES cells. PGNs and DNs had the most overlap for most chromosomes. **f**, Violin plots of the distribution of compartment lengths show similar lengths between cell types. *Right*, percentage of the genome covered by A or B compartments in each cell type shows similar distribution between cell types.

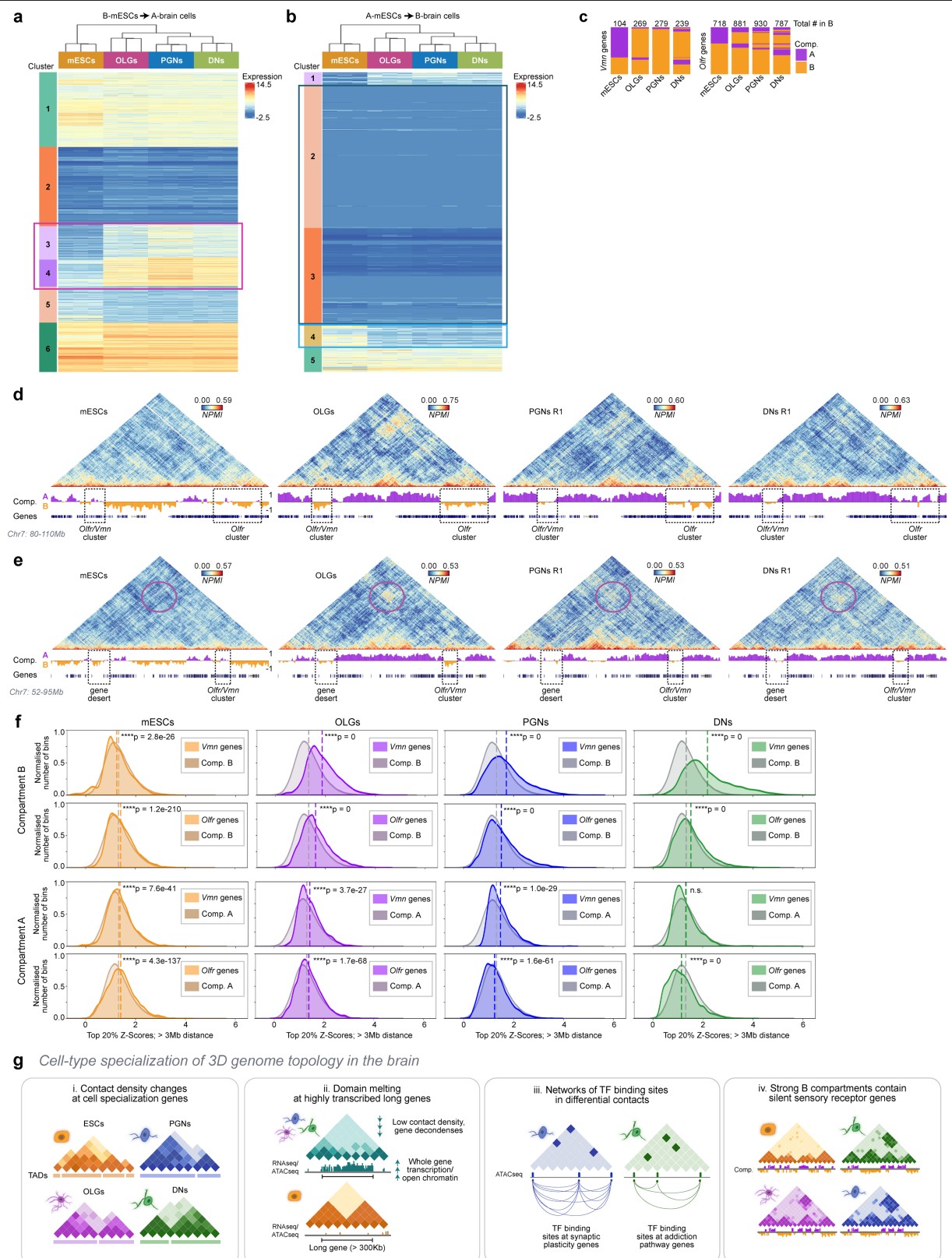

**Extended Data Fig. 12 |** See next page for caption.

**Extended Data Fig. 12 | Genomic regions involved in strong long-range contacts in brain cells regions contain sensory receptor clusters in B compartments. a**, Heatmap of gene expression for genes that change compartments between compartment B in mES cells to compartment A in all brain cells. Clustering of genes by expression shows six distinct clusters where clusters 3 and 4 contain genes that increase their expression between mES cells and all brain cell types. Gene ontology (GO) in Fig. 5a was done on genes from clusters 3 and 4 combined (pink box). Expression is calculated as the R-log value for each cell type (see Methods). **b**, Heatmap of gene expression for genes that change from compartment A in mES cells to compartment B in brain cells. Clustering of genes by expression identifies five clusters. Genes in cluster 4 are expressed in mES cells and show lower expression in the brain cell types; they were used for GO analysis presented in Fig. 5a (light blue box). Genes in clusters 2 and 3 are not expressed in mES cells nor brain cells; they were combined and used for GO analyses presented in Fig. 5b (dark blue box). Expression is calculated as the R-log value for each cell-type. **c**, A higher proportion of *Olfr* and *Vmn* genes are found in B compartments in brain cells, compared to mES cells. **d**, GAM contact matrices show interactions between an *Olfr/Vmn* gene cluster and a second *Olfr* cluster (dashed boxes) separated by 25 Mb (Chr7: 80,000,000-110,000,000). The contacts between the two receptor clusters are strongest in OLGs, where the B compartment is strongest. **e**, GAM contact matrices show strong interactions that span a 30Mb distance between compartment B regions in OLGs, PGNs and DNs (purple circle), but not mES cells (Chr7: 52,000,000-95,000,000). Dashed boxes indicate contacts containing *Olfr* and *Vmn* gene clusters. **f**, Distribution of the top 20% of Z-Score normalized contacts for each genomic window at distances > 3 Mb (Two-sided Mann-Whitney U test; exact p-values are indicated on the plot). **g**, Summary diagram. The 3D genome is extensively reorganized in brain cells to reflect its gene expression specialization. (i) Contacts are rearranged at multiple scales, where formation of new TAD borders can coincide with genes important for cell specialization in all cell types. (ii) Domain melting occurs at very long genes which are highly transcribed and with high chromatin accessibility in brain cells. (iii) The most specific contacts in neurons contain complex networks of binding sites of neuron-specific transcription factors. Contacts bridge genes expressed in the neurons where the contacts are observed, with specialized functions, such as in synaptic plasticity (PGNs) and addiction (DNs). (iv) Finally, B compartments contain clusters of sensory receptor genes silent in all cell types which form strong contacts across tens of megabases.

# nature research

# Reporting Summary

Nature Research wishes to improve the reproducibility of the work that we publish. This form provides structure for consistency and transparency in reporting. For further information on Nature Research policies, see our Editorial Policies and the Editorial Policy Checklist.

## Statistics

For all statistical analyses, confirm that the following items are present in the figure legend, table legend, main text, or Methods section.

| n/a | Confirmed | |
|---|---|---|
| ☐ | ☒ | The exact sample size (*n*) for each experimental group/condition, given as a discrete number and unit of measurement |
| ☐ | ☒ | A statement on whether measurements were taken from distinct samples or whether the same sample was measured repeatedly |
| ☐ | ☒ | The statistical test(s) used AND whether they are one- or two-sided<br>*Only common tests should be described solely by name; describe more complex techniques in the Methods section.* |
| ☒ | ☐ | A description of all covariates tested |
| ☐ | ☒ | A description of any assumptions or corrections, such as tests of normality and adjustment for multiple comparisons |
| ☐ | ☒ | A full description of the statistical parameters including central tendency (e.g. means) or other basic estimates (e.g. regression coefficient) AND variation (e.g. standard deviation) or associated estimates of uncertainty (e.g. confidence intervals) |
| ☐ | ☒ | For null hypothesis testing, the test statistic (e.g. *F*, *t*, *r*) with confidence intervals, effect sizes, degrees of freedom and *P* value noted<br>*Give P values as exact values whenever suitable.* |
| ☒ | ☐ | For Bayesian analysis, information on the choice of priors and Markov chain Monte Carlo settings |
| ☐ | ☒ | For hierarchical and complex designs, identification of the appropriate level for tests and full reporting of outcomes |
| ☐ | ☒ | Estimates of effect sizes (e.g. Cohen's *d*, Pearson's *r*), indicating how they were calculated |

*Our web collection on statistics for biologists contains articles on many of the points above.*

## Software and code

Policy information about availability of computer code

**Data collection**  Leica Laser Microdissection v8.2; Leica Application Suite X v3.5.5.19976; BD FACSDiva v8.0.2

**Data analysis**  bedtools v2.29.2; UpSetR v1.4.0; Seurat v3.1.4; samtools v1.3.1; deeptools v3.1.3; DESeq2 v1.24.0; bowtie2 v2.3.4.3; 10x Genomic Cellranger v1.2.0; ArchR v0.9.1; GenomicAlignments v1.20.1; Regulatory Genomics Toolbox v0.12.3; leiden' package v0.3.3; STAR v2.4.2a, RSEM v1.2.25, Ea-utils v1.1.2-537, Picard-tools v2.5.0; Sambamba v0.6.8; GOElite v1.2.4; LAMMPS, v.5june2019; Anaconda package v.4.7.12; POV Ray v.3; Fiji software v2.0.0-rc-69/1.52p; Adobe Photoshop CS6; UCSC utilities http://hgdownload.soe.ucsc.edu/admin/exe/; MELTRON and trans-cis contact ratio pipelines were deposited in https://github.com/pombo-lab/Meltron; custom python and R scripts for GAM window calling, GAM quality control, GAM genome sampling quality and resolution, production of NPMI matrices, aggregated maps, k-means clustering, calculation of insulation score and compartment calling were deposited in https://github.com/pombo-lab/WinickNg_Kukalev_Harabula_Nature_2021

For manuscripts utilizing custom algorithms or software that are central to the research but not yet described in published literature, software must be made available to editors and reviewers. We strongly encourage code deposition in a community repository (e.g. GitHub). See the Nature Research guidelines for submitting code & software for further information.

## Data

Policy information about availability of data

All manuscripts must include a data availability statement. This statement should provide the following information, where applicable:
- Accession codes, unique identifiers, or web links for publicly available datasets
- A list of figures that have associated raw data
- A description of any restrictions on data availability

Raw fastq sequencing files for all samples from DN, PGN and OLG GAM datasets, together with non-normalized co-segregation matrices, normalized pair-wise chromatin contacts maps and raw GAM segregation tables are available from the GEO repository under accession number GSE94364. Raw fastq sequencing files for

April 2020

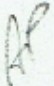

# Field-specific reporting

Please select the one below that is the best fit for your research. If you are not sure, read the appropriate sections before making your selection.

☒ Life sciences ☐ Behavioural & social sciences ☐ Ecological, evolutionary & environmental sciences

For a reference copy of the document with all sections, see nature.com/documents/nr-reporting-summary-flat.pdf

# Life sciences study design

All studies must disclose on these points even when the disclosure is negative.

| | |
|---|---|
| Sample size | The appropriate number of samples for a GAM dataset varies and depends on multiple parameters such as nuclear volume, level of chromatin compaction, quality of DNA extraction, etc. Since most of these parameters can be assessed only after the data has been collected and processed, we recommend that the optimal resolution is defined during the collection of each GAM dataset, rather than trying to estimate optimal sample size before data collection. GAM data can be collected in multiple batches from the same starting material, therefore the sample size can be increased until the desired resolution is achieved. Resolution is determined by comparing the distribution of intra-chromosomal co-segregation frequencies for all possible pairs of loci at a given resolution, using the standard Poisson distribution. In case multiple datasets from different samples are analyzed together, we recommend choosing the highest possible resolution appropriate for every dataset involved in the analysis. In the present study, we measured co-segregation frequencies for all GAM datasets, finding that 98.8 – 99.9% of all mappable pairs of windows were sampled at least once at 50 kb resolution considering all genomic distances. The script to test the quality of genome sampling at given resolution was uploaded to GitHub (https://github.com/pombo-lab/WinickNg_Kukalev_Harabula_Nature_2021/blob/main/code/GAM.define.working.resolution.py). For single cell ATAC-seq data, no statistical method was used to predetermine sample size, as in https://www.nature.com/articles/s41593-018-0079-3?proof=t For scRNA-seq (mESCs), no statistical method was used to predetermine sample size. Libraries were generated twice, from mESCs from different biological replicates, to account for experimental variability. |
| Data exclusions | The quality of individual GAM libraries was determined using a combination of several quality metrics: clustering of positive windows, sequencing depth and lack of sample contamination. Due to the nature of genome sampling by ultrathin cryosectioning, good quality positive windows are expected to cluster next to each other, while noise is expected to behave randomly and not cluster on the linear genome sequence. Positive windows in low quality GAM samples (i.e. from the water controls, or samples not amplified during the whole-genome amplification reaction) often do not cluster with other positive windows, termed "orphan windows". In this study, an individual GAM sample was considered to be of good quality if it had < 70% orphan windows, > 50,000 uniquely mapped reads and no sign of cross-well contamination, as determined by low Jaccard index score to the distribution of positive windows in all samples processed at the same time. For single-cell ATACseq (midbrain VTA), single cells were considered of low quality (and removed from the analysis) if TSS enrichment score was < 4 and there were < 2500 unique fragments per cell. After processing of raw data and clustering, the DN population was identified (see Methods) and single-cell IDs were extracted. ATAC-seq fragments derived from DN single-cells were subset from the original VTA position sorted BAM file and grouped into a subset containing only DN fragments. The subset file was uploaded to GEO (GSE174024). For scRNA-seq (mESCs), libraries were excluded from the analysis if they were derived from cells that appeared as debris or doublets/multiplets upon visual inspections of the C1 chip, or if the libraries appeared as outliers in number of sequencing reads or mapping statistics, as fully detailed in the Methods section. |
| Replication | For the two neuronal cell types, single animal replicates were produced and had similar results in all metrics tested. Variations in replicates are reported through the main and supplemental data. |
| Randomization | Randomization was not relevant to our study. The experiments and the subsequent analysis were performed on wild type animals or cell lines, where no treatment or disease comparison was performed. As described in the Methods section, our samples were processed in different labs by different people. There was no selection criteria for the wild type mice used in the study. |
| Blinding | Blinding was not relevant to our study. We did not perform clinical trials, nor compared disease models or different treatments. As described in the Methods section our samples were processed in different labs by different people. There was no selection criteria for the wild type mice used in the study. |

# Reporting for specific materials, systems and methods

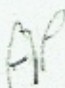

We require information from authors about some types of materials, experimental systems and methods used in many studies. Here, indicate whether each material, system or method listed is relevant to your study. If you are not sure if a list item applies to your research, read the appropriate section before selecting a response.

## Materials & experimental systems

| n/a | Involved in the study |
|---|---|
| ☐ | ☒ Antibodies |
| ☐ | ☒ Eukaryotic cell lines |
| ☒ | ☐ Palaeontology and archaeology |
| ☐ | ☒ Animals and other organisms |
| ☒ | ☐ Human research participants |
| ☒ | ☐ Clinical data |
| ☒ | ☐ Dual use research of concern |

## Methods

| n/a | Involved in the study |
|---|---|
| ☒ | ☐ ChIP-seq |
| ☐ | ☒ Flow cytometry |
| ☒ | ☐ MRI-based neuroimaging |

## Antibodies

| Antibodies used | Pel-Freez Arkansas, Catalog number P60101-0; Sheep anti-tyrosine Hydroxylase, Lot number ajo1217p;<br>Merck, Catalog number MAB3422, Mouse anti-pan-histone, clone H11-4, Lot number 2842169;<br>Abcam, Catalog number ab13970 Chicken anti-GFP; Lot number GR236651-21;<br>Invitrogen, Donkey anti-sheep Ig, AlexaFluor-488, Catalog number A-11015;<br>mouse anti-nucleophosmin B23 was a kind gift from Harris Busch;<br>Abcam, Goat anti-chicken If, AlexaFluor-488, Catalog number ab150169;<br>Invitrogen, Donkey anti-mouse Ig, AlexaFluor-488, Catalog number A-10035;<br>Invitrogen, Donkey anti-mouse Ig, AlexaFluor-555, Catalog number A-32773; |
|---|---|
| Validation | Pel-Freez Arkansas, Catalog number P60101-0, Sheep anti-tyrosine Hydroxylase, Lot number ajo1217p - validated by Western blot in rat caudate lysate (https://www.pelfreez-bio.com/wp-content/uploads/2014/07/74041-PDS-P60101-Tyrosine-Hydroxylase-Antibody-Sheep-Rev-02.pdf)<br>Merck, Catalog number MAB3422, Mouse anti-pan-histone, clone H11-4, Lot number 2842169 - validated by Western blot on Jurkat lysates (https://www.merckmillipore.com/DE/de/product/Anti-Histone-Antibody-clone-H11-4,MM_NF-MAB3422?ReferrerURL=https%3A%2F%2Fwww.google.com%2F&bd=1#anchor_Product%20Information)<br>Abcam, Catalog number ab13970, Chicken anti-GFP; Lot number GR236651-21 - validated by Western blot in whole cell lysates of mouse cardiomyocytes overexpressing a GFP plasmid (https://www.abcam.com/GFP-antibody-ab13970.html?gclsrc=aw.ds\|aw.ds&gclid=CjwKCAjwmeiiBhA6EiwA-uaeFVTDLhxCilj2b-NEfOFbj1nn-BBuPJCXCP4jcjC7mY5NkInrgp6TFBoCekUQAvD_BwE)<br>mouse anti-nucleophosmin B23 was validated by western blot in HeLa nuclear extract (Valdez, B.C., et al. Identification of the Nuclear and Nucleolar Localization Signals of the Protein p120. J. Biol. Chem. 269, 23776-23783 (1994)) |

## Eukaryotic cell lines

Policy information about cell lines

| Cell line source(s) | The mouse embryonic stem cells clone 46C derived from E14tg2a cells were provided by Dr. Domingos Henrique from Instituto de Medicina Molecular, Faculdade Medicina Lisboa, Lisbon, Portugal |
|---|---|
| Authentication | 46C E14tg2 mESCs are not listed in the ICLAC Register of Misidentified Cell Lines. The 46C E14tg2 mESC line was generated by insertion of an eGFP cassette under the control of the Sox1-promoter in E14 tg2 cells. Reads aligned with GFP sequence were identified in the GAM sequencing data from mESCs. Additionally, genome sequencing data from GAM mESC samples was mined for SNPs. Though GAM sequencing reads are sparsely distributed across the genome, there was a 64% overlap of GAM mESC SNPs with SNPs identified from the parental E14tg2 genome sequencing data (https://www.ncbi.nlm.nih.gov/sra?term=SRX389523). |
| Mycoplasma contamination | The cells were negative for Mycoplasma contamination. The Mycoplasma test was performed according to the manufacturer's instructions (AppliChem Cat#A3744,0020) |
| Commonly misidentified lines (See ICLAC register) | No commonly misidentified cell lines were used in the study (46C E14tg2 mESC are not listed in the ICLAC Register of misidentified cell lines https://iclac.org/databases/cross-contaminations/) |

## Animals and other organisms

Policy information about studies involving animals; ARRIVE guidelines recommended for reporting animal research

| Laboratory animals | All animals used in this study were from the species Mus musculus.<br>The following mouse strains were used:<br>- C57Bl/6NI (RRID: IMSR_CR:027; WT) for snATAC-seq experiments mice, adult male, ages 7 and 9 weeks;<br>C57BL/6NI mice were housed in a temperature controlled room at 22±2°C with humidity of 55±10% in individually ventilated cages with 12-hours light/12-hours dark cycles with free access to food and water ad libitum.<br>- C57Bl/6NCrl (RRID: IMSR_CR:027; WT) for GAM experiments, mice purchased from Charles River, adult male, 2-3 months old;<br>- TH-GFP mice (B6.Cg-Tg(TH-GFP)21-31/C57B6), adult male, 2-3 months old;<br>C57Bl/6NCrl and TH-GFP mice had access to food and water ad libitum and were kept on a 12 h:12 h day/night cycle at 20-23°C at |
|---|---|

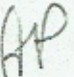

45% (+/-5%) humidity.
- Sox10::Cre-RCE::loxP-EGFP animals were obtained by crossing Sox10::Cre animals on a C57BL/6j genetic background with RCE::loxP-EGFP animals on a C57BL/6xCD1 mixed genetic background, both mouse lines available from The Jackson Laboratories. Adult male, sacrificed at P21. The mice received regular chew diet (either R70 diet or R34, Lantmännen Lantbruk, Sweden, or CRM-P, 801722, Special Diet Services). General housing parameters such as relative humidity, temperature, and ventilation follow the European convention for the protection of vertebrate animals used for experimental and other scientific purposes treaty ETS 123. Briefly, consistent relative air humidity of 50%, 22 °C and the air quality is controlled with the use of stand-alone air handling units supplemented with HEPA filtrated air. Monitoring of husbandry parameters is done using ScanClime (Scanbur) units. Water was provided by using a water bottle, which was changed weekly.
- Satb2flox/flox mice that carry the floxed exon 4 of the Satb2 gene have been generated by microinjection of embryonic stem cells clone Satb2_G07, JM8.N4 subline from KOMP repository, into blastocysts from C57Bl/6NCrl (RRID: IMSR_CR:027; WT) mice, purchased from Charles River. Adult males, sacrificed at 19 weeks old. All mice had access to food and water ad libitum and were kept on a 12 h:12 h day/night cycle at 22.5 °C (+/-1°C) at 55% (+/-10%) humidity.

| Wild animals | No wild animals were used in the study |

| Field-collected samples | No field-collected samples were used in the study |

| Ethics oversight | Experimental procedures involving C57Bl/6NI were approved by the regulations of local animal care committee (Landesamt für Gesundheit und Soziales, Berlin, Germany) and follow the Directive 2010/63/EU of the European Parliament on the protection of animals used for scientific purposes, organ preparation was done under license X9014/11. Experimental procedures involving C57Bl/6NCrl and TH-GFP animals were approved by the Imperial College London's Animal Welfare and Ethical Review Body. Experimental procedures involving Sox10::Cre-RCE::loxP-EGFP animals were performed following the European directive 2010/63/EU, local Swedish directive L150/SJVFS/2019:9, Saknr L150 and Karolinska Institutet complementary guidelines for procurement and use of laboratory animals, Dnr 1937/03-640. The procedures described were approved by the local committee for ethical experiments on laboratory animals in Sweden (Stockholms Norra Djurförsöksetiska nämnd), lic.nr. 130/15. Experimental procedures involving Satb2flox/flox mice were done according to the Austrian Animal Experimentation Ethics Board (Bundesministerium für Wissenschaft und Verkehr, Kommission für Tierversuchsangelegenheiten) |

Note that full information on the approval of the study protocol must also be provided in the manuscript.

## Flow Cytometry

### Plots

Confirm that:

☒ The axis labels state the marker and fluorochrome used (e.g. CD4-FITC).

☒ The axis scales are clearly visible. Include numbers along axes only for bottom left plot of group (a 'group' is an analysis of identical markers).

☒ All plots are contour plots with outliers or pseudocolor plots.

☒ A numerical value for number of cells or percentage (with statistics) is provided.

### Methodology

| Sample preparation | Male C57Bl/6NI (RRID: IMSR_CR:027; WT) mice, ages 7 and 9 weeks, were sacrificed by cervical dislocation. Brains were removed and the tissue containing the midbrain VTA was dissected from each hemisphere at room temperature and rapidly frozen on dry ice. Frozen tissue was homogenized in 500 microL 0.1X lysis buffer (10 mM Tris-HCl, pH 7.4, 10 mM NaCl, 3 mM MgCl2, 1% BSA, 0.01% Tween-20, 0.01% Nonidet P40 Substitute 0.001% Digitonin). Chilled wash buffer (500microL, 10 mM Tris-HCl, pH 7.4, 10 mM NaCl, 2 mM MgCl2, 1% BSA, 0.1% Tween-20) was added to the lysed cells, and the suspension was passed through 30 micrometer CellTrics strainers (Th Geyer, cat# 7648779). The final ~500 microL nuclei suspension was stained with DAPI (final concentration 0.03 microg/mL) for ~5 min. |

| Instrument | BD FACSAria III Flow Cytometer |

| Software | BD FACSDiva v 8.0.2 |

| Cell population abundance | Target population (intact nuclei) abundance was between 1-5% (see Extended Data Figure 4h-i) |

| Gating strategy | A first gate excluded debris in a FSC/SSC-plot and a consecutive, second gate in a DAPI-A/DAPI-H-plot was used to exclude doublets and nuclei with incomplete DNA content. |

☒ Tick this box to confirm that a figure exemplifying the gating strategy is provided in the Supplementary Information.

31.8.2021

