## [Peer Review File · Nature]

Manuscript Title: Cell-type specialization is encoded by specific chromatin topologies

Redactions – unpublished data

Reviewer Comments & Author Rebuttals

Reviewer Reports on the Initial Version:

Referee #1 (Remarks to the Author):

Winick-Ng and colleagues report the use of Genome Architecture Mapping (GAM) technologies to create maps of genome folding in dopaminergic neurons from the midbrain, pyramidal glutamatergic neurons from the hippocampus, and oligodendrocytes from the cortex. The authors create a new v2.0 of GAM – ImmunoGAM – this facilitates the important advance of isolating specific cell types in brain tissue and the creation of genome folding maps using very low cell numbers. The data generated are high quality and will be broadly valuable as a resource for understanding genome folding in the brain. The analysis of the data could be significantly improved to bring insights to light from such hard-fought elegant data. A systematic analysis first distinguishing and identifying A/B compartments, TADs, subTADs, and loops genome-wide, as well as statistical/probabilistic modeling to identify cell type-specific compartments, TADs, subTADs, and loops is so important for a paper such as this one. The biggest area that could immensely improve this paper is that the folding patterns are often conflated, or not called with cutting edge methods at the highest resolution, but if this was done the paper would markedly improve. There is the beginnings of a TAD analysis in Figures 2/3 – but it is only focused on very low resolution large TADs – and could benefit from calling the full sweep of subTADs. There was quite a bit of concern that the TAD melting phenotype is not a breakdown of large scale boundaries, but rather a missed detection of subTAD strengthening, and a large scale switch from B to A compartments in parallel with increased expression of the long genes. The long-range contact analysis in Figure 4 could benefit immensely from rigorous computational detection of bona fide corner dot loops and distinguishing such loops from non-specific long-range interactions within TADs/subTADS and A/B compartments. Can corner dots be seen and identified with GAM – and statistical analysis done to find cell type specific corner dots? Without these extra steps, it is difficult to interpret or understand Figure 4 and the significance of the findings, as many of the visual examples look to be compartment interactions, not corner dots. Transcription factor motifs will be found at the base of any possible structural feature, but the approach used in Figure 4 will not allow the reader to distinguish contacts in compartments vs. intra-TAD interactions, vs. nonspecific contacts between subTADs, vs. bona fide loops. Without this rigorous set of steps, the power and elegance of the GAM data and its power for discovery is under-utilised, but by restructuring the analysis, and connecting the features to those known in the 3D genome field (Rao et al, 2014) – this paper could greatly increase in impact.

Specific Comments

1. p. 5 – lines 89-91 – Normalization of Hi-C matrices is essential and the methodology used has a marked effect on the biological result. It was unclear what was meant by “informed about inter-locus 2D physical distances” – and unclear what normalization methods were used and how they influenced the raw data.
2. I had a particularly difficult time determining how many nuclei/cells went into the GAM heatmaps for each of the three cell types, despite searching Tables 1-2, Supp Fig 1, Main Fig 1, and the text. The statement was made on p. 5 ((627-1755 cells passed QC; Table 1), however this was confusing to me, as Table 1 seemed to be a summary of cells analyzed from published scRNAseq data, not immunoGAM.
3. The exact number of cells that go into a proximity ligation Hi-C-type experiment can yield dramatically different library complexity, and ultimately even slight differences in cell number can lead to replicates of the same biological condition showing differential TADs, compartments or loops, but it is not biological and rather due to technical differences in complexity. It is difficult to computationally correct for such differences, as they are nonlinear and dependent on multiple technical and biological factors. It would be useful to see the nuclei contributing to each cell type, and then a Supplementary Figure with the raw heatmaps, and how the heatmaps transform at each step in the normalization process.
4. It would also be useful to see how the GAM heatmaps change in appearance as a function of the number of pooled nuclei. The authors show isolates from two mice in Figure 1, which is a nice touch for reproducibility, but it was still not clear how many nuclei contributed, if they were different, and how the data were normalized.
5. In the 60 Mb maps in Figure 1D – are the long-range contacts illustrated A/B compartments? They are referred to in the text as ‘strong contact patches’. However, considering labeling them in the figure/text as A/B compartments, as long-range contacts or contact patches can add ambiguous language to a field already riddled with jargons and a definitions-crisis - and confuse a broader audience.
6. It was difficult to see the differential contacts around the Htr1a gene specific to the neural cell types where it is expressed compared to ES and OLG. It would be helpful to see a zoom in to the

locus, to explore the data for presence of TADs, subTADs, and loops, as the current vantage point and color scale is difficult to assess. The authors refer to a Schizophrenia SNP, however, I could not see it annotated in 1C – and including the SNP – with arrowheads at the loops it makes to Ntr1a and all differential looping/subTAD features would be helpful toward training the eye to critical cell type-specific structural features.

7. I was surprised that the paper transitioned after Figure 1 directly to scRNAseq. I would have been excited to see right away in Figure 1 a systematic, and genome-wide identification of A/B compartments, TADs, subTADs, loops, and interchromosomal interactions – and the identification of all such features which are differential across the four cell types (including ES cells). Given how valuable, precious, and unique the GAM data is in the three brain cell types, it would be of tremendously high impact to detect and statistically analysis all of the precise features. Perhaps gene expression could be brought in after the annotation of all the expected features and how they are difference across the four cell types.

8. For Figure 2 – the authors begin to call TADs. It would be of immense value to add the annotations of TADs and subTADs identified with the IS method directly on the maps in 1C/1D and 2A – as well as supplemental figures – so that the readers can assess how many TADs/subTADs are identified and get a sense for the false positive/negative rate of the domain calls.

9. Methods to identify TADs/subTADs give wildly divergent results, and the insulation score method is not sensitive to the identification of subTADs, nor does it provide a model to assess statistical changes in boundary strength across cell types. The authors could markedly increase the value of the hard work of creating the beautiful GAM data sets by using new methods such as TADTree, 3DNetMod, or TadPole to identify the full sweep of TADs/subTADs in these data. Moreover, a statistical test to specifically find robust cell type specific domains and boundaries is so critical for the claims made here. It appears in 2b that only boundaries, not domains, are reported as cell type specific by using a simple overlap method, but this will miss the large majority of the fluctuations, as most boundaries do not move from simple on to off states, but rather undergo more measured changes in strength.

10. In 1b – I could add up the percentages and only get to 65% - what is the class of the other boundaries up to 100%? It appears that ~2,000 domains are identified genome-wide. Typically in Hi-C data reveal 15,000-20,000 TADs/subTADs in a mammalian data set, including neurons, so it might be that a different TAD calling method will give higher sensitivity to truly leverage the data.

11. Once all TADs and subTADs that are cell type specific have been identified with a different method, the authors could show the insulation score plots at each subclass of boundary to provide

evidence that the boundary loses strength in the cell types in which it is not present. From what I could guess from the current 1C middle plot – it seems that all cell types have the same insulation strength despite cell type specific domain boundaries being plotted, which might mean in his plot that different boundaries are used for each color/cell-type. This is not as informative as examining insulation score at each cell type specific class of boundary and showing that some cells do not have insulation as predicted by the statistical method.

12. The authors state that with the IS method, only 50-65% of boundaries are the same in two replicates of the same condition. Did the authors only use the reproducible boundaries in the comparisons among cell types? The overlaps are extremely low, in particular because the average TAD length here is 1 Mb, indicating that they are the largest domains, and that subTADs were not detected. This is consistent with the domain calls in Figure 3c/d, which most definitely are large TADs, but do not include any nested subTADs. SubTADs are typically highly cell type specific, but Mb-scale or larger TADs are often highly cell type invariant. Given that the IS-identified domains appear to be Mb-scale TADs, I would have expected a higher overlap than 14% (all four cell types) or 8% (all brain cell types) given previously published work. My hope is that using a much more sensitive TAD/subTAD caller will help clear all of these quantitative issues up.

13. Figure 3: The Neurexin3 gene spanning 2 TADs is interesting. In exploring in detail the maps provided in Figure 3c/d: In ES cells, I see that the two TADs interact so that the 5' and 3' ends of *Nrxn3* can touch in ES cells. However, I see that the TAD boundary is still intact in DNs – to my eye I do not see melting at all of the boundary. If anything, the boundary appears to be getting stronger because in DNs the interactions between the two TADs goes away, thus the 5' and 3' ends of the gene can no longer interact. I do agree that the interaction strength throughout the second TAD is lower, but this seems to be a different structural phenotype than TAD boundary melting. The phenotype that I see is reduction in within-TAD interactions, and loss of nested subTADs – which could be quantified via some of the methods described above.

14. It is a nice touch to transition in p. 14 to look for TAD structural changes genome-wide. I still had quite a bit of difficulty in the maps seeing a TAD melting phenotype. If TAD melting is meant to convey the melting away of a boundary and the merging of two TADs in neurons compared to ES cells, I do not see this phenotype in any of the examples in Figure 3. However, I do agree that there are structural changes – it appears that long-range interactions within a defined TAD in ES cells are significantly weakened in DNs – which the authors interpret as chromatin decondensation. If the loss of interactions within TADs is indeed the phenotype, then it would be possible to still use the framework of the statistical test in Figure 3f, but for long-range contacts within large TADs around the > 400 kb long genes. Also, the extension of the analysis genome-wide would greatly benefit for quantifying and identifying subTADs.

15. I also note that in Figure 3H – I see more of gained/strengthening of subTAD boundaries in the brain cell type for all three gene examples shown, in addition to the loss of intra-TAD long-range interactions.

16. Figure 4- the paper would immensely benefit from using classic methods (Rao et al 2014) to find bona fide corner dot loops and distinguish such features from other forms of long-range chromatin interactions (non-specific intraTAD interactions, A/B compartments). It would be immensely valuable to see the degree to which corner dot loops change across cell types and to analyse such genes anchoring bona fide loops from “long-range interactions”. For some of the examples shown in Figure 4, the contacts look a lot like A/B compartments, and in this case the motifs enriched would simply be factors associated with B compartments (repressors, heterochromatin binding proteins) or A compartments (any activating transcription factor). This is interesting, but very different than the inquiry of loops and what factors/motifs contribute to the anchoring and bringing together of loops. Both analyses are of value, but should be restricted to the type of architectural feature.

17. Lines 413-515 emphasize that remarkably the interactions are between differentially expressed genes – this is consistent with the notion of compartments, thus emphasizing the importance of quantifying compartments separately from loops. It is quite well established that A-A- compartments come together over vast distances in a given cell type to bring together cell type specific active genes. It’s still important that this is happening in neurons as well, but connecting this observation to the existing literature and known features will only increase the quality of the paper.

18. I became quite tangled up by line 489 or so when the paper transitioned directly to compartments, as thus far, the analysis in Figure 4 also looked a lot like compartments. This only further emphasizes the importance of calling bona fide corner dot loops and using such loops to redo Figure 4 in the context of loops instead of general long-range interactions, of which it appears the methods used would conflate many types of contacts, including loops and compartments and general nonspecific chromatin interactions into the analysis, rendering it less powerful.

19. Zoomed in maps in Figure 4 to allow the reader to see bona fide loops and enhancer promoter interactions would be compelling and powerful toward the goal of ruling out that many of the contacts are compartments or non-specific.

20. In looking at the data in Figure 5e/f and the direct colocalization of visual domains with compartments, I became concerned that the TAD melting phenotype from Figure 3 was actually compartments switching from B to A during the activation of the long genes in neurons. This also emphasized to me that extra effort, although painstaking, to call compartments, TADs, subTADs, and loops separately would be immensely helpful in making the claims in the paper more powerful.

21. The observation that B compartments only strengthen around Olfactory genes even though the genes are already off is quite interesting.

Referee #2 (Remarks to the Author):

This study by Winick-Ng et al. developed immunoGAM by applying the GAM method to primary mouse brain tissues and coupled with immunolabeling assisted cell-type selection. The approach highlighted several key advantages of the GAM method such as requiring fewer input cells compared to Hi-C, reliable cell type identification through immunolabeling, and the possibility of further integration with spatial genomic methods. The authors then performed extensive computational analyses using this dataset generated from three mouse brain cell types. The study has made a number of interesting findings including the melting of TAD at long neuronal genes, chromatin interaction network of TF binding sites, and dynamic A/B compartments between mESC and brain cell types. Overall the study has significant novelty regarding the method, data resource, and biological findings. However, the current manuscript is missing a large number of statistical testing. According to Nature policy, p-value and the statistical test should be reported for most quantitative results. In certain cases (specific comments 6 & 9), the absence of rigorous statistical testing has raised questions about whether the biological findings were valid.

Specific comments

1. Although both the experimental methods and computational analyses are quite impressive, it was somewhat unclear what information provided by immunoGAM was unique and cannot be acquired using Hi-C based methods. Mouse Cre-lines that specifically label the three cell types were used in the study (TH-GFP for dopaminergic neurons; Sox10-Cre for oligodendrocytes and Satb2-flox for CA1 pyramidal neurons). So technically one could also perform tissue dissociation, FACS and perform cell-type-specific Hi-C experiments. It would be helpful if the authors can explicitly discuss the information that can be learned with immunoGAM but would be difficult with Hi-C.

2. Given that the microscopic information is available for each collected cell in immunoGAM, would it be possible to correlate the cellular spatial location (e.g. spatial gradient in CA1) with chromatin conformation signature at single-cell level? If possible, such analysis would provide unique spatially

resolved (at a cellular level) chromatin organization information that cannot be obtained by Hi-C based methods.

3. Can the authors explain why the mapping rate and unique mapping rate of immunoGAM data (Supplementary Table 2) are so low? Also, why was the percentage of non-unique mapping so high? On average 50%-70% of reads were mapped and ~half of the mapped reads were uniquely mapped.

4. This was probably discussed in previous publications. But the usage of DeepVent for random priming is intriguing since this polymerase is not a common choice for random priming DNA synthesis compared to Klenow exo-. Also it was intriguing the amplification was not performed with a longer reverse crosslinking incubation given these tissues were heavily crosslinked with 4% PFA perfusion.

5. Are there epigenomic signatures that distinguish melted TADs, in addition to the transcriptional activation described in the manuscript? Any notable DNA or histone methylation domain or MECP2 binding? It would be especially interesting if differential epigenomic signatures can be found between adjacent TADs but with different behaviors - e.g. the downstream, but not the upstream TAD at *Nrx3* was decondensed.

6. I have concerns about the resolution of TF binding motif analysis since the identification of candidate regulatory elements and TF binding motifs were performed in 50kb windows. 50kb is a very large space and can contain hundreds of putative enhancers. Permutation analysis should be performed in genomic regions containing actively expressed genes and/or containing similar numbers of putative enhancers to test whether the statistical significance of the enrichments.

7. The study reported drastically different TF regulatory networks in DN and PGN neurons (Fig. 4g). However, this analysis was vulnerable to potential biases. In particular, the denser network in PGN may be technically caused by more robust regulatory element calling, or due to the intrinsic property of PGN cells containing more active enhancers. Specifically, 1) The snATAC-seq dataset generated from PGN could be more sensitive than the LiDNase-seq database from DN. This hypothesis could be examined by comparing the number of open chromatin peaks or signal-to-noise ratio. 2) PGN cells may intrinsically contain more open chromatin sites and could lead to a denser TF network. It may be possible to set different statistical thresholds for network edges, based on sample-specific permutation, to account for the global difference in the calling of regulatory elements.

8. Are the genomic regions showing stronger B compartments in brain cell types associated with any epigenomic signatures such as polycomb or DNA methylation domains?

9. Line 608-609. Given the high drop-out rate of single-cell transcriptome detection, what statistical method was applied to make sure the identification of Vmn or Olfr expressing cells was reliable?

10. I have concerns about the reported difference between the cells with or without Vmn and/or Olfr was due to cellular-subtype stratification, instead of reflecting neuronal activation. The authors have made a profound claim that the subpopulation expressing Olfr or Vmn without an in vivo perturbation experiment. It may be possible to analyze the dataset in Hrvatin 2017 Nat. Neuroscience (mouse visual cortex dark/light stimulation experiment) to see if the conclusion holds? I am not really convinced that this section, which entirely relied on analyzing published transcriptome data, is very relevant to this chromatin conformation study, especially with the lack of in vivo perturbation experiment.

Referee #3 (Remarks to the Author):

In the paper by Winick-Ng, Kukalev, Harabula et al, entitled "Cell-type specialization in the brain is encoded by specific long-range chromatin topologies", the authors introduce ImmunoGAM, a protocol that combines immune selection of cells with multiplexed-GAM. With ImmunoGAM, the authors have managed to study the genome three-dimensional organization of specific mouse brain types at about 50Kb resolution. This can be done with a relative low number of cells (~1,000), which makes ImmunoGAM perfect for situations where the tissue of interest is heterogeneous in cell types and contains a limited number of the required cell types. Then, the authors analysed the resulting GAM interaction matrices to assess how different these cell types in brain are compared to mouse ESC. The authors found a series of interesting results, (i) TAD organization is to a large extent not-conserved and changes between cell types, (ii) identified unique borders in each cell type was associated to genes related to the cell specialization, (iii) most long and highly expressed genes were associated to changes in these TADs into the so-called "TAD-melting", (iv) genome organization per cell type can be used to generate a network of associated Transcription Factors relevant to the biology of each cell-type, and (v) genome compartmentalization was also found to be associated to repression of specific gene families important in diseases of the brain.

I found the work well conducted but still with limited technological novelty as both GAM and PRISM are published and multiplexGAM is BioRxived. Beyond this general comment, there are other concerns:

- There are many instances in the text where the authors justify statements by using “Visual inspection”. There are many published methods for dealing with the analysis of interaction matrices (arguably not all developed for but usable with GAM data) that could be used to quantify those statements. The authors should make an effort to provide statistical significance to many of their statements. This is important for many analyses but specially to GAM matrix comparison between the different studied cell types. Moreover, in all figures where GAM matrices are shown for visual comparison, it is important that the colour scale is exactly the same in all matrices.

- A very interesting observation is that TADs change to a large extent between cell types. This is against a somehow accepted pre-conception in the field that TADs are conserved between cell types. Therefore, the authors need to provide more evidences. Key questions are: does the Insulation Score analysis appropriate for GAM? How sensitive is to GAM data compared to Hi-C data? Would other methods result in the same observed variability? How does the resolution of the data affect the results? Using 1 bin to determine conservation is appropriate?

- The domain melting concept is interesting with respect its relationship with expression. However, I wonder whether the selection of the top 3% of the TAD melting sites is relevant to the expression levels in Fig 3g. What would happen if the authors select a similar number randomly? What if those were selected by low TAD melting scores?

- The authors find structural differences between genomes of different cell types. Then, they sub-set the genome for regions where those differences are larger (normally using a top X% of the differences rather than those differences that are statistically significant) and make GO analysis on those genomic regions. The GO terms show some interesting hits, but those have been (apparently) hand-picked by the authors from a larger list of other terms. Could the authors justify this selection?

- PRISM models show what the authors describe as “highly decondensed” regions when comparing models of mESC and DN cells. Why they say “decondensed”? A higher radius of gyration at the model’s resolution may not reflect chromatin decondensation. Additionally, could the authors measure “torsional stress” from the models to justify the statement relating this to expression of genes? Could the authors do additional models for long and expressed genes to find similar results? Just one region modelled may not be enough to make such statements.

- It would also be important to show results of highly expressed genes that are shorter than 400Kb. Is the “TAD melting” effect exclusive of long genes? Why? Does it depend on how TAD borders were determined?

- The authors used ATAC-seq data (accessible DNA) and motif search as Transcription Factor (TF) binding proxy. This is a good initial approximation, however, having a motif and open DNA does not necessarily mean that the TF factor is binding in the specific cell type. Could the authors assess, at least for a limited number of TFs, their TF networks using ChIP-Seq?

- In the discussion section, the authors state that “Our results strongly support that prediction of functional or disease states in a cell type could be inferred from chromatin architecture”. It is true that this work, and many others, show correlative findings, however, those correlations may not be sufficient to make accurate predictions unless proven.

- The header for the section “ImmunoGAM maps 3D genome architecture in specific mouse brain” misses “cell” somewhere.

Author Rebuttals to Initial Comments:

Referee #1 (Remarks to the Author):

Winick-Ng and colleagues report the use of Genome Architecture Mapping (GAM) technologies to create maps of genome folding in dopaminergic neurons from the midbrain, pyramidal glutamatergic neurons from the hippocampus, and oligodendrocytes from the cortex. The authors create a new v2.0 of GAM – ImmunoGAM – this facilitates the important advance of isolating specific cell types in brain tissue and the creation of genome folding maps using very low cell numbers. The data generated are high quality and will be broadly valuable as a resource for understanding genome folding in the brain.

We thank the reviewer for highlighting the value of our study. We are committed to sharing the GAM data and associated analyses as a resource. Further to the resources included in the previous submission (GEO file: GAM raw data, ESC scRNA-seq; SI tables: several items including melting scores, TAD borders, TF motif differential contact analysis), we have now added the following items to GEO: bulk mESC ATAC-seq and DN scATAC-seq fastq files, bigwig tracks, and peaks; and added additional SI tables with insulation scores, and compartment eigenvector values (SI Tables 4,5). We have released a public UCSC session (http://genome-euro.ucsc.edu/s/Kjmorris/Winick_Ng_2021_GAMbrainpublicsession) with all data utilized or produced in this study. Bioinformatic pipelines developed in this study will also be released, namely the MELTRON script to analyse domain melting (<https://github.com/pombo-lab/Meltron/>), and the trans-cis ratio script (https://github.com/pombo-lab/GAM_trans_cis_ratio/).

The analysis of the data could be significantly improved to bring insights to light from such hard-fought elegant data. A systematic analysis first distinguishing and identifying A/B compartments, TADs, subTADs, and loops genome-wide, as well as statistical/probabilistic modeling to identify cell type-specific compartments, TADs, subTADs, and loops is so important for a paper such as this one. The biggest area that could immensely improve this paper is that the folding patterns are often conflated, or not called with cutting edge methods at the highest resolution, but if this was done the paper would markedly improve. There is the beginnings of a TAD analysis in Figures 2/3 – but it is only focused on very low resolution large TADs – and could benefit from calling the full sweep of subTADs.

We appreciate the advice from the reviewer to improve and integrate several of the analyses of 3D genome topology. We have extensively reformatted the figures to present a more integrated perspective of cell-type specialization of 3D genome topology across length scales, and expanded many of the investigations of compartments, TADs, domain melting, differential contacts, and added new features, such as trans-cis contact ratios. With the extensive reorganisation of chromatin structures identified in specialized brain cells, we pave the way for novel avenues of investigation in 3D genomics and in the neurosciences.

We and others have shown that the definitions of TADs, sub-TADs and metaTADs are artificial simplifications of contact distributions in the population of cells, and are highly dependent on

method and pipeline for their classification (Soler-Vila et al., 2020). We took a conservative approach for the analysis of TAD borders, choosing the insulation score method which minimises differences between borders in GAM and Hi-C (Beagrie, Thieme, et al. BioRxiv 2021). In Rev.1 point 9, we present additional efforts to test alternative TAD calling methods developed for Hi-C data (i.e. TadPole, TadTree, Artamus, 3DNetMod), finding that only the insulation square method is appropriate to detect clear points of insulation in GAM data (which are visibly confirmed in Hi-C data).

There was quite a bit of concern that the TAD melting phenotype is not a breakdown of large scale boundaries, but rather a missed detection of subTAD strengthening, and a large scale switch from B to A compartments in parallel with increased expression of the long genes.

The reviewer's concerns helped us realise that the nomenclature 'TAD melting' was confusing. Our strategy to classify melting events is agnostic to the length scale(s) at which decondensation occurs. Although it is possible that some melting events involve loss of contacts between domains at a given layer, and this is of course interesting, it was not an obvious feature in the many loci we inspected. We renamed 'TAD melting' to 'domain melting' to more precisely reflect that we study melting regions of interest. In the revised manuscript, the analysis of melting is focused on long genes (>400kb), due to their broad interest in neuroscience. For example, the DN melting gene Dscam loses contacts at both local and long-range layers (Fig. 2e). We then show that the occurs most extensively at their 3' ends (Fig. 2f). We take this opportunity to explain that the terminology 'melting' was chosen in fairness to the much earlier, pre-Hi-C reports by imaging of large-scale chromatin decondensation seen in induced genes in Drosophila (Zuckerandl E. 1976 J Mol Evol).

We have significantly updated Figs. 2 and 3 to more comprehensively describe the domain melting phenomenon. We produced new scATAC-seq data for DNs in the mouse VTA which, along with published scATAC-seq in PGNs and OLGs, we used to investigate the association of domain melting with chromatin accessibility (Fig. 2i). We now report that melting occurs at genes with high levels of open chromatin which are not always the most highly expressed (e.g. Csm2 in DNs or Pard3 in both DNs and PGNs). Surprisingly, domain melting most frequently occurs in genes that belong to A compartment in both ESCs and the cell type of interest (e.g. Rbfox1 in PGNs; Fig. ED 5g), while only a subset undergo B to A transitions from mESCs to brain cells. To more directly understand domain melting, we investigated the Rbfox1 gene using single cell imaging by cryoFISH in PGNs and ESCs, uncovering changes in physical compaction and nuclear position that are prevalent in the population of PGNs (Fig. 3e-i, ED fig. 6f-g).

The long-range contact analysis in Figure 4 could benefit immensely from rigorous computational detection of bona fide corner dot loops and distinguishing such loops from non-specific long-range interactions within TADs/subTADs and A/B compartments. Can corner dots be seen and identified with GAM – and statistical analysis done to find cell type specific corner dots? Without these extra steps, it is difficult to interpret or understand Figure 4 and the

significance of the findings, as many of the visual examples look to be compartment interactions, not corner dots. Transcription factor motifs will be found at the base of any possible structural feature, but the approach used in Figure 4 will not allow the reader to distinguish contacts in compartments vs. intra-TAD interactions, vs. nonspecific contacts between subTADs, vs. bona fide loops. Without this rigorous set of steps, the power and elegance of the GAM data and its power for discovery is under-utilised, but by restructuring the analysis, and connecting the features to those known in the 3D genome field (Rao et al, 2014) – this paper could greatly increase in impact.

We have made extensive efforts to address this point. We have shown that Hi-C derived loops from mESCs and cortical PGNs (Bonev et al. 2017) are also strong contacts in GAM data from mESC and hippocampal PGNs (see below Rev.1 point 16). We also show that all the GAM-derived cell-type specific loops that were prioritised for further analyses in Fig. 4 are also locally strong contacts (ED Fig. 8a). Our strategy to study cell-type specific loops was deliberately designed to enable exploration of neighbourhoods of complex contacts, which may contain combinatorial TF binding information about cell-type specialisation. We utilized this strategy as we strongly believe that not only isolated loops, as typically reported in current mainstream studies, hold biological relevance. We find local contact synergies to be an unexplored aspect in the 4D nucleome field, which we hope our study will inspire others to pursue (see response to point 19, Rebuttal Fig. 11).

We have also significantly expanded the analyses in Fig. 4 (in response to all reviewers), by (a) collecting scATAC-seq in DNAs to substitute the published low-input DNase-seq, (b) adding the results of statistical tests to support our claims (e.g. ED Fig. 8f), and (c) integrating cell-type contact loops with known features (e.g. TADs, compartments; ED Fig. 8b-c). The current GAM datasets have robust properties at 50kb resolution, and future studies will certainly enable more detailed analyses of chromatin contacts from higher resolution GAM data in brain cells.

Specific Comments

1. p. 5 – lines 89-91 – Normalization of Hi-C matrices is essential and the methodology used has a marked effect on the biological result. It was unclear what was meant by “informed about inter-locus 2D physical distances” – and unclear what normalization methods were used and how they influenced the raw data.

In order to clarify the steps by which GAM data is processed into contact matrices, we have improved the section of the main text where NPMI is first introduced (lines 123-129). We have added a new supplemental ED Fig. 2a-c, to explicitly show how each step of the NPMI calculation effectively removes specific biases, and outperforms D' used in the first GAM manuscript (Beagrie et al. 2017). We discuss in more detail the effects of normalization in point 4 below, and agree that this is an important point that was missing in the original submission. We have also included the reference to Fiorillo et al. (2021, Nat Methods) to support the claim

that GAM data informs about inter-locus physical distances (see Fig. 3, therein), which is made in line 332 of the manuscript.

2. I had a particularly difficult time determining how many nuclei/cells went into the GAM heatmaps for each of the three cell types, despite searching Tables 1-2, Supp Fig 1, Main Fig 1, and the text. The statement was made on p. 5 (627-1755 cells passed QC; Table 1), however this was confusing to me, as Table 1 seemed to be a summary of cells analyzed from published scRNAseq data, not immunoGAM.

We thank the reviewer for pointing out the confusion. In our original submission, we had included the number of cells in Table 1, and unfortunately had made a mistake in three of them (for PGNs R1, PGNs R2 and OLGs). We agree that this is an important point, and we have now included this information in ED Fig. 1d.

3. The exact number of cells that go into a proximity ligation Hi-C-type experiment can yield dramatically different library complexity, and ultimately even slight differences in cell number can lead to replicates of the same biological condition showing differential TADs, compartments or loops, but it is not biological and rather due to technical differences in complexity. It is difficult to computationally correct for such differences, as they are nonlinear and dependent on multiple technical and biological factors. It would be useful to see the nuclei contributing to each cell type, and then a Supplementary Figure with the raw heatmaps, and how the heatmaps transform at each step in the normalization process.

*Please see point 4 for a detailed description of how the number of nuclei contribute to the GAM dataset collected for each cell type. This is an important point that we had discussed in Beagrie et al. 2017 (ED Fig. 5, therein), but we agree that it should also be explicit in the present manuscript. We have now included raw heatmaps and their transformations in ED Fig. 2a (for PGNs). We provide an extended version below for all cell types, together with additional normalization examples (**Rebuttal Fig. 1**).*

Rebuttal Figure 1. NPMI effectively normalises biases in GAM data. (a) NPMI corrects for locus co-segregations biases caused by small fluctuations in window detection frequency (WDF). Example shown for chr7:60,000,000-80,000,000. **(b)** Heatmaps of mean observed/expected bias in mESCs (GC content, mappability, WDF, DpnII restriction density) is shown for co-segregation, D-Prime (used in Beagrie et al. 2017), PMI and NPMI scaling. NPMI scaling results in the lowest absolute bias percentage for all tested categories (box plots on right). **(c)** Heatmaps of mean observed/expected bias and absolute bias percentage plots for PGNs R1.

4. It would also be useful to see how the GAM heatmaps change in appearance as a function of the number of pooled nuclei. The authors show isolates from two mice in Figure 1, which is a nice touch for reproducibility, but it was still not clear how many nuclei contributed, if they were different, and how the data were normalized.

GAM is a method that relies on statistical sampling of locus co-segregation to assess locus distances. A suitable working resolution of GAM datasets relies on good sampling of co-segregation events across all possible windows. GAM allows for optimisation of working resolution based on the size of the available datasets, and vice versa. We chose the resolution of 50kb in the present study, as it gives co-segregations of >99.5% of all possible pairs of mappable windows at least once within 5Mb (ED Fig. 1d). Each type of dataset may require different numbers of cells to effectively sample all possible pairs of windows at a given resolution. For example, the DNs required larger sample sizes to find all possible combinations of window pairs, because of their larger nuclear volume (e.g. average nuclear radius = 5.7 μ m vs 4.5 μ m in DNs and mESCs, respectively), considering the cryosection thickness was fixed at ~220-230 nm. NPMI normalizes for the number of nuclei, and always ranges between -1 and +1, independently of cell numbers. In this manner, GAM is independent of sequencing depth, but relies on minimum numbers of nuclei for robust sampling.

We showcase the effect of number of nuclei per GAM dataset in several ways:

a. By eroding datasets using random down-sampling the number of nuclear slices to 20, 40, 60, 80% of the full dataset, we find contact matrices that are fairly comparable with the full datasets already from 60% of the data onwards (Reb. Fig. 2).

Rebuttal Figure 2. Erosion of GAM datasets with random downsampling of the number of nuclei included. Erosion of GAM datasets were performed by random down-sampling to 20, 40, 60, or 80% of the full dataset. Example shown for chr7:60,000,000-80,000,000.

b. By choosing an appropriate resolution that gives good sampling across all datasets. We found that a working resolution of 50 kb enables cosegregation of all possible pairs of genomic regions at least once within 5Mb, in all datasets (**Reb. Fig. 3**). Some of the datasets produced can be analysed at higher resolutions, for example at 30kb for PGNs in either replicate. Merging of replicate datasets would have enabled higher resolutions, at the cost of averaging biological variability, which we chose not to do in the present study, but can be done in future studies.

Rebuttal Figure 3. Distribution of the detection frequency for intra-chromosomal pairs of genomic windows. Typically, every possible pair of 50kb genomic windows in each chromosome are found at least 6-10 times, and the highest co-segregation windows more than 20 times.

c. We performed correlations between eroded GAM datasets datasets (eroded 10 independent times each) to perform correlations of NPMI values, TADs and compartments to the full datasets. NPMI matrices have Spearman's correlations of >0.9 between 70% eroded data to the full dataset in all cell types (**Reb. Fig. 4a**). We additionally find high overlap with TAD borders and compartments from eroded datasets with as little as 40% of the original data, using either the number of compartment/TAD borders that overlap (~70-80% overlap, **Reb. Fig 4b**), or a measure of concordance previously utilized to compare TAD calling methods (~50-85% overlap, **Reb. Fig 4c**; similar to Soler-Vila, Cuscó et al. 2020) (from <https://github.com/stylianos-kampakis/supervisedPCA-Python/blob/master/Untitled.py>).

Rebuttal Figure 4. Correlations between eroded and full GAM datasets. (a) Spearman correlation of eroded NPMI matrices to the full GAM datasets. Dashed lines indicate a 90% correlation at ~70% of the eroded data for all cell types and replicates. (b) Percentage of TAD borders (*left*) and compartments (*right*) overlapping between eroded and full GAM datasets. (c) Measure of concordance for TAD borders (*left*) and compartments (*right*) overlapping between eroded and full GAM datasets.

5. In the 60 Mb maps in Figure 1D – are the long-range contacts illustrated A/B compartments? They are referred to in the text as ‘strong contact patches’. However, considering labeling them in the figure/text as A/B compartments, as long-range contacts or contact patches can add ambiguous language to a field already riddled with jargons and a definitions-crisis - and confuse a broader audience.

*We have added the compartment definition to matrices in Fig. 1d and ED Fig. 2e. Although the long-range ‘patches’ often occur between two compartment B regions (Figs 1d, 5e-f, ED Fig. 10c-d), not all compartments B form the long-range strong contact patches, as would be expected if patches were just typical compartments B. We have added statistical analyses to panel ED Fig. 10d, showing that B compartments containing *Vmn* or *Olfr* genes establish significantly stronger contacts than other B compartments not containing these genes. The patches are specific to brain cells. The long-range contact patches that we have reported here (and in our preprint Winick-Ng et al. 2020 BioRxiv) have already been confirmed elsewhere by Tan et al. (2021) at CpG-poor regions containing olfactory receptor genes in adult cortical neurons by single cell Hi-C.*

*We agree with the referee’s concerns in relation to excessive renaming of 3D topological features. In the updated manuscript text (see lines 156-159), we now refer to these ‘strong contact patches’ as clusters of strong long-range contacts, but we express our reservation against this choice, as we do not believe it improves clarity in the 3D genomics field, especially as the organisation of *Olfr* genes in olfactory neurons are extensively studied and of general interest in neurodegeneration.*

6. It was difficult to see the differential contacts around the *Htr1a* gene specific to the neural cell types where it is expressed compared to ES and OLG. It would be helpful to see a zoom in to the locus, to explore the data for presence of TADs, subTADs, and loops, as the current vantage point and color scale is difficult to assess. The authors refer to a Schizophrenia SNP, however, I could not see it annotated in 1C – and including the SNP – with arrowheads at the loops it makes to *Ntr1a* and all differential looping/subTAD features would be helpful toward training the eye to critical cell type-specific structural features.

*We agree that the *Htr1* region was not an ideal first example to highlight differences in topology between cell types, especially as this first example aimed at gently introducing the reader to our study of more complex genome-wide features of 3D genome topology. We have replaced panel 1c in Fig. 1, with a smaller 3Mb region centered on the well-studied clustered protocadherin (*Pcdh*) locus, where it should now be easier to appreciate local changes in 3D genome topology between cell types. To help guide the reader, we have annotated the insulation scores, expression and open chromatin data, and highlighted the changes between cell types with dashed lines. The changes in contacts have not been reported in the literature, but the strengthening of contacts that we now report between *Pcdh* clusters can also be detected by mining published Hi-C data from cortical PGNs (Bonev et al. 2017) as shown in **Reb. Fig. 5.***

Rebuttal Figure 5. Hi-C interaction matrices for the *Pcdh* locus (Bonev et al. 2017). Hi-C matrices for ESCs (left), NPCs (middle) and cortical PGNs (right) for the *Pcdh* locus (top and bottom; chr18: 36,000,000-39,000,000). Increased contacts between the alpha and beta/gamma loci can be observed for NPCs and PGNs. Maps downloaded at <https://bonevlab.com/resources/>.

7. I was surprised that the paper transitioned after Figure 1 directly to scRNAseq. I would have been excited to see right away in Figure 1 a systematic, and genome-wide identification of A/B compartments, TADs, subTADs, loops, and interchromosomal interactions – and the identification of all such features which are differential across the four cell types (including ES cells). Given how valuable, precious, and unique the GAM data is in the three brain cell types, it would be of tremendously high impact to detect and statistically analysis all of the precise features. Perhaps gene expression could be brought in after the annotation of all the expected features and how they are differences across the four cell types.

We thank the reviewer for this request which we feel has greatly improved the flow of the manuscript. We have reorganized the manuscript extensively to present all features (e.g. compartments, insulation score maps) in parallel, and relate them throughout as much as possible. We have updated Fig. 1b-d to highlight orthogonal data resources that we use throughout the manuscript, and make available by directing the reader to these resources early in the manuscript (lines 131-136). We include expression, chromatin accessibility, TADs and compartments to the examples in Fig. 1c-d. We have also integrated compartment changes with domain melting, and differential contacts with compartments and TAD borders (ED Figs. 5g, 8b-c, respectively). All data produced directly, or through downstream processing, that is used for any of the analyses or displays related to the manuscript is made available (e.g. SI tables, GEO, github).

8. For Figure 2 – the authors begin to call TADs. It would be of immense value to add the annotations of TADs and subTADs identified with the IS method directly on the maps in 1C/1D and 2A – as well as supplemental figures – so that the readers can assess how many TADs/subTADs are identified and get a sense for the false positive/negative rate of the domain calls.

We have added insulation score heatmaps (100-1000kb insulation squares) to figures (Figs. 1c, 2a, 3a-b, ED Figs. 2d, 4a, 6a,b), where relevant and the TAD borders which were called at insulation square 500 to Fig. 2a.

9. Methods to identify TADs/subTADs give wildly divergent results, and the insulation score method is not sensitive to the identification of subTADs, nor does it provide a model to assess statistical changes in boundary strength across cell types. The authors could markedly increase the value of the hard work of creating the beautiful GAM data sets by using new methods such as TADTree, 3DNetMod, or TadPole to identify the full sweep of TADs/subTADs in these data. Moreover, a statistical test to specifically find robust cell type specific domains and boundaries is so critical for the claims made here. It appears in 2b that only boundaries, not domains, are reported as cell type specific by using a simple overlap method, but this will miss the large majority of the fluctuations, as most boundaries do not move from simple on to off states, but rather undergo more measured changes in strength.

a) Further evidence to support the claim of cell-type specific borders. We agree that methods to identify TADs give wildly divergent results, and for this reason we have chosen a deliberately agnostic approach to report insulation as a continuum at multiple length scales (100-1000kb; Figs. 1c, 2a, 2d,e, and 3a,b). We note that GAM data naturally captures increased contacts between TADs, as it more directly senses distances and is not restricted to capturing ligation events (see Beagrie, Thieme et al. 2021 for a more extensive comparison between domains and compartments between GAM and Hi-C data). Only in Fig. 2b,c, we have chosen to report on borders obtained from a single length (500kb) to exemplify underlying biology that accompanies the extensive changes in insulation. We chose the 500kb insulation square length as it results in similar numbers of TADs reported in other studies by Hi-C (2200 to 2500 in our four cell types, compared e.g. with ~2300 TADs in cortical neurons obtained with a high resolution Hi-C dataset, Bonev et al. 2017). We have strengthened the claim that many borders are cell-type specific by investigating the differences in insulation between the cell type where they are found and all the other cell types is highly significant (Fig. 2b lower row; Mann-Whitney test, p-value < 1.8e-07 for all comparisons). We also show that TAD border insulation in common borders is significantly increased in neuronal cell types relative to ESCs (ED Fig. 4d, p-value < 8.6e-10). We added SI Table 4 which provides all insulation scores, as a resource for future explorations.

*As an additional reassurance, we measured the fraction of cell-type specific borders using additional insulation squares and found a similar proportion (4-6, 5-6, 7-9 % for 300, 400, 600kb insulation squares, respectively) compared with 5-8 % at 500kb. We illustrate that similar proportions of cell-type specific TADs are discovered irrespectively of insulation square in **Reb. Fig. 6** (for 500 and 600kb). The percentages shown here are slightly different than reported in Fig. 2b (7-10% at 500kb), only because they were done with a faster pipeline that calculates absolute overlaps, instead of the more conservative approach chosen the manuscript that allows*

1 bin separation (i.e. 50 kb) between 150kb borders; we have not added these additional plots to the manuscript for simplicity, but can do so if preferred.

TAD borders called at insulation square 500

TAD borders called at insulation square 600

Rebuttal Figure 6. Comparison of TAD borders for different insulation square sizes. UpSet plots showing the absolute overlap of TAD boundaries called at 500kb (*left*) or 600kb (*right*) insulation square size. The pattern of overlaps is similar for both lengths, with 600kb having a slightly higher percentage of cell-type specific boundaries (5-8% and 7-9% for 500 and 600kb, respectively).

b) TAD calling results with other methods. We tested three new TAD calling methods on GAM data, which use different principles for boundary detection including *TadPole* (hierarchical clustering), *Armatus* (linear score) and *TadTree* (statistical modeling). We note that none of these methods, nor other 20 methods, could detect more than 2500 TADs at 50kb resolution in GM12878 cells, as tested in (Soler-Vila et al., 2020). For example, *TADpole* reports ~500 TADs).

We found that these methods are not appropriate to detect clear points of insulation in GAM data (which are visibly confirmed in Hi-C), either because of how they handle the data, because of assuming the discrete values reflect ligation events, or because GAM data captures more inter-TAD contacts than Hi-C (see **Rebuttal Fig. 7** for chr 19, mESC). We also tested a fourth method, *3DNetMod*, but its current implementation is incompatible with GAM data (range of signal in NPMI is -1 to +1). We feel that the concept of TADs is still evolving, and there is no gold standard approach to detect them experimentally, or to call them computationally from bulk data. A detailed technical investigation of TAD calling would be out of the scope of the present study.

Rebuttal Figure 7. TAD calling in GAM and Hi-C data with different methods. TADs were called for chr19 using Hi-C from mESCs (40kb resolution; Dixon et al. 2012) and GAM mESCs (40kb resolution; Beagrie, Thieme et al. 2021, BioRxiv) with the Insulation square method (Crane et al. 2015 Nature), TADPole (Soler-Vila et al. 2020 Nucleic Acids Research), Armatus (Filippova et al. 2014, Algorithms for Molecular Biology) and TadTree (Weinreb and Raphael 2015 Bioinformatics). Each method gave expected outputs for Hi-C input data although widely variable. For GAM data, only the Insulation square method produced TADs which visibly reflect the points of insulation between domains.

10. In 1b – I could add up the percentages and only get to 65% - what is the class of the other boundaries up to 100%? It appears that ~2,000 domains are identified genome-wide. Typically in Hi-C data reveal 15,000-20,000 TADs/subTADs in a mammalian data set, including neurons, so it might be that a different TAD calling method will give higher sensitivity to truly leverage the data.

We apologise that the figure and legend were not explicit to avoid this confusion. The UpSet plot in Fig. 2b reports only the seven most abundant combinations. We have made this feature more explicit in Fig. 2b and include all combinations in ED Fig. 4c, as in the previous version. As reported in point 9, most TAD callers detect up to 2500 TAD borders, similar to the number we find in this study, and in line with published reports in high-resolution Hi-C from neuronal tissues (Bonev et al. 2017, Soler-Vila et al, 2020). We are aware that 3DNetMod also reports ~2800 layer-1 TAD borders (in human hESCs, a recent BioRxiv report Emerson et al. 2021). Unfortunately, the 3DNetMod pipeline cannot yet accommodate GAM data, and we agree it will be exciting to explore subTADs in future studies utilizing the resources produced here.

11. Once all TADs and subTADs that are cell type specific have been identified with a different method, the authors could show the insulation score plots at each subclass of boundary to provide evidence that the boundary loses strength in the cell types in which it is not present. From what I could guess from the current 1C middle plot – it seems that all cell types have the same insulation strength despite cell type specific domain boundaries being plotted, which might mean in his plot that different boundaries are used for each color/cell-type. This is not as informative as examining insulation score at each cell type specific class of boundary and showing that some cells do not have insulation as predicted by the statistical method.

We appreciate this important advice, and now provide evidence that cell-type specific boundaries have high insulation scores in the cell types in which they are not found (Fig. 2b - lower row of graphs) with accompanying statistical analyses.

12. The authors state that with the IS method, only 50-65% of boundaries are the same in two replicates of the same condition. Did the authors only use the reproducible boundaries in the comparisons among cell types? The overlaps are extremely low, in particular because the average TAD length here is 1 Mb, indicating that they are the largest domains, and that subTADs were not detected. This is consistent with the domain calls in Figure 3c/d, which most definitely are large TADs, but do not include any nested subTADs. SubTADs are typically highly cell type specific, but Mb-scale or larger TADs are often highly cell type invariant. Given that the IS-identified domains appear to be Mb-scale TADs, I would have expected a higher overlap than 14% (all four cell types) or 8% (all brain cell types) given previously published work. My hope is that using a much more sensitive TAD/subTAD caller will help clear all of these quantitative issues up.

We understand the concerns of the reviewer, and it would have been easier to either merge the replicate data or to take only conserved features. However, it is not known whether boundaries conserved between animals are the only true boundaries, or whether their differences are not biologically relevant. We believe that in the 4D Nucleome field we need to increasingly consider biological diversity especially to better understand the contribution of 3D genome organisation in complex diseases, where a sporadic disease has low penetrance in the population. For example, it was especially interesting and important in relation to the melting of long genes, which we discover in both replicates, but not always to the same extent in each gene.

To achieve the most robust insulation score measures from single animals in Fig. 2b, we chose replicate 1, as these datasets have slightly better quality, defined by statistical tests of genome sampling (Extended Data Fig. 1, Supplemental Table 1). As shown in answer to Rev.1 point 4, the TAD borders reported at 500kb insulation are robust as they have 75-85% concordance with the eroded dataset containing only 40% of the data (i.e. with median ~340 of 850 nuclear slices). Due to length constraints, we have chosen not to include the important tests shown in Reb. Fig. 4 in the manuscript, but would be happy to do so, if requested. To take advantage of having two separate replicates from different mice, we generally chose to perform independent analyses in the two replicates and compare them to verify our claims (e.g. ED Figs. 2d,e, 4a,b,e, 5a,c-e,g-i, 6a,b,i-k, 8j, 9a,b,c,e,f).

As an additional reassurance, we show below comparisons between TAD borders done separately for DN/PGN R1 and R2, using the simplified overlap method described in point 9. In both cases, cell-type specific borders occur at a rate of 5-8% for each cell type, and 16-17% conserved between all cell-types.

TAD borders called at insulation square 500- PGN and DN R1

TAD borders called at insulation square 500- PGN and DN R2

Rebuttal Figure 8. Comparison of TAD borders for replicate GAM datasets. UpSet plots showing the absolute overlap of TAD boundaries called with PGN and DN replicate 1 (R1; left) or replicate 2 (R2; right). The pattern of overlaps is similar for both replicates, with both replicates containing ~5-8% of cell-type specific and 16-71% conserved boundaries.

We respectfully disagree with the commonly accepted hypothesis that Mb-scale TADs are highly cell-type invariant (e.g. reviewed in Eres & Gilad 2021 Trends in Genetics). Most studies so far have used dividing cells or mixed cell populations, where 20-30% variation in TAD borders is often detected (Dixon et al. 2012, 2015), including our own work using Hi-C from in vitro differentiated DNs (Fraser et al. 2015) and from FACS purified neurons (Bonev et al. 2017). Single-cell Hi-C has shown that TAD borders vary during the cell cycle (Nagano et al. 2017), while super-resolution chromatin tracing has shown that TAD boundaries are highly variable between single cells and cell types, with boundaries emerging as a property of population averaging (Bintu et al. 2018). We thus hypothesized that mapping chromatin contacts in pure

populations of non-dividing cell types, especially without tissue disruption would reveal cell-type specificity in domain organisation, which was a main motivation for the present study.

13. Figure 3: The Neurexin3 gene spanning 2 TADs is interesting. In exploring in detail the maps provided in Figure 3c/d: In ES cells, I see that the two TADs interact so that the 5' and 3' ends of Nrnx3 can touch in ES cells. However, I see that the TAD boundary is still intact in DNs – to my eye I do not see melting at all of the boundary.

If anything, the boundary appears to be getting stronger because in DNs the interactions between the two TADs goes away, thus the 5' and 3' ends of the gene can no longer interact. I do agree that the interaction strength throughout the second TAD is lower, but this seems to be a different structural phenotype than TAD boundary melting. The phenotype that I see is reduction in within-TAD interactions, and loss of nested subTADs – which could be quantified via some of the methods described above.

We apologize that it was not clear that TAD melting was not specifically studied at, or a feature of, TAD boundaries. Seeing the alternative definition understood by the reviewer in point 14, that 'TAD melting' is the loss of interactions within TADs, we realised that 'TAD melting' is a confusing term to describe the decondensation of any genomic region of interest. Therefore, we have renamed the phenomenon as "domain melting", which in the present study we applied only to long genes. The MELTRON pipeline is deliberately agnostic to the length-scale at which the decrease in insulation scores occurs (100-1000kb), whether by expansion of a boundary, the loss of contacts within a TAD, or loss of contacts between TADs. We have made efforts to more clearly describe in the text, and in the figures, what is measured and defined as 'domain melting', with the independent validation by single cell imaging in PGNs (Fig. 3f-i, Rbfox1).

The insightful comments from the reviewer about 5' and 3' contacts inspired several new and interesting analyses. First, we show that loss of insulation can occur at highly expressed genes at both the 5' and 3' regions, especially at the latter (Fig. 2f). In the FISH experiments, we specifically designed probes that targeted the 5' and 3' regions separately and indeed discovered that the distance between the two ends of Rbfox1 typically doubles between mESCs and PGNs, sometimes being >1micrometer (Fig. 3e,h-i).

14. It is a nice touch to transition in p. 14 to look for TAD structural changes genome-wide. I still had quite a bit of difficulty in the maps seeing a TAD melting phenotype. If TAD melting is meant to convey the melting away of a boundary and the merging of two TADs in neurons compared to ES cells, I do not see this phenotype in any of the examples in Figure 3. However, I do agree that there are structural changes – it appears that long-range interactions within a defined TAD in ES cells are significantly weakened in DNs – which the authors interpret at chromatin decondensation. If the loss of interactions within TADs is indeed the phenotype, then it would be possible to still use the framework of the statistical test in Figure 3f, but for long-range contacts within large TADs around the > 400 kb long genes. Also, the extension of the analysis genome-wide would greatly benefit for quantifying and identifying subTADs.

Please see comments to point 13 about improvements and clarification about domain melting. We feel that the expansion of the melting analysis genome-wide is beyond the scope of the current study, and may be more suited in a study focused on higher-depth GAM dataset for which many orthogonal datasets are available, namely chromatin occupancy of TFs and other chromatin regulators, distance mapping to lamina, nucleoli and speckles, etc. We are actively exploring these questions in other systems, and would not have enough space to include and digest such extensive additional analyses.

15. I also note that in Figure 3H – I see more of gained/strengthening of subTAD boundaries in the brain cell type for all three gene examples shown, in addition to the loss of intra-TAD long-range interactions.

Please see comments to points 9, 12 and 13.

16. Figure 4- the paper would immensely benefit from using classic methods (Rao et al 2014) to find bona fide corner dot loops and distinguish such features from other forms of long-range chromatin interactions (non-specific intraTAD interactions, A/B compartments). It would be immensely valuable to see the degree to which corner dot loops change across cell types and to analyse such genes anchoring bona fide loops from “long-range interactions”. For some of the examples shown in Figure 4, the contacts look a lot like A/B compartments, and in this case the motifs enriched would simply be factors associated with B compartments (repressors, heterochromatin binding proteins) or A compartments (any activating transcription factor). This is interesting, but very different than the inquiry of loops and what factors/motifs contribute to the anchoring and bringing together of loops. Both analyses are of value, but should be restricted to the type of architectural feature.

Discovering interesting and biologically relevant contacts is a core challenge in 3D genomics, and one at the core of this study as highlighted in Fig. 4. In Hi-C data, CTCF-mediated corner dots are a striking feature of contact maps, which are successfully identified by loop-calling approaches such as Peakachu (Salameh et al. 2020) and Rao et al. 2014. In GAM data, due to its increased complexity capturing physical distances as a continuum and more complex interactions involving transcription factors, RNA polymerase II and active genes (see also Beagrie, Thieme et al. 2021 BioRxiv), corner dots are not a most obvious feature of contact maps.

*Nevertheless, as reassurance, it was interesting to take the published Hi-C-derived loops from cortex PGNs and ESCs using Peakachu, and find that they are also strong contacts in GAM data (**Rebuttal Fig. 9**).*

Rebuttal Figure 9. Aggregate maps of loops found in Hi-C can be confirmed in GAM data. (a) Loops for mESCs and PGNs, from published Hi-C datasets (from Bonev et al. 2017) and called with the Peakachu tool, were obtained from Salameh et al. 2020. Aggregate maps of average Z-scores produced for each contact and a 200kb radius (4 genomic bins), show a

strong enrichment at the contact, immediate surrounding bins, and towards the diagonal. (b) For each set of loops, chromosome- and distance-matched contacts were randomly sampled three times from the genome-wide distribution (one exemplar is shown for each cell type).

As we were most interested in exploring the activity of neuronal specific TFs and their roles in cell-type specific contacts, we focused our efforts on devising a new approach to discover which combinations of putative TF binding sites are present in specific contacts, irrespectively of whether they exist in isolation or embedded with other differential contacts. Prior work from Bonev et al. (2017) took a straightforward approach to report on the association of TFs with loops, without classic loop calling methods, by considering any window containing a given TF of interest (e.g. Pax6, Neurod2, Tbr1) and reported them as bona fide loops when plotted in an aggregate plot (Bonev et al. 2017, Fig. 6 therein).

We wanted to develop a more refined approach to extract TF regulatory activity specifically at the most highly differential contacts between the two neuronal types, and in this manner explore whether they bring together neuronal-specific genes. To honour the reviewer's concern and bridge the two approaches, we now report in the manuscript that the most cell-type specific contacts are also locally strong when visualised as aggregate maps (ED Fig. 8a). We have also compared the cell-type contacts with known features (i.e. TADs, compartments), and found that the genomic windows in differential contacts are mostly not overlapping with TAD borders (and therefore are not corner dots), and are enriched for contacts between A-A compartments, as expected as we select them based on the presence of accessible chromatin (ED Fig. 8b-c).

17. Lines 413-515 emphasize that remarkably the interactions are between differentially expressed genes – this is consistent with the notion of compartments, thus emphasizing the importance of quantifying compartments separately from loops. It is quite well established that A-A- compartments come together over vast distances in a given cell type to bring together cell type specific active genes. It's still important that this is happening in neurons as well, but connecting this observation to the existing literature and known features will only increase the quality of the paper.

We understood from this comment that the referee is concerned about whether the results in Fig. 4 are a trivial consequence of A-A contacts. These analyses have now been strengthened in

several ways including the use of permutation tests (see Rev. 2 point 6), to reassure that the associations of specific differential contacts with expressed genes in both windows (Fig. 4c) are not random nor simply explained by A-A contacts. Following the reviewer's concern, we have additionally included a permutation test to ask whether compartment A membership was sufficient to discover the striking associations with highly specialized neuronal genes. It is true, as expected and pointed out by the reviewer, that contacts with A membership are naturally more likely to contain active genes (25-28% in both contacting windows) compared with randomly permuted expressed genes (mean= 15%). However, the contacts that our approach prioritises for further analyses, through the presence of open chromatin containing putative binding sites of differentially expressed TFs, allowed us to achieve highly statistically significant enrichments for the presence of active genes in both windows (up to 40-50%). These important statistical analyses are included in ED Fig. 8f of the manuscript, and below (**Reb. Fig. 10**).

TF-containing contacts are enriched for gene-gene interactions above permuted genes and A-A contacts

Rebuttal Figure 10. Overlap of TF-pair containing contacts with randomly permuted expressed genes. Overlap of TF-pair containing contacts with 1000 random circular permutations of PGN and DN expressed gene regions. Observed enrichments of contacts with genes in both windows are significantly higher than the expected distribution (**empirical p < 0.001 for all observed values tested; Z-test). The enrichments were also seen for all contacts between A compartment windows, though to a smaller level than for the TF-pair containing contacts.

18. I became quite tangled up by line 489 or so when the paper transitioned directly to compartments, as thus far, the analysis in Figure 4 also looked a lot like compartments. This only further emphasizes the importance of calling bona fide corner dot loops and using such loops to redo Figure 4 in the context of loops instead of general long-range interactions, of which it appears the methods used would conflate many types of contacts, including loops and compartments and general nonspecific chromatin interactions into the analysis, rendering it less powerful.

Following the earlier advice from the reviewer, we now introduced compartment calls in Fig. 1d and where relevant throughout the manuscript, to help relate the main observations to compartments throughout the paper. We believe that the improvements in the flow of the manuscript should resolve this valid concern of the reviewer.

19. Zoomed in maps in Figure 4 to allow the reader to see bona fide loops and enhancer promoter interactions would be compelling and powerful toward the goal of ruling out that many of the contacts are compartments or non-specific.

All analyses in Fig. 4 relied on the presence of regulatory regions captured by ATAC-seq in differential contacts, so are focused on the analyses of putative enhancer-enhancer contacts, enhancer-gene, or gene-contacts. We have now included a new panel in ED Fig. 8a using standard loop aggregate maps, showing that the family of contacts prioritised for further analyses are locally strong, as in loops called by Hi-C. We have also strengthened the statistical analyses of different contacts in Fig. 4 to show specifically that choosing contacts that have specific TF combinations significantly enriches the presence of expressed genes in both contacting windows (i.e. at the bases of the loop) over the A-A contact expectation (ED. Fig. 8f; see also point 17).

We have improved the zoomed in panels containing the *Egr1* maps of differential contacts (Fig. 4f), to include the pseudo-bulk ATAC-seq and putative TF binding sites from PGNs and DNs. By comparing the two, it is clear that the increased contacts in PGNs are accompanied by local reorganisation of open chromatin regions to specifically include putative binding sites for PGN-upregulated TFs (*Egr1*, *CTCF*, *Neurod1/2*), which are exclusively found within PGN ATAC-seq peaks in this region. We include an equivalent display here for the *Shisa6-Dlg4* region shown in Fig. 4d (**Rebuttal Fig. 11**). This is another remarkable example where a cluster of strong contacts (separated by ~4.5 Mb) are specifically found in PGNs, and which coincide with the appearance of high density of open chromatin regions in PGNs enriched for the *Neurod* group of putative binding motifs.

Rebuttal Figure 11. Complex contacts that contain accessible TF-motifs connect differentially expressed genes across Mb distances. Differential Z-Score matrix showing PGN-upregulated genes that form contacts across a ~4.5 Mb linear genomic distance (pink box; chr11:65,400,000-70,400,000). Upper right inset shows PGN-significant differential contacts containing the *Neurod* group (contacts are shown in pink). PGN and DN ATAC-seq peaks, and TF motifs found within ATAC-seq peaks are shown below the matrices. For the interacting region, TF motifs are highly abundant in PGNs, and depleted in DNs (dashed pink boxes below matrices). Genes highlighted in blue are upregulated in PGNs.

We were excited to discover such richness of TF binding and contact complexity within the PGN-DN differential contacts, that goes beyond the discovery of isolated contacts in the mainstream loop analyses. In this study, we take the first slice out of this complexity driven by, or containing, specific TF combinations. Our analyses argue for the investigation of contacts irrespectively of whether they are seen as isolated dots in contact matrices, as TFs are known to cooperate, and it is perhaps expected that at long genomic distances the emergence of specific contact patterns is likely to rely on increased density of local features. Out of 218 expressed TFs in the two cell types, we prioritised investigating only the 50 TFs that were most differentially expressed in PGNs or DNs, and out of the possible 1275 feature-pair combinations in contact loops, we investigated further only the top 20 TF-feature pairs. We believe that there remains a wealth of exciting further discoveries in TF regulatory contributions to complex topologies that will require extensive further developments.

20. In looking at the data in Figure 5e/f and the direct colocalization of visual domains with compartments, I became concerned that the TAD melting phenotype from Figure 3 was actually compartments switching from B to A during the activation of the long genes in neurons. This also emphasized to me that extra effort, although painstaking, to call compartments, TADs, subTADs, and loops separately would be immensely helpful in making the claims in the paper more powerful.

*This was an especially important comment that led to unexpected results. We have now extensively explored the relationship between melting and compartment changes, and remarkably we find that most genes that are melting in OLGs, DNs or PGNs are already in compartment A in mESCs. Nevertheless, some interesting genes escape this general pattern and indeed undergo B to A transitions between mESC and brain cells, such as *Nrxn3*, *Dscam* (DNs), *Nlgn1* (PGNs) and *Magi2* (OLGs). These interesting results are shown in ED Fig. 5g. These observations show that many melting genes are already in an open conformation that becomes even more decondensed upon full activation, reminiscent of the phenomena of polytene chromosome puffing of interbands upon e.g. heat-shock activation (Ashburner & Bonner 1979, Lis et al. 1983). Remarkably, we also find that *Rbfox1* which is predominantly associated with the nucleolus in mESCs, is actually embedded in an A compartment, a result which we believe will inspire further work from the nucleolus expert community.*

21. The observation that B compartments only strengthen around Olfactory genes even though the genes are already off is quite interesting.

We thank the reviewer for this comment.

Referee #2 (Remarks to the Author):

This study by Winick-Ng et al. developed immunoGAM by applying the GAM method to primary mouse brain tissues and coupled with immunolabeling assisted cell-type selection. The approach highlighted several key advantages of the GAM method such as requiring fewer input cells compared to Hi-C, reliable cell type identification through immunolabeling, and the possibility of further integration with spatial genomic methods. The authors then performed extensive computational analyses using this dataset generated from three mouse brain cell types. The study has made a number of interesting findings including the melting of TAD at long neuronal genes, chromatin interaction network of TF binding sites, and dynamic A/B compartments between mESC and brain cell types. Overall the study has significant novelty regarding the method, data resource, and biological findings. However, the current manuscript is missing a large number of statistical testing. According to Nature policy, p-value and the statistical test should be reported for most quantitative results. In certain cases (specific comments 6 & 9), the absence of rigorous statistical testing has raised questions about whether the biological findings were valid.

We deeply appreciate the concerns of the reviewer and we have strengthened the statistical analyses throughout the manuscript (Figs. 2b-c,f,h-i, 3d,g-h,k-l; ED Figs. 4d, 5c-f,h-i, 6i-m, 8f, 10d). Specifically for comment 6, we performed permutation analyses for gene-gene contacts (Extended Data Fig. 8f), which show strikingly significant enrichments, while more clearly explaining our strategy for the differential contact analyses, and discovery of contacts that are informative of cell type. For comment 9, we chose to remove the analyses of Olfr gene escapees from single cell data due to the expansion of our analyses of melting and due to the length of the manuscript. We had initially included those results because we find them very exciting, but agree with the reviewer that a deeper study with perturbations will be necessary to fully unravel our initial observations.

Specific comments

1. Although both the experimental methods and computational analyses are quite impressive, it was somewhat unclear what information provided by immunoGAM was unique and cannot be acquired using Hi-C based methods. Mouse Cre-lines that specifically label the three cell types were used in the study (TH-GFP for dopaminergic neurons; Sox10-Cre for oligodendrocytes and Satb2-flox for CA1 pyramidal neurons). So technically one could also perform tissue dissociation, FACS and perform cell-type-specific Hi-C experiments. It would be helpful if the authors can explicitly discuss the information that can be learned with immunoGAM but would be difficult with Hi-C.

We agree that it is important to highlight features of 3D genome topology that are specifically better enabled by GAM. We have revised the text in the introductory part to help this be more explicit. For example, we now specifically say that immunoGAM overcomes limitations of Hi-C and SPRITE technologies, by directly using selected cell types present within a complex tissue,

without the need for prior dissociation or flow sorting to isolate cell types of interest, both of which can alter cell physiology, as shown in Van den Brink et al. (2017, Single-cell sequencing reveals dissociation-induced gene expression in tissue subpopulations), a reference which we have now cited in the manuscript. GAM has additional advantages for the detection of contacts in active regions and complex contacts (investigated extensively in our recent preprint Beagrie, Thieme et al. 2021, BioRxiv). We also highlight the power of using very small cell numbers (~1000 cells), which is essential for studying precious biological samples from single individuals.

The advantages of immunoGAM can be especially appreciated in specific fields of research, such as in the neurosciences, early development, cancer, etc, where one is interested in a very specific cell type, which may be very rare or have a specific geographic localisation, or with a specific molecular feature (e.g. nuclear translocation of a TF, without changes in cell markers), all of which would be impossible with current Hi-C or SPRITE technologies. Here, we selected the PGNs specifically from the CA1 layer using their geography in the tissue. If there were other PGNs within the same region, it would not be possible to selectively isolate them by scHi-C. Furthermore, not all cells survive the harsh extractions required to dissociate cells for in-solution single-cell methods. For example, the recovery rate dopaminergic neurons in VTA in scRNA-seq or ATAC-seq experiments is ~1-2% (published data and own experience with latest scRNAseq methods), although it is known that ~15% of the VTA cells are dopaminergic neurons. If similar efficiency in DN recovery can be achieved in single-cell Hi-C, one would need to produce scHi-C data from 100,000 VTA cells (combined from ~5-10 animals for a single replicate) to get a total of the ~1,000 DNs we achieved in each of our GAM datasets from single animals. Mapping and analysing 100,000 cells by scHi-C is prohibitively expensive in most laboratories, in contrast to our current GAM datasets. For bulk Hi-C, the number of cells required to FACS sort for specific markers would be even higher. By using GAM, we can focus efficiently and robustly on obtaining high quality maps from only the cells of interest.

Another advantage of GAM is the sparse use of the biological samples, which is especially of interest in human diagnostics. As shown in the manuscript, we were able to use the same tissue samples to perform IF (Fig. 1a), GAM (Fig 1c,d and thereafter) and FISH (Fig. 3f-i).

The above-mentioned advantages make immunoGAM a method of choice in the study of specific or rare cell types of interest, and in precious samples, which is urgently needed in neuroscience and beyond. We can add some of these statements to the manuscript at the advice of the reviewer or the editor.

2. Given that the microscopic information is available for each collected cell in immunoGAM, would it be possible to correlate the cellular spatial location (e.g. spatial gradient in CA1) with chromatin conformation signature at single-cell level? If possible, such analysis would provide

unique spatially resolved (at a cellular level) chromatin organization information that cannot be obtained by Hi-C based methods.

We certainly see spatially resolved chromatin organization as a future application of GAM, especially when combined with multimodal RNA detection from the same slices, which we believe will become possible in the near future. For the first time, the results presented here demonstrate cell type specific chromatin topologies in the brain. We have described extensive differences of 3D features between cell types and showed their consistency between animal replicates. We agree with the reviewer that spatially resolved chromatin organization is an exciting approach to probe differences within the same cell type. We are actively developing the measurement of additional data modalities with GAM. As it is inherent to the LMD technology to allow recording of spatial coordinates, we do not envision major challenges in collecting spatial information about where the cells come from in future studies.

3. Can the authors explain why the mapping rate and unique mapping rate of immunoGAM data (Supplementary Table 2) are so low? Also, why was the percentage of non-unique mapping so high? On average 50%-70% of reads were mapped and ~half of the mapped reads were uniquely mapped.

The mapping rates are relatively low due to WGA by-products, which are a feature of the Multiple annealing and looping-based amplification cycles (MALBAC)-based WGA approaches that rely on random primer amplifications. The mapping rates of our home-made WGA formulation (~60%) actually improve on the mapping rates of the commercial MALBAC kit (~40%; Yikon Genomics). The low mapping rates to the mouse genome are not due to cross-contamination with other genomes (e.g. in the PGN datasets, we have ~7% of reads mapping to the human genome, and 1% for E. coli).

The % of non-unique read mapping is relatively high due to the use of Tn5 in the library preparation, which digests the same WGA fragments from different positions. Once these redundant fragments get made into different library fragments, they result in higher read redundancy. Tn5-based libraries are preferred for GAM data sequencing to increase fragment sequence variation, since all fragments produced by WGA have the same primer adaptors at each end and would require wasteful and time-consuming dark cycles in the current Illumina machines (as in Beagrie et al. 2017). By sequencing WGA fragments from internal positions, we greatly reduce the cost of sequencing and actually decreases the frequency of noise reads from absent windows from our earlier protocol (Beagrie et al. 2017). In conclusion, our mapping rates are fully inherent to the WGA and sequencing approaches used, and we have now clarified this point in the methods (see lines 1715-1719).

4. This was probably discussed in previous publications. But the usage of DeepVent for random priming is intriguing since this polymerase is not a common choice for random priming DNA synthesis compared to Klenow exo-. Also it was intriguing the amplification was not performed

with a longer reverse crosslinking incubation given these tissues were heavily crosslinked with 4% PFA perfusion.

*We are happy to clarify this point, which we have not commented upon previously. Our whole-genome amplification protocol is based on the published MALBAC protocol (Chapman et al. 2015; Zong et al. 2012), with several modifications. The most recent MALBAC publications use DeepVent DNA polymerase because it is a strand displacement polymerase, which has high fidelity and is recommended for GC-rich or looped genomic DNA sequences, has extremely high thermal stability and is easily available from commercial sources. To our knowledge, Klenow has not been tested in MALBAC approaches, possibly because it is not thermally stable and not compatible with the last steps of the linear amplification (up to 95°C). Longer reverse crosslinking incubations had no effect in mESCs where we initially developed the GAM protocol. In the present study, 160/585 GAM samples from DN samples were collected with a longer reverse cross-linking (24h) and showed mild to moderate improvements in QC metrics, such as higher total genome coverage (median = 9% and 6% for 24h and 4h, respectively) and lower percentage of orphan windows (median = 26% and 36% for 24h and 4h, respectively; **Reb. Fig. 12**, Supplemental Table 2). We have clarified this point in Methods, stating that we recommend a 24h reverse-crosslinking incubation time in future applications of GAM (lines 1753-1758).*

Rebuttal Figure 12. Comparison of reverse-cross linking incubation times for GAM quality control metrics. GAM samples produced for DN replicate 1 (R1) samples were reverse cross-linked for 4h (425 samples) or 24h (160 samples). 24h samples had moderate improvements in quality control metrics, including lower percentages of orphan windows and higher percentages of total genome coverage.

5. Are there epigenomic signatures that distinguish melted TADs, in addition to the transcriptional activation described in the manuscript? Any notable DNA or histone methylation domain or MECP2 binding? It would be especially interesting if differential epigenomic signatures can be found between adjacent TADs but with different behaviors - e.g. the downstream, but not the upstream TAD at *Nrx3* was decondensed.

We were excited to follow up this interesting suggestion. To study the regulation that underlies the melting of highly expressed genes in brain cells, we first thought of unbiasedly detecting open chromatin regions that occurred at melting genes using scATAC-seq. To this end, we produced new scATAC-seq data for DN samples in the mouse VTA (to substitute the published liDNase-seq used in the previous manuscript version), which was also an important new resource to answer this referee's concerns in point 7.

Using the new DN scATAC-seq data, along with published scATAC-seq in PGNs and OLGs, we discovered that melting is not only associated with high transcription, but also with high chromatin accessibility, sometimes occurring in genes that are not highly expressed at the mRNA level (e.g. *Csmd2* in DNs or *Pard3* in both DNs and PGNs; Fig. 2d-e,i, 3a-b, ED Fig. 5e,g). These results suggest that melting is not trivially a result of increased mRNA transcription, but can occur in response to, or lead to, increases in chromatin accessibility.

Next, we used the open chromatin regions to search for sequence motifs and enrichments for putative TF binding sites to further understand the mechanisms of domain melting. Our new scATAC-seq data was produced with recent 10X Genomics single-nucleus ATAC-seq technology, which we found suitable for TF motif searches with MEME-ChIP. In DNs, we searched for TF motif enrichments at the promoter (canonical TSS \pm 2500 bp) or at the coding region of melting versus non-melting genes. Remarkably, we found enrichment only at the promoter regions of melted genes for putative TF binding sites of a single TF, which most closely matched *Egr1* (Reb. Fig. 13). We also attempted the same analyses for PGNs, using the published scATAC-seq data, but unfortunately it was not suitable for MEME analyses, due to variable peak sizes and/or low depth. We tried several approaches to enable the analyses (e.g. centre peaks in 500bp regions as in the DN snATAC-seq, FIMO analysis from MEME suite) without success. With increased quality of single cell datasets, these types of analyses will become increasingly possible.

Rebuttal Figure 13. MEME-ChIP enrichment for DN melting genes. MEME-ChIP enrichment analysis was performed for snATAC-seq peaks in DNs R1 melting genes (> 400kb with melting score > 5) at the canonical TSS (\pm 2.5kb). Non-melting genes (> 400kb with melting score < 5) were used as the background dataset. The only significant motif enriched most closely matched *Egr1*.

As an additional exploration, we took advantage of published ChIP-seq data (Sun et al. 2019) from glutamate neurons isolated from frontal cortex (a different region), and we were initially intrigued by observing an enrichment of *Egr1* at the TSS of melted genes (Reb. Fig. 14), but this enrichment was not significant (Mann-Whitney test, $p=0.09$, comparison of reads in 250bp windows centred on the TSS), possibly because the neurons are different or *Egr1* is not the only or main TF involved in melting in PGNs.

We chose not to include the two sets of results shown in Reb. Figs. 13 and 14 (above) in the manuscript because of length, and because we feel they are observations that should be better explored in the future, in the context of perturbation, to avoid reporting accidental correlations.

Rebuttal Figure 14. Egr1 ChIP-seq read depth at the promoters of long genes. Average read depth (per 25 nucleotides) was calculated for PGN R1 melting (> 400kb with melting score > 5) and non-melting (> 400kb with melting score < 5) genes, for ± 2.5 kb surrounding the canonical TSS. The increased read depth observed for melting genes (TSS ± 250 bp) was not statistically significant (Mann-Whitney test, $p = 0.09$).

Next, we explored the association of melting with histone modifications, by taking advantage of the recently published scCUT&Tag data (Bartosovic et al. 2021) for whole brain histone marks from H3K4me3, H3K27ac, H3K36me3 and H3K27me3. These datasets were not suitable to compare with DNPs or PGNs, as their cell cluster resolution only identified neurons as ‘excitatory’ or ‘inhibitory’, each including several different neuronal subtypes. In OLGs, we could calculate read coverage in whole coding regions and found that melting tends to coincide with apparent increased enrichment in H3K4me3, which is statistically significant (Mann-Whitney test, $p < 0.001$; **Reb. Fig. 15a**). Although we found that increased H3K4me3 is positively correlated with expression in OLGs (**Reb. Fig. 15b**), as expected, the association of increased H3K4me3 with melting is not specific to the most highly expressed genes (**Reb. Fig. 15c**). Together with the observation that melting is highly correlated with either high expression or high chromatin accessibility, these observations suggest that domain melting is not simply associated with one mechanism of decondensation.

Rebuttal Figure 15. Single-cell CUT&Tag H3K4me3 in OLG long genes with domain melting. (a) Pseudo-bulk RPKM values were calculated for H3K4me3 CUT&Tag enrichment across the entire gene body (for all genes >400kb). Density plots show that genes classified as melting (melting score > 5) have significantly higher H3K4me3 RPKMs compared to non-melting genes, in both mature OLG (mOL) and OLG precursor (OPC) clusters (Mann-Whitney test, *** $p < 0.001$). (b) Correlation between H3K4me3 RPKMs (for the OPC cluster) and transcription levels (length-scaled RNA RPM; lsrRPM) for melting and non-melting genes. (c) Long genes (> 400kb) were grouped according to highest (*upper panel*) or lowest (*lower panel*) 25% of expression. Comparing melting to non-melting genes showed no significant differences in H3K4me3 RPKMs for highest/lowest expression genes (Mann-Whitney test).

Less striking, but significant, enrichments were found for H3K27ac, H3K36me3 and H3K27me3 in OLGs (**Reb. Fig. 16**). We chose not to include the results shown in Reb Figs. 15 and 16 in the manuscript because of length, but could add some of these panels if the reviewer feels they provide additional value.

Rebuttal Figure 16. Single-cell CUT&Tag H3K27ac (left), H3K36me3 (middle) and H3K27me3 (right) in OLG long genes with domain melting. Differences in melting and non-melting genes were tested by Mann-Whitney (* $p<0.05$, ** $p<0.01$, *** $p<0.001$, **** $p<0.0001$).

6. I have concerns about the resolution of TF binding motif analysis since the identification of candidate regulatory elements and TF binding motifs were performed in 50kb windows. 50kb is a very large space and can contain hundreds of putative enhancers. Permutation analysis should be performed in genomic regions containing actively expressed genes and/or containing similar numbers of putative enhancers to test whether the statistical significance of the enrichments.

We share the concerns of the reviewer, and for this reason we had only performed the TF search within the open chromatin regions of the 50kb differential contact windows. This is an essential part of our strategy which we had not made very clear. We have now explicitly measured our TF search space (i.e. total number and coverage of ATAC-seq peaks in each 50kb window) which we found covers only ~1300bp (3% of the 50kb windows). We have included this important feature in ED Fig. 7a, and have more clearly explained this aspect in the text (lines 601-609). We believe that our strategy to study differential contacts would have not worked without this restriction to the cell-type specific ATAC-seq peaks, as the referee suggested.

*We performed permutation analyses for gene-gene contacts (ED Fig. 8f), which show strikingly significant enrichments, while more clearly explaining our strategy for the differential contact analyses, and discovery of contacts that are informative of cell type. These analyses had also been requested by reviewer 1, and therefore we also show these important results for Rev. 1 point 17, **Rebuttal Fig. 10**:*

(text below is the same as in Rev. 1, point 17)

We understood from this comment that the referee is concerned about whether the results in Fig. 4 are a trivial consequence of A-A contacts. These analyses have now been strengthened in several ways including the use of permutation tests (see Rev. 2 point 6), to reassure that the associations of specific differential contacts with expressed genes in both windows (Fig. 4c) are not random nor simply explained by A-A contacts. Following the reviewer's concern, we have additionally included a permutation test to ask whether compartment A membership was sufficient to discover the striking associations with highly specialized neuronal genes. It is true, as expected and pointed out by the reviewer, that contacts with A membership are naturally more likely to contain active genes (25-28% in both contacting windows) compared with randomly permuted expressed genes (mean= 15%). However, the contacts that our approach prioritises for further analyses, through the presence of open chromatin containing putative binding sites of differentially expressed TFs, allowed us to achieve highly statistically significant enrichments for the presence of active genes in both windows (up to 40-50%). These important statistical analyses are included in ED Fig. 8f of the manuscript, and below (**Reb. Fig. 10**).

Rebuttal Figure 10. Overlap of TF-pair containing contacts with randomly permuted expressed genes. Overlap of TF-pair containing contacts with 1000 random circular permutations of PGN and DN expressed gene regions. Observed enrichments of contacts with genes in both windows are significantly higher than the expected distribution (**empirical $p < 0.001$ for all observed values tested; Z-test). The enrichments were also seen for all contacts between A compartment windows, though to a smaller level than for the TF-pair containing contacts.

TF-containing contacts are enriched for gene-gene interactions above permuted genes and A-A contacts

(end of repeated text)

7. The study reported drastically different TF regulatory networks in DN and PGN neurons (Fig. 4g). However, this analysis was vulnerable to potential biases. In particular, the denser network in PGN may be technically caused by more robust regulatory element calling, or due to the intrinsic property of PGN cells containing more active enhancers. Specifically, 1) The snATAC-seq dataset generated from PGN could be more sensitive than the LiDNase-seq database from DN. This hypothesis could be examined by comparing the number of open chromatin peaks or signal-to-noise ratio. 2) PGN cells may intrinsically contain more open chromatin sites and could lead to a denser TF network. It may be possible to set different statistical thresholds for network edges, based on sample-specific permutation, to account for the global difference in the calling of regulatory elements.

We took this concern very seriously. We produced new high-quality snATAC-seq for DNs, and revised several of the aspects of our strategy to make it more stringent. Namely, we restricted

our search to contacts within 5Mb, and to contacts that have minimum strength in the dataset where they are stronger. For the latter, we chose 0.15 NPMI as it removes lower confidence differential contacts, while not distorting the distance distribution of differential contacts (ED Fig. 7g).

a) We produced new snATAC-seq data for DNs, which contains 75% of the liDNase-seq peaks, and where we find an additional 35,000 new peaks (i.e. a two fold increase in peaks compared to the liDNase-seq peaks). With the new snATAC-seq peaks, both PGNs and DNs now have similar numbers of peaks (~53,300 and 55,300, respectively, found in at least 5% of cells).

We were pleased to see that even after major refinements to the analyses, we found similar combinations of TF motifs. We note that the top TF pair in both cell types, *Maz-Nr3c1*, is no longer in the top 20 TF feature pairs, as now both PGNs and DNs were equally enriched for this abundant TF pair. Remarkably, the new sets of TF-enriched contacts found for DNs more closely capture the expected biology of this cell type (e.g with *Foxa1* showing up as the main TF motif, instead of CTCF). We were also excited to find the enrichment of relevant genes in these contact loops, especially genes involved in addiction pathways in DNs from the VTA.

b) We appreciate the advice of the reviewer, and the revised analyses have strengthened our conclusions. First, by using the highest quality snATAC-seq data in DNs, we find a similar number of ATAC-seq regions in the sets of windows that contribute to differential contacts. Second, we tested sample-specific thresholds for the network analyses (e.g. based on the % of contacts, instead of a hard threshold of number of contacts; ED Fig. 8d). As the reviewer predicted, we now find both cell types to have interconnected networks, however the PGN networks remain more densely connected with approximately 3x more edges between TFs in PGNs (note different scale of number of contacts; ED Fig. 8e)

8. Are the genomic regions showing stronger B compartments in brain cell types associated with any epigenomic signatures such as polycomb or DNA methylation domains?

Due to length constraints, we have not been able to expand the analyses of Fig. 5, and actually decided to remove panels g-i (on single cell escapee of *Olfir/Vmn* expression). We understand this is an important topic, which we now discuss further in the text (lines 969-970) by citing the work by Tan et al. 2021, showing that the strong compartmentalization of cortical PGNs is related to "methyl-CpH deserts".

9. Line 608-609. Given the high drop-out rate of single-cell transcriptome detection, what statistical method was applied to make sure the identification of *Vmn* or *Olfir* expressing cells was reliable?

We agree that these analyses were preliminary, and decided to remove them, especially in light of the many additional analyses done to investigate the new phenomena of domain melting at long neuronal expressed genes.

10. I have concerns about the reported difference between the cells with or without Vmn and/or Olfcr was due to cellular-subtype stratification, instead of reflecting neuronal activation. The authors have made a profound claim that the subpopulation expressing Olfcr or Vmn without an in vivo perturbation experiment. It may be possible to analyze the dataset in Hrvatin 2017 Nat. Neuroscience (mouse visual cortex dark/light stimulation experiment) to see if the conclusion holds? I am not really convinced that this section, which entirely relied on analyzing published transcriptome data, is very relevant to this chromatin conformation study, especially with the lack of in vivo perturbation experiment.

We had initially included those results because we find them very exciting, but agree with the reviewer that a deeper study with perturbations will be necessary to fully unravel our initial observations.

Referee #3 (Remarks to the Author):

In the paper by Winick-Ng, Kukalev, Harabula et al, entitled "Cell-type specialization in the brain is encoded by specific long-range chromatin topologies", the authors introduce ImmunoGAM, a protocol that combines immune selection of cells with multiplexed-GAM. With ImmunoGAM, the authors have managed to study the genome three-dimensional organization of specific mouse brain types at about 50Kb resolution. This can be done with a relative low number of cells (~1,000), which makes ImmunoGAM perfect for situations where the tissue of interest is heterogeneous in cell types and contains a limited number of the required cell types. Then, the authors analysed the resulting GAM interaction matrices to assess how different these cell types in brain are compared to mouse ESC. The authors found a series of interesting results, (i) TAD organization is to a large extent not-conserved and changes between cell types, (ii) identified unique borders in each cell type was associated to genes related to the cell specialization, (iii) most long and highly expressed genes were associated to changes in these TADs into the so-called "TAD-melting", (iv) genome organization per cell type can be used to generate a network of associated Transcription Factors relevant to the biology of each cell-type, and (v) genome compartmentalization was also found to be associated to repression of specific gene families important in diseases of the brain.

I found the work well conducted but still with limited technological novelty as both GAM and PRISM are published and multiplexGAM is BioRxived.

We thank the reviewer for their comments highlighting the value of the unique biological observations revealed in this study. We have focused on expanding the toolkit for computational analyses to explore in detail the biological observations revealed in the GAM data. In particular, in our revised manuscript we have expanded our computational pipelines, and especially we now

introduce MELTRON as a novel strategy to unbiasedly discover massive changes in condensation between two cell types (or treatments).

As the reviewer mentions, immunoGAM is ideal for heterogeneous tissues, an aspect also highlighted by the other reviewers, and that is essential for its wider applicability. We anticipate that immunoGAM will be applied to many other complex tissues other than the brain, not only in mouse, but in human, both in the context of normal development (as in this paper), and in disease. Importantly, the application of GAM in a complex tissue had several challenges which needed to be overcome, which we now make clearer in the text. These are several technical advances which we fully describe in the methods and ED Fig. 1a, but have not excessively claimed in the main text, since they seem less interesting for the broad readership of Nature. For example, the ability to select a small population of ~1000 cells of interest from a complex tissue while maintaining structural preservation, to create cell-type specific contact maps, is uniquely achieved by immunoGAM. The alternative of sorting cells would likely require the pooling of several animals, may disrupt cell mechanisms (such as transcription), and can be immensely challenging for the isolation of single cell types. Producing scHi-C datasets unbiasedly for all cell types in a tissue may not always be of interest, affordable, or retain rare cell types of interest, especially in a clinical setting. All of the advancements we introduce in the current manuscript are important stepping stones to enable the use of GAM technologies in the study of rare genetic variants in precious human samples.

Beyond this general comment, there are other concerns:

1. There are many instances in the text where the authors justify statements by using “Visual inspection”. There are many published methods for dealing with the analysis of interaction matrices (arguably not all developed for but usable with GAM data) that could be used to quantify those statements. The authors should make an effort to provide statistical significance to many of their statements. This is important for many analyses but specially to GAM matrix comparison between the different studied cell types. Moreover, in all figures where GAM matrices are shown for visual comparison, it is important that the colour scale is exactly the same in all matrices.

We had used the expression ‘visual inspection’ as a way to invite the reader to look at the matrices themselves and make direct observations before moving on to genome-wide analyses. We have now avoided this expression throughout. We deeply appreciate the concerns of the reviewer about statistical analyses, and we have strengthened them extensively throughout the manuscript (Figs. 2b-c,f,h-i, 3d,g-h,k-l; ED Figs. 4d, 5c-f,h-i, 6i-m, 8f, 10d).

We have now clarified in the Figure legends and Methods (lines 1786-1788) that scale bars are adjusted. We feel that it is more sound to adjust the colour scale of NPMI matrices to a range between the 0 value and the 99th percentile, as this approach is less sensitive to differences in the numbers of samples in each dataset. NPMI effectively normalizes for over and under-

detected windows, and it inherently corrects for biases in GC content and mappability as we show for Rev 1. Point 3 (ED Fig. 2a-c, **Reb. Fig. 1**). Other studies make a similar choice. For example, Hi-C matrices from mESC and neuronal cell types are also shown with scale bar adjustments to account for the total coverage on a given chromosome in each cell type in Bonev et al. (2017; Fig. 1b therein).

(text and figure below is the same as in the response to Rev 1. Point 3)

We have now included raw heatmaps and their transformations for PGNs in Fig. ED 2a. We have provided an extended version below with more examples, together with additional normalization examples (**Rebuttal Fig. 1**).

Rebuttal Figure 1. Normalization of immunoGAM biases. (a) NPMI corrects for differences in the co-segregation matrix caused by change in the window detection frequency (WDF). Example shown for chr7:60,000,000-80,000,000. (b) Heatmaps of mean observed/expected bias in mESCs (GC content, mappability, WDF, DpnII restriction density) is shown for co-segregation, D-Prime (used in Beagrie et al. 2017), PMI and NPMI scaling. NPMI scaling results in the lowest absolute bias percentage for all tested categories (box plots on right). (c) Heatmaps of mean observed/expected bias and absolute bias percentage plots for PGNs R1.

(end of repeated text)

2. A very interesting observation is that TADs change to a large extent between cell types. This is against a somehow accepted pre-conception in the field that TADs are conserved between cell types. Therefore, the authors need to provide more evidences. Key questions are: does the Insulation Score analysis appropriate for GAM? How sensitive is to GAM data compared to Hi-C data? Would other methods result in the same observed variability? How does the resolution of the data affect the results? Using 1 bin to determine conservation is appropriate?

a) “Key questions are: does the Insulation Score analysis appropriate for GAM? How sensitive is to GAM data compared to Hi-C data? Would other methods result in the same observed variability?”

Our analysis indicates that this is indeed the case as we discuss in Rebuttal Fig. 7 in our response to Rev. 1 point 9, where we show that the insulation score method reliably detects TAD boundaries similarly between Hi-C and GAM, and to those reported in previous Hi-C studies (e.g. Bonev et al. 2017). Calling TADs with three other methods, Armatus, TADtree or TADpole, either gave very large or very few TADs in GAM. 3DNetMod currently does not accept GAM data formats. We feel that the concept of TADs is still evolving, and there are no gold standard approaches to detect them experimentally, or to call them computationally from bulk data.

(text and figure below is the same as in Rev. 1, point 9, related to Rebuttal Fig. 7)

TAD calling results with other methods. *We tested three new TAD calling methods on GAM data, which use different principles for boundary detection including TadPole (hierarchical clustering), Armatus (linear score) and TadTree (statistical modeling). We note that none of these methods and the other 20 methods tested can detect more than 2500 TADs at 50kb resolution in GM12878 cells, as shown in (Soler-Vila et al., 2020; e.g. TADpole reports ~500 TADs). We found that these methods are not appropriate to detect clear points of insulation in GAM data (which are visibly confirmed in Hi-C), either because of how they handle the data, because of assuming discrete values reflecting ligation events, or because GAM data captures more inter-TAD contacts than Hi-C (see **Reb. Fig. 7** for chr 19, mESC). We also tested a fourth method, 3DNetMod, but its current implementation is incompatible with GAM data (range of*

signal in NPMI is -1 to $+1$). We feel that the concept of TADs is still evolving, and there is no gold standard approach to detect them experimentally, or to call them computationally from bulk data. A detailed technical investigation of TAD calling would be out of the scope of the present study.

Rebuttal Figure 7. TAD calling in GAM data with additional methods. TADs were called for chr19 in Hi-C mESCs (40kb resolution; Dixon et al. 2012) and GAM mESCs (40kb resolution; Beagrie, Thieme *et al.* 2021, BioRxiv) with the Insulation square method (Crane et al. 2015 Nature), TADPole (Soler-Vila et al. 2020 Nucleic Acids Research), Armatus (Filippova et al. 2014, Algorithms for Molecular Biology) and TadTree (Weinreb and Raphael 2015 Bioinformatics). Each method gave expected outputs for Hi-C input data. For GAM data, only the Insulation square method produced TADs which visibly reflected the points of insulation between domains.

(end of repeated text)

*We have separately performed a direct comparison of TADs called with the insulation score approach in GAM and Hi-C (Beagrie, Thieme et al. 2021 BioRxiv). Below we show an extract from this preprint (Fig. 3; **Reb. Fig. 17**), that shows that most TAD borders identified in GAM are also seen in Hi-C (both in mESCs). Most (95%) GAM borders coincide with the strongest (and most robust) borders found in Hi-C, and are correctly enriched for known markers of TAD borders. We also find that there is more conservation between borders found by GAM and Hi-C than by applying two TAD calling methods to Hi-C.*

Rebuttal Figure 17 [redacted] Beagrie, Thieme *et al.* 2021 BioRxiv

b) “How does the resolution of the data affect the results? Using 1 bin to determine conservation is appropriate?”

*We expect that if we used lower resolution (e.g. 100kb) we would identify larger TADs. At higher resolution, we should expect to detect more borders, which we believe is also true in Hi-C. In the present study, we chose to cautiously work at 50kb resolution, as all six GAM datasets had very good sampling rates at this resolution. We have clarified in the main text the criteria used to call cell-type specific borders, adding the following sentence: “By considering borders to be 150kb wide, centred on the lowest point of insulation, we measured the cell-type specific borders by requiring more than 1 genomic bin (50kb) of separation between borders”. As an additional reassurance, we measured the fraction of cell-type specific borders with a faster pipeline that calculates absolute overlaps, finding 5-8% of cell-type specific borders at 500kb insulation square size (see **Reb. Fig. 6** from Rev. 1 point 9, page 11 and below), compared to the 7-10% reported in Fig 2b.*

(text and figure below is the same as in Rev. 1, point 9, related to Rebuttal Fig. 6)

Further evidence to support the claim of cell-type specific borders. *We agree that methods to identify TADs give wildly divergent results, and for this reason we have chosen to report*

insulation as a continuum at multiple length scales (100-1000kb) as an agnostic approach to report insulation (1c, 2a, 2d-e, and 3a-b; 100-1000kb insulation squares). We note that GAM data naturally captures increased contacts between TADs, as it more directly senses distances and is not restricted to capturing ligation events (see Beagrie et al. 2020 for a more extensive comparison between domains and compartments). Only in Fig. 2b,c, we have chosen to report on borders obtained from a single length (500kb) to exemplify underlying biology that accompanies the extensive changes in insulation. We chose the 500kb insulation square length as it results in similar numbers of TADs reported in other studies (2200 to 2500 in our four cell types, compared e.g. with ~2300 TADs in cortical neurons obtained with a high resolution Hi-C dataset, Bonev et al. 2017). We have strengthened the claim that many borders are cell-type specific by investigating the differences in insulation between the cell type where they are found and all the other cell types is highly significant (Fig. 2b lower row; Mann-Whitney test, p -value $< 1.8e-07$ for all comparisons). We also show that TAD border insulation in common borders is significantly increased in neuronal cell types relative to ESCs (ED Fig. 4d, p -value $< 8.6e-10$). We added SI Table 4 which provides all insulation scores, as a resource for future explorations.

As an additional reassurance, we measured the fraction of cell-type specific borders measured at additional insulation squares and found a similar proportion (4-6, 5-6, 7-9 % for 300, 400, 600kb insulation squares, respectively) compared with 5-8 % at 500kb. We illustrate that similar proportions of cell-type specific TADs are discovered irrespectively of insulation square in **Rebuttal Fig. 6** (for 500 and 600kb). The percentages shown here are slightly different than reported in Fig 2b (7-10% at 500kb), only because they were done with a faster pipeline that calculates absolute overlaps, instead of the more conservative approach chosen the manuscript that allows 1 bin separation between 150kb borders; we have not added these additional plots to the manuscript for simplicity, but can do so if preferred.

Rebuttal Figure 6. Comparison of TAD borders for different insulation square sizes. UpSet plots showing the absolute overlap of TAD boundaries called at 500kb (left) or 600kb (right) insulation square size. The pattern of overlaps is similar for both lengths, with 600kb having a slightly higher percentage of cell-type specific boundaries (5-8% and 7-9% for 500 and 600kb, respectively).

(end of repeated text)

3. The domain melting concept is interesting with respect its relationship with expression. However, I wonder whether the selection of the top 3% of the TAD melting sites is relevant to the expression levels in Fig 3g. What would happen if the authors select a similar number randomly? What if those were selected by low TAD melting scores?

We took the reviewer's comment very seriously and have extensively revised and expanded this section of manuscript (Figs. 2 and 3).

*First, to reassure the reviewer that our original results were statistically significant, we compared the association of top 3% melted genes with expression, and found it is significant ($p < 0.05$, Wilcoxon Rank Sum test; **Reb. Fig. 18**). As requested by the reviewer, we additionally tested a similar number of random genes (i.e. 15 genes), and found that the association of melting with high expression is completely lost in all cell types. We repeated the random selections, using permutation analyses where we sampled 15 genes 1000 times, and confirmed that the association of melting (top 3%) with high expression is highly statistically significant for all cell types and replicates.*

Rebuttal Figure 18. Top 3% melting score genes have increased transcription. (a)

Long genes (> 400kb) with the highest domain melting scores have significantly higher transcription (length-scaled RNA RPM; lsRRPM), compared to genes with lower melting scores (left panel; Wilcoxon Rank Sum test, $p < 0.04$). Randomly selecting 15 genes (corresponding to the top 3%) produces no enrichment in lsRRPM (middle panel). The p-value distribution

for 1000 random permutations (selecting 15 genes; right panel), shows that the observed enrichment for melting genes is highly significant for all cell types and replicates.

Nevertheless, we decided to revise our criteria for melting, following the advice from the reviewer to use differences that are statistically significant (instead of the top differences; comment 4). In the manuscript, we now describe melting of genes with melting scores >5

(equivalent to $p < 1 \times 10^{-5}$; Kolmogorov-Smirnov test), as we found this to be a threshold which purposefully distinguishes the majority of genes with no differences in insulation from genes with statistically significant differences (**Reb. Fig. 19a**, Fig. 2g-i, ED Fig. 5c-e). The expression of melted genes remains significantly higher than non-melted genes ($p < 0.01$, Wilcoxon Rank Sum test; Fig. 2g). We repeat the manuscript figure below, together with permutation test results for 1000 permutations of the same number of genes with melting scores > 5 (**Reb. Fig. 19b**).

Rebuttal Figure 19. Genes with melting scores > 5 have increased transcription. (a) Density of domain melting scores in each cell type and replicate. Applying a threshold melting score = 5 robustly filters the majority of genes that have no differences in insulation between mESCs and brain cell-types. (b) Long genes ($> 400\text{kb}$) with the domain melting scores > 5 have significantly higher transcription (length-scaled RNA RPM; lsRRPM), compared to genes with non-melting scores (*upper panel*; Wilcoxon Rank Sum test, $**p < 0.01$). The p-value distribution for 1000 random permutations (randomly selecting n = number with melting score > 5 in each cell type; *bottom panel*), shows that the observed enrichment for melting genes is highly significant for all cell-types and replicates.

4. The authors find structural differences between genomes of different cell types. Then, they sub-set the genome for regions where those differences are larger (normally using a top X% of

the differences rather than those differences that are statistically significant) and make GO analysis on those genomic regions. The GO terms show some interesting hits, but those have been (apparently) hand-picked by the authors from a larger list of other terms. Could the authors justify this selection?

We have revised our procedure of GO term selection, by reporting the GO terms with the top Z-Scores in the figures (i.e. GO term over-representation; Fig. 2c, Fig.4d-e, Fig. 5c-d, ED Fig. 3l). In Fig. 2c, we present the top four terms, plus one additional hand-picked GO term, still with significant p-values < 0.01 and Z-Scores > 2, to highlight additional genes of special interest for their known roles in cell-type specific functions (a choice we make clear in the figure legend). These can be removed at the request of the referee.

5. PRISM models show what the authors describe as “highly decondensed” regions when comparing models of mESC and DN cells. Why they say “decondensed”?

A higher radius of gyration at the model’s resolution may not reflect chromatin decondensation. Additionally, could the authors measure “torsional stress” from the models to justify the statement relating this to expression of genes?

Could the authors do additional models for long and expressed genes to find similar results? Just one region modelled may not be enough to make such statements.

To expand further our discovery of melting of long neuronal genes in brain cell types, we chose to apply an independent single cell approach that could allow us to directly visualize melting in-situ in the brain, instead of adding more in silico modelling of chromatin contacts. We developed the application of cryoFISH in the hippocampus, and went on to perform several cryo-FISH analyses of Rbfox1 in mESCs and PGNs. Using the same PGN samples as analysed by GAM, we found direct evidence that the whole gene (1.1 Mb) decondenses extensively in PGNs (area occupied expands by a factor of 2 between ESCs and PGNs). We also showed that the distance between promoter and end of Rbfox1 doubles between mESCs and PGNs (from ~370 nm to 650 nm), which confirms the decreased contacts observed in the matrices at this position. We chose the terminology ‘decondensation’ and ‘melting’ in fairness to the much earlier, pre-Hi-C reports by imaging of large-scale chromatin melting/decondensation/puffing seen in induced genes in Drosophila (e.g. Zuckerkandl E. 1976 J Mol Evol).

We hope that the referee will support our choice of strengthening the manuscript by adding an orthogonal, single-cell analysis of melting by cryoFISH at a different gene (Rbfox1) in a different tissue (PGNs), which required extensive effort in probe design and imaging, in lieu of expanding the polymer physics simulations in more genes. We fully agree that it will be extremely interesting in the future to expand the polymer modelling analyses of melting to further dissect the underlying properties and mechanisms, especially in systems that are more easily amenable to test specific predictions derived from the models by follow up experimental perturbations, such as using in-vitro differentiated cell types. By crossing our classification of domain melting with previous analyses of topoisomerase I inhibition in ex-vivo glutamatergic

neurons (ED Fig. 5f), we can add that the genes with the highest melting scores (top 3%) are significantly more likely to be downregulated upon topoisomerase-I inhibition than genes with intermediate ($\leq 3\%$ and melting score > 5) or no melting. This short exploration gives a first indication that melting may be associated with topoisomerase-dependent resolution of torsional stress, which may be an important regulatory step to enable the highest expression levels characteristic of the long genes when they are seen to melt.

6. It would also be important to show results of highly expressed genes that are shorter than 400Kb. Is the “TAD melting” effect exclusive of long genes? Why? Does it depend on how TAD borders were determined?

Melting can indeed be observed at shorter genes (see overall decrease of insulation scores over the coding region of Scna2 (~147kb; Fig. 2a), a gene with complex roles in different forms of epilepsy and autism spectrum disorder. In the present manuscript, we have chosen to focus the genome-wide study of domain melting exclusively on very long neuronal genes as we know very little about their regulation, and many have complex roles in neurons where disruptions result in serious diseases, often without effective therapies (e.g. epilepsy, autism, schizophrenia). As we began to explore the melting phenomenon even in this restricted group of 479 genes longer than 400kb, we have now uncovered so many exciting observations that we felt it was better to focus on learning as much as possible about melting for these genes, than expanding melting genome-wide, or for shorter genes. We are actively pursuing extensive studies of melting more broadly with higher resolution datasets, and in systems readily amenable to perturbations and from which we can produce robust outputs of transcript isoform expression.

Conversely, we felt it was important that melting does not become a generalisation for gene activity in brain cells, and for this reason we explicitly show the Egr1 gene (~5kb long). Egr1 is expressed in DNs and highly expressed in PGNs, and its locus becomes more condensed when most active, instead of melting (Fig. 4f). We have also now included the Pcdh cluster (where each of the 3 clusters are ~190-275kb), which when expressed also show increased long-range contacts (Fig. 1d). We find that both melting or increased condensation can accompany gene upregulation in brain cells, and the exact mechanisms are highly locus dependent and cell-type specific, which further supports the importance of producing atlas of 3D genome contacts in specialized cell types, a resource we begin with the present datasets.

The melting analysis does not rely on TAD border definitions, and it uses the whole range of insulation scores found within a large region of interest. For increased clarity, we have renamed the phenomenon to ‘domain melting’ as our analyses covered whole genes as our ‘regions of interest’ and were not specifically focused on TAD definitions.

7. The authors used ATAC-seq data (accessible DNA) and motif search as Transcription Factor (TF) binding proxy. This is a good initial approximation, however, having a motif and open

DNA does not necessarily mean that the TF factor is binding in the specific cell type. Could the authors assess, at least for a limited number of TFs, their TF networks using ChIP-Seq?

We searched for publicly available datasets for TF occupancy in brain cells, and found ChIP-seq datasets for *Egr1* in cortical glutamate neurons and *Neurod2* in mixed cells from the cortex (Sun et al. 2019, and Guner et al. 2017 for *Egr1* and *Neurod2*, respectively). We added the *Egr1* ChIP-seq track in Fig. 4f and we could confirm that *Egr1* ChIP-seq peaks coincided with the position of *Egr1* putative binding sites within ATAC-seq peaks, to strengthen the point we raise about auto-regulation of *Egr1*. A similar number of ATAC-seq peaks is found at the *Egr1* locus in DNs but no putative *Egr1* binding sites were detected, even though *Egr1* is expressed in DNs.

Even though the *Egr1* and *Neurod2* ChIP-seq data available were not produced from the same brain regions or specific cell-types as the GAM datasets, they show a reasonable genome-wide overlap with ATAC-seq regions found to contain their corresponding putative binding sites. For *Egr1*, which was collected from the same neuronal type in a different tissue, we find that 57% of ChIP-seq peaks overlap with ATAC-seq regions in PGNs (**Reb. Fig. 20**). For *Neurod2*, the ChIP-seq data was produced from mixed cells from the cortex (including non-neuronal cells), we find that 36% coincides with ATAC-seq regions. Many ATAC-seq regions with the putative TFs do not coincide with the ChIP-seq peaks, which could relate with tissue-specific binding, or with limitations of the ChIP analyses, or more complex dynamics of TF binding. Our analyses of differential contacts were aimed at discovering regions that contain many different TFs, as we felt they would more likely be important to determine cell type specific contacts but also to give less weight to individual TF putative binding events more likely to be noise. Reassuringly, we found that 39% of the *Egr1* and *Neurod2* ChIP-seq peaks that overlap with each other also coincide with ATAC-seq peaks that contain both the *Egr1* and *Neurod2* motif (758/1943 total peaks).

Overlap of PGN TF-containing ATAC-seq peaks with cortex ChIP-seq peaks

Rebuttal Figure 20. Overlap of published PGN cortex *Egr1* (upper panel) or mixed-cell cortex *Neurod2* (lower panel) ChIP-seq with single-cell ATAC-seq peaks containing putative TF-binding motifs.

We believe that searching for TF binding sites within accessible chromatin regions defined by ATAC-seq within the most cell-type specific loops provides a great exploratory tool to learn unbiasedly about the role of differentially expressed TFs in a given system (either comparing two cell types or two treatments). In the context of neuronal activation, it is immensely valuable to identify open chromatin regions containing putative TF binding sites, irrespectively of whether they are already bound; this information indicates the potential for binding, with implications for environmental interactions. For example, in PGNs many of the putative TF binding sites are found in contacts containing genes essential for synaptic activation, which would likely not be bound by specific TFs unless there was an activation event.

8. In the discussion section, the authors state that “Our results strongly support that prediction of functional or disease states in a cell type could be inferred from chromatin architecture”. It is true that this work, and many others, show correlative findings, however, those correlations may not be sufficient to make accurate predictions unless proven.

We have toned down the text to read “Our results suggest that chromatin architecture could be used to make predictions of functional or disease states in a cell type.” and we are careful to include language in the same paragraph which explicitly states that future studies are needed to clarify and test these findings (lines 940-943).

9. The header for the section “ImmunoGAM maps 3D genome architecture in specific mouse brain” misses “cell” somewhere.

We have corrected the header to read “ImmunoGAM reveals features of 3D genome architecture in specific mouse brain cells” (lines 47-48).

Reviewer Reports on the First Revision:

Referee #1 (Remarks to the Author):

The authors have added new analyses, re-analysis, and statistics to clarify or improve the technical aspects of the manuscript.

Given the critical importance of the claim that overturns the field - namely, that Mb-scale TADs are indeed massively changed across neural cell types - we still recommend that the authors repeat the full sweep of TAD analyses in a supplementary figure but using a different TAD calling method that does not rely on insulation score. It would need to be proven above and beyond that at least 2 computational methods of TAD calling could lead to the same biological conclusion of massive genome-wide changes in TADs across neural cell types. The authors used one locus to state that some other TAD callers aren't as useful for GAM. However, it seems critical to pick the second-best method and prove that the biological findings remain robust. Even a supplementary figure reproducing the main figure results with different TAD caller would go far toward addressing these concerns. Moreover, the addition of text stating that the resolution of 50 kb used for GAM in this manuscript does not call any subADs also seems important so that readers outside the field do not become puzzled by the 2500-2800 number and not realize that this number represents only the largest scale chromatin domains and not all the nested ones.

With the above computational and text changes, I believe the authors will have improved the manuscript to the quality required for publication in Nature.

Referee #2 (Remarks to the Author):

The authors have done an incredible job responding to the critiques by strengthening the statistical tests, expanding the domain melting analysis, and removed the somewhat preliminary analysis about neuronal populations expressing Olfr and Vmn. I only have three minor comments about the revised manuscript.

1. I appreciate the authors providing a detailed analysis of the type of applications that can be performed with GAM but not Hi-C. To me, the ability to select and work with a very small number of cells in their in situ context is clearly important. Also mentioned below, if the authors can indeed

show immunoGAM can identify domain decondensation with a greater sensitivity compared to Hi-C, that would be a significant advantage.

2.Line 329-332. The authors raised an important point here that large decondensation events may only be detected using immunoGAM but not Hi-C-based methods, presumably because immunoGAM can measure physical distances. I think this important point should be supported by some analyses, rather than just being speculated as a discussion point. There are multiple published datasets of mESC and mouse neurons. Can the authors perform some analysis to substantiate this claim?

3.I appreciate the manuscript included the section “Single-cell expression and chromatin accessibility maps for brain cells and mESCs” to describe the published and newly generated single-cell datasets used in the analyses. But the section should probably be shortened or move the majority of contents to material and methods.

Referee #3 (Remarks to the Author):

I appreciate the effort made by the authors to address my specific concerns on the original version of the manuscript. Specially, I would like to acknowledge the more exhaustive revision of the TAD boundaries definition, clearer explanations on the innovative aspects of the work as well as the more statistically robust analysis of “melting” and transcription.

Author Rebuttals to First Revision:

Referee #1 (Remarks to the Author):

The authors have added new analyses, re-analysis, and statistics to clarify or improve the technical aspects of the manuscript.

Given the critical importance of the claim that overturns the field - namely, that Mb-scale TADs are indeed massively changed across neural cell types - we still recommend that the authors repeat the full sweep of TAD analyses in a supplementary figure but using a different TAD calling method that does not rely on insulation score. It would need to be proven above and beyond that at least 2 computational methods of TAD calling could lead to the same biological conclusion of massive genome-wide changes in TADs across neural cell types. The authors used one locus to state that some other TAD callers aren't as useful for GAM. However, it seems critical to pick the second-best method and prove that the biological findings remain robust. Even a supplementary figure reproducing the main figure results with different TAD callers would go far toward addressing these concerns.

We thank the reviewer for their comments recognizing the improved manuscript, and appreciate the advice to further support our analyses showing extensive cell-type specific TAD borders. To our knowledge, TAD conservation has been typically performed through pairwise comparisons, in contrast to our study which identifies extensive cell-type specific borders through multi-way comparisons (Fig. 4b. Upset plot). We focused on multi-way comparisons from the outset, as we were interested in all types of conservation and specificity of TAD borders across cell types.

*Motivated by the referee's advice, we decided to explore multi-way TAD border comparisons in Hi-C data. We took two studies that collected Hi-C data in two timelines of in-vitro differentiation of neurons (glutamate neurons, GNs, Bonev et al. 2017; and dopaminergic neurons, DN, in Fraser et al. 2015, from our own lab). TAD borders were called using the insulation square method or directionality index in Bonev 2017 and Fraser 2015, respectively. When pairwise comparisons are performed using Hi-C data, we find 80-88% and 67-91% of conservation in Fraser 2015 and Bonev 2017 TAD borders, respectively, which is more conservative than the 70-85% of conservation reported previously in Fraser 2015 (**Rebuttal Figure 1**). However, when multi-way comparisons are performed instead, we find extensive reorganisation of TAD borders with typically 10% (range 3-20%) of cell-type specific borders, similar to the 7-10% we report for GAM data in Fig. 2b. Conservation between TAD borders found in each set of three cell types is only 50%, arguing that borders are much less conserved than previously appreciated. This level of conservation is nevertheless higher than the 14% reported in our study, and these differences are possibly due to the comparison between only three cell types (in contrast with four cell types in our study), or may alternatively result from the use of in-vitro differentiated neurons instead of adult brain cells.*

Rebuttal Figure 1. TAD boundary overlap in HiC datasets. Publicly available TAD boundary calls were obtained for mESCs, *in vitro* differentiated neuronal progenitors (NPCs) and *in vitro* differentiated DN or GNs (neurons) from two independent studies. (a) Pairwise and multiway comparisons for TAD boundaries called using the directionality index, obtained from Fraser et al. 2015 (‘neurons’ are *in vitro* differentiated DN; PMID: 2670085, Supplementary Table EV3). Boundaries were provided as discrete genomic coordinates, and extended by 75kb on each side to match the chosen length for GAM TAD comparisons of 150kb. (b) Pairwise and multiway comparisons for TAD boundaries called using the insulation score method, obtained from Bonev et al. 2017 (‘neurons’ are *in vitro* differentiated GNs; processed data was downloaded from 3D Genome Browser; <http://3dgenome.fsm.northwestern.edu/publications.html>). The midpoint of each boundary was extended by 75kb on each side to match the chosen length of 150kb for TAD borders in the GAM comparisons presented in the manuscript. For both studies, boundaries were considered conserved if they were separated by < 50kb edge-to-edge, using *bedtools closest* with the following parameters: *-t first -d*. Therefore, cell-type specific borders require separations of more than 200kb between their discrete genomic coordinates and the ones of their nearest neighbours. Note that the matrices of pairwise comparisons are not symmetrical, due to the direction of the comparison; the first dataset is specified on the y axis, and the second on the x axis.

We note that the fact that multi-way comparisons (in GAM or Hi-C) result in lower conservation than pairwise comparisons is not surprising, as there is no reason to assume that pairwise differences found, for example, between ESC-NPC should occur in exactly the same places as between ESC-Neuron. Therefore, once multi-way comparisons are done, an extensive cell-type specificity of TAD borders becomes undisputed, and in retrospect it is remarkable that the conservation of TADs has become such a widespread concept.

From the above arguments, we can be certain that the observation of extensive TAD reorganisation is seen in Hi-C and GAM, in three biological systems, both in vivo and in vitro, and in data obtained from different labs (the Fraser 2015 Hi-C data was produced in the lab of Josee Dostie). Importantly, TAD reorganisation is also not dependent on the TAD callers used. In Bonev 2017 and our study, the insulation square method was used for Hi-C and GAM, respectively, whereas Fraser 2015 utilized a method based on directionality index.

We agree with the referee that it is important for the readers to understand the effect of multi-way comparisons in the discovery of cell-type specific reorganization of TAD borders. To make this point explicit in the manuscript, we propose to add GAM pairwise comparisons. As shown below (**Rebuttal Figure 2** and a new panel in Extended Data Fig. 4c), the pairwise comparisons from GAM/in vivo TAD borders show conservation (79-89%) in the range described in the Hi-C literature. The addition of these pairwise comparisons, and statements in lines 228-230, will hopefully be sufficient to highlight the importance of multiway comparisons to detect extensive reorganisation of TAD borders. We may also expect that future studies with more detailed analyses of subTADs in either Hi-C or GAM, are likely to further expand the concept that TADs/domains can be highly dynamic between cell types, states, and even within cell populations as highlighted in single cell imaging experiments.

Pairwise overlap of TAD borders for is500 borders

Rebuttal Figure 2. Pairwise comparisons of TAD boundary overlaps in GAM datasets. Pairwise comparisons of overlap between TAD boundaries determined using insulation square size of 500kb. The 50kb genomic window with lowest insulation is taken and expanded by 1 bin on either side, to give TAD borders with uniform length of 150kb. The percentage of common TAD boundaries varies depending on the direction of the comparison; the first dataset is specified on the y axis, and the second on the x axis.

We feel strongly against the use of TAD callers that we know are not suitable for calling TAD borders in GAM data, and that miss many borders clearly visible in both GAM and Hi-C. We note that in our previous rebuttal, we had tested the different TAD callers across a whole chromosome (and not only for a single locus).

Referee 1 continued. Moreover, the addition of text stating that the resolution of 50 kb used for GAM in this manuscript does not call any subTADs also seems important so that readers outside the field do not become puzzled by the 2500-2800 number and not realize that this number represents only the largest scale chromatin domains and not all the nested ones.

We appreciate the reviewer's advice to more clearly define the type of domains called by the insulation score method. We have added a sentence in the methods on lines 1808-1810 to read "This approach does not detect meta-TADs or sub-TADs, and results in numbers and lengths of domains similar to previous reports^{5,8}." We have also clarified the text on lines 223-224 to read "At 50kb resolution, we detected domains with similar numbers and lengths of TADs found in all cell types...".

We have further clarified the justification for using the insulation score method on lines 131-134 which now reads "... we calculated contact densities and topological domains with the insulation square method¹⁸, where it was previously shown that the domain borders detected in GAM data are also found in Hi-C, where they are the most robust (most insulated)^{3,11} ..."

Referee 1 continued. With the above computational and text changes, I believe the authors will have improved the manuscript to the quality required for publication in Nature.

We value the reviewer's advice to further expand the arguments that support the claim of extensive domain organisation. We hope the reviewer trusts that we see the value of investigating TAD border positions and their meta- and sub-structures, and the need for the development of robust computational methods to study them. We hope they will support us in our concerns to apply methods which we strongly believe are currently not suitable for GAM data.

Referee #2 (Remarks to the Author):

The authors have done an incredible job responding to the critiques by strengthening the statistical tests, expanding the domain melting analysis, and removed the somewhat preliminary analysis about neuronal populations expressing *Olfir* and *Vmn*. I only have three minor comments about the revised manuscript.

We are grateful for the reviewer's appreciation of our revision, and the encouragement to directly address whether melting of long neuronal genes can be detected in Hi-C.

I appreciate the authors providing a detailed analysis of the type of applications that can be performed with GAM but not Hi-C. To me, the ability to select and work with a very small number of cells in their in situ context is clearly important. Also mentioned below, if the authors can indeed show immunoGAM can identify domain decondensation with a greater sensitivity compared to Hi-C, that would be a significant advantage.

*We took published Hi-C datasets of mESC and mouse neurons from Bonev et al. 2017, and we applied the MELTRON pipeline. The datasets that were freely available to download were from mESCs and in vitro differentiated glutamatergic neurons (GNs; data downloaded with Juicer, a full list of available datasets can be found at: <https://hicfiles.tc4ga.com/juicebox.properties>). As suggested by the reviewer, we detected fewer melting genes in Hi-C (21 genes, compared with 142 in adult PGNs R1 using GAM) with a maximum melting score of 31, compared with a maximum melting score of 79 in GAM PGN data (**Rebuttal Figure 3**). Using matching expression data (Bonev et al 2017, available from the GEO repository under accession numbers GSM2533847 and GSM2533848), the genes found to be melted in Hi-C GNs were significantly less expressed than the average non-melting long gene, although 11 of the 21 genes were shared with in-vivo PGNs in GAM.*

Rebuttal Figure 3. Domain melting in Hi-C in-vitro differentiated glutamatergic neurons. Hi-C data for mESCs and *in-vitro* differentiated glutamatergic neurons (GNs) was obtained from Bonev et al. 2017 (processed data was downloaded from a public server with Juicer: http://hicfiles.s3.amazonaws.com/external/bonev/ES_mapq30.hic; http://hicfiles.s3.amazonaws.com/external/bonev/CN_mapq30.hic). Insulation scores (range 100-1000kb) were calculated for each cell type, as previously described (Crane et al. 2015). Next, the MELTRON pipeline was used to detect differences in insulation between mESCs and GNs for genes >400kb. Genes with melting scores > 5 were considered as ‘melting genes’. To explore the association of genes melting in Hi-C with gene expression, we downloaded matched expression data from the same neurons (Bonev et al. 2017; available from the GEO repository under GSM2533847 and GSM2533848 for 2 replicates of GNs). Melting scores compared to transcription data are shown for *in-vivo* GAM PGNs (R1; *left*) and *in-vitro* Hi-C GNs (*right*). Hi-C detects 21 melting genes, compared to 142 melting genes in GAM PGNs R1. The 21 melting genes detected in Hi-C have significantly lower transcription compared to non-melting genes (*p=0.02, Wilcoxon rank-sum test). The GN melting gene ‘*Iqcm*’, and 15 non-melting genes are not shown in the Hi-C plot, as there is no matching transcription data available.

Examples of genes melting in Hi-C, and Grik2 which is not melting in Hi-C, are shown in Rebuttal Figure 4. Together, these results suggest that Hi-C indeed has less sensitivity to detect melting events at long genes than GAM.

Rebuttal Figure 4. Examples of melting genes in Hi-C, and *Grik2*, which melts in GAM. Hi-C matrices and cumulative probability plots for mESCs and *in vitro* differentiated glutamatergic neurons. (a) Hi-C matrices for *Dnah7c* (Chr1:44,900,000-48,350,000), which has the highest melting score of 31, for all genes > 400kb. The cumulative probability of insulation square scores ranging from 100-1000kb is shown for mESCs compared to glutamatergic neurons (lower panel), with the indicated maximum distance (*d*) used to calculate the melting score. *Dnah7c* is found melting by GAM in PGNs (melting score 7 and 24 in PGNs R1 and R2, respectively) (b) Matrices (Chr14:91,500,000-95,400,000) and cumulative probability plot for *Pcdh9*, which has the highest expression of any gene melting in the Hi-C neurons but low melting score of 9. *Pcdh9* is found melting by GAM in PGNs R2 only (melting score 45) (c) Matrices (Chr10:48,000,000-50,500,000) and cumulative probability plot for *Grik2*, which is melting in GAM PGNs (melting score 13 and 26 in PGNs R1 and R2, respectively) but not in the Hi-C GNs.

It remains possible that the lower sensitivity of detecting melting events using Hi-C is not due to the technology itself, but to melting being a property of terminally differentiated cells which may not be seen in immature neurons differentiated in vitro. [redacted]

Redacted - Rebuttal Figure 5

We believe that the melting phenomena is likely to encompass a number of different mechanisms, as described in the manuscript, for example in association with increased transcription, and/or open chromatin, often but not always involving large-scale reorganisation relative to their chromosome territories. Future detailed investigations of melting will be necessary to distinguish effects of cell type, cell maturation, type of gene, or genome context. For this reason, we favour that we do not include in the present manuscript claims that Hi-C is less able to detect melting. We confirm that we will agree to leave them in the public open reviews, or add them to the manuscript if advised by the reviewer or editor, [redacted]

2.Line 329-332. The authors raised an important point here that large decondensation events may only be detected using immunoGAM but not Hi-C-based methods, presumably because immunoGAM can measure physical distances. I think this important point should be supported by some analyses, rather than just being speculated as a discussion point. There are multiple published datasets of mESC and mouse neurons. Can the authors perform some analysis to substantiate this claim?

We thank the reviewer for this important advice, and hope that the analyses presented in point 1 and Rebuttal Figures 3-5 help to clarify the statement that GAM detects melting with higher

sensitivity than Hi-C. We have decided to remove the speculation about why melting has not been previously reported in Hi-C from the manuscript (see strikethrough text in the revised manuscript on lines 339-340), as we do not yet know the explanation of why melting is not strongly detected in Hi-C.

3.I appreciate the manuscript included the section “Single-cell expression and chromatin accessibility maps for brain cells and mESCs” to describe the published and newly generated single-cell datasets used in the analyses. But the section should probably be shortened or move the majority of contents to material and methods.

We agree that the single-cell section is quite technical, especially as most of the details were already described in the methods. We have reduced the text size to still introduce the origin of the single cell datasets, but only minimally present the quality controls (please see lines 178-207; text suggested for removal is indicated with a strikethrough).

Referee #3 (Remarks to the Author):

I appreciate the effort made by the authors to address my specific concerns on the original version of the manuscript. Specially, I would like to acknowledge the more exhaustive revision of the TAD boundaries definition, clearer explanations on the innovative aspects of the work as well as the more statistically robust analysis of “melting” and transcription.

We are grateful for the reviewer’s previous advice and the validation that we took it on board appropriately.

Note for all referees:

*We have added new Supplemental Tables 9 and 10 which provide X,Y,Z coordinates for each polymer model structure for the Nr3x3 locus, derived from mESC or DN GAM data, respectively (related to Fig. 3c-d, and ED Fig. 6c-d). We have added a sentence to the methods on lines 2191-2193 which reads “A full list of X,Y,Z coordinates for mESC and DN polymer model structures can be found in **Supplemental Tables 9 and 10**, respectively.”*

Reviewer Reports on the Second Revision:

Referee #1 (Remarks to the Author):

The authors have done a large amount of work for the two revisions, and the manuscript is unequivocally high quality. I remain concerned about the phrasing throughout the main, and supplement, of stating that there are 2k-3k TADs - and my own scientific opinion is that the manuscript would be more accurate and in-line with the field if it instead prominently and explicitly stated something to the effect of: "We focused specifically on Mb-scale TADs and identified (~2.5?k genome-wide with mean size ~650kb). We discovered at this length scale. We note that subTADs (or nested contact domains) and Mega-TADs were not analyzed, and that our numbers for TAD-scale structures specifically is inline with previous reports." This seems accurate, clears up confusion, prevents creating future confusion, and in no way detracts from the tour de force work. However, I also acknowledge that this is my preference, and my opinion, and the authors/editor should ascertain what they deem best for their paper and the field.

Referee #2 (Remarks to the Author):

The authors have successfully addressed all my comments.

Author Rebuttals to Second Revision:

Referee #1 (Remarks to the Author):

The authors have done a large amount of work for the two revisions, and the manuscript is unequivocally high quality. I remain concerned about the phrasing throughout the main, and supplement, of stating that there are 2k-3k TADs - and my own scientific opinion is that the manuscript would be more accurate and in-line with the field if it instead prominently and explicitly stated something to the effect of: "We focused specifically on Mb-scale TADs and identified (~2.5?k genome-wide with mean size ~650kb). We discovered at this length scale. We note that subTADs (or nested contact domains) and Mega-TADs were not analyzed, and that our numbers for TAD-scale structures specifically is inline with previous reports." This seems accurate, clears up confusion, prevents creating future confusion, and in no way detracts from the tour de force work. However, I also acknowledge that this is my preference, and my opinion, and the authors/editor should ascertain what they deem best for their paper and the field.

We thank the reviewer for their comments recognizing the high quality of the manuscript, and appreciate their concern to unequivocally present the TAD border analyses in a way which cannot be mis-interpreted. We agree that there is value in preventing any confusion in the methods that we have applied in this study.

We have added and revised the following paragraph (see lines 1475-1486):

“TAD calling was performed by calculating insulation scores in NPM1 GAM contact matrices at 50 kb resolution, as previously described^{2,9}. The insulation score method was chosen as it was previously shown that the domain borders detected in GAM data are also found in Hi-C, where they are the most robust (most insulated)^{2,9}. The insulation score was computed individually for each cell type and biological replicate, with insulation square sizes ranging from 100 to 1000kb. TAD boundaries were called using a 500kb-insulation square size and based on local minima of the insulation score. This approach does not detect meta-TADs or sub-TADs, and results in numbers and lengths of domains similar to previous reports^{6,58}. Future work with higher resolution GAM datasets will enable further analyses of the reorganisation of domains at finer genomic scales to investigate changes in sub-TADs, shown previously to occur following cell commitment to neuronal lineages⁵⁹.”

Due to additional length constraints to comply with Nature policies, we have limited this discussion to the material and methods section, and hope that the reviewer and editor will understand and trust this decision. If requested we can move some or all of these statements to the main text.

Referee #2 (Remarks to the Author):

The authors have successfully addressed all my comments.

We are grateful for the reviewer’s advice through both revisions, and the validation that we have fully addressed all of their comments.